# Resolving discrepancies between chimeric and multiplicative measures of higher-order epistasis

Uthsav Chitra[1,3], Brian Arnold[1,2,3] & Benjamin J. Raphael [1]✉

Epistasis - the interaction between alleles at different genetic loci - plays a fundamental role in biology. However, several recent approaches quantify epistasis using a chimeric formula that measures deviations from a multiplicative fitness model on an additive scale, thus mixing two scales. Here, we show that for pairwise interactions, the chimeric formula yields a different magnitude but the same sign of epistasis compared to the multiplicative formula that measures both fitness and deviations on a multiplicative scale. However, for higher-order interactions, we show that the chimeric formula can have both different magnitude and sign compared to the multiplicative formula. We resolve these inconsistencies by deriving mathematical relationships between the different epistasis formulae and different parametrizations of the multivariate Bernoulli distribution. We argue that the chimeric formula does not appropriately model interactions between the Bernoulli random variables. In simulations, we show that the chimeric formula is less accurate than the classical multiplicative/additive epistasis formulae and may falsely detect higher-order epistasis. Analyzing multi-gene knockouts in yeast, multi-way drug interactions in E. coli, and deep mutational scanning of several proteins, we find that approximately 10% to 60% of inferred higher-order interactions change sign using the multiplicative/additive formula compared to the chimeric formula.

A key problem in biology is deriving the map from genotype to fitness, or the average reproductive success of a genotype. This map is often referred to as the *fitness landscape* (first conceptualized in ref. 1). In its simplest form, the fitness effects of alleles at one locus are independent of those at other loci, such that multilocus fitness is either an additive or multiplicative function of the alleles across loci. However, the fitness landscape is complicated by the presence of *epistasis*, or genetic interactions where alleles at one locus modify the effects of alleles at other loci. Epistatic interactions reveal functional relationships between genes, with the sign of an epistatic interaction (positive or negative) often used to understand how genes are organized into genetic pathways[2], model protein function, and evolution[3],

understand mechanisms of antibiotic resistance[4], and interpret genome-wide association studies (GWAS)[5].

While epistasis is a property of any quantitative trait, many studies have measured epistatic interactions using experimental fitness data from haploid genomes (reviewed in refs. 6–8). Most of these studies measure *pairwise epistasis*, or an interaction between a pair of genetic loci that is computed by comparing the observed fitness of the double-mutant to the expected fitness under a null model with no epistasis. The *sign* of the epistatic interaction is determined by whether the observed fitness is greater than or less than the expected fitness, resulting in a positive or negative interaction, respectively. The choice of the null model for the expected fitness depends on the quantitative

[1]Department of Computer Science, Princeton University, Princeton, NJ, USA. [2]Center for Statistics and Machine Learning, Princeton University, Princeton, NJ, USA. [3]These authors contributed equally: Uthsav Chitra, Brian Arnold. ✉e-mail: braphael@princeton.edu

trait used as a proxy for fitness. Nearly all formulae for pairwise epistasis assume either an additive null model, where the expected fitness is the sum $f_{01}+f_{10}$ of the fitness values of the single-mutants, or a multiplicative null model, where the expected fitness is the product $f_{01}f_{10}$ of the fitness values of the single-mutants. For example, an additive null model is often used when fitness is measured using cellular growth rate (e.g., in fitness assays of microbes[9]) or fluorescence (e.g., in proteins[10]), while a multiplicative null model is typically used when fitness is measured by the total size of a clonal population[11]. Under an additive null model, epistasis $\epsilon$ is computed as the difference $\epsilon = f_{11} - (f_{10}+f_{01})$ between observed and expected double-mutant fitness.

For the multiplicative null model, there is no agreement in the literature about how to quantify deviation from the null model. In the statistics literature, it is standard to compute multiplicative interaction effects using a *ratio* $\epsilon = \frac{f_{11}}{f_{01}f_{10}}$ between the observed and expected values (e.g., refs. 12,13). On the other hand, many studies in the genetics literature compute epistasis as the *difference* $\epsilon = f_{11} - f_{01}f_{10}$ between observed and expected fitness values of a double-mutant (e.g., refs. 14–23). We call the first formula the *multiplicative formula*, as it preserves the multiplicative measurement scale, while we call the second formula the *chimeric formula*, as it measures deviations from a multiplicative model on an additive scale and thus is a "chimera" of additive and multiplicative scales.

Here, we show that the chimeric and multiplicative formula result in different quantitative measures of pairwise epistasis, which may affect findings on the strength of an epistatic interaction. Nevertheless, we also show that the two formulae always yield the same *sign* (or *direction*) of a pairwise interaction. The sign is often the quantity of interest in genetics studies, e.g., negative epistatic interactions are used to quantify functional redundancy[24,25]. Thus, the focus of existing literature on the sign of interactions, as well as the focus on pairwise epistasis, may explain why the differences between the multiplicative and chimeric formula are not broadly recognized.

The discrepancies between the multiplicative and chimeric formula are more consequential for *higher-order* interactions between three or more loci, which are becoming more widely studied with larger genetic datasets and high-throughput measurements of fitness[24,26–28]. Recent studies in yeast genetics[24,26] and antibiotic resistance[27] independently derived analogous chimeric formula to quantify epistasis between three or more loci and higher-order interactions between components, respectively, under a multiplicative fitness model. These chimeric formulae were derived de novo and without consideration of the two distinct formula − chimeric and multiplicative − for pairwise epistasis, nor the consequences of conflating multiplicative and additive scales. However, unlike in the pairwise setting, we show that for three or more loci, the chimeric formula is *not* guaranteed to produce the same *sign* of an interaction as the multiplicative formula. Thus, the chimeric formula may indicate a *positive* epistatic interaction while the multiplicative formula shows a *negative* epistatic interaction, and vice-versa. Such inconsistencies raise questions about the validity of reported higher-order epistasis in biological applications.

We resolve the mathematical and biological inconsistencies between the different epistasis formulae by deriving connections between epistasis and the parameters of the multivariate Bernoulli distribution (MVB), a probability distribution on binary random variables[29]. In particular, we show that a wide array of approaches for quantifying epistasis – including the additive, multiplicative, and chimeric formulae, as well as the regression models commonly used in GWAS and QTL analyses[2,5] and the Walsh coefficients for measuring background-averaged epistasis[30–32] – are equivalent to computing different parameterizations of the MVB, showing that the MVB provides a unifying statistical framework for the different epistasis measures.

We use the connections to the multivariate Bernoulli distribution to analyze the higher-order (i.e., ≥ 3-way interactions) chimeric epistasis formulae derived by Kuzmin et al.[24,26] and Tekin et al.[27]. We show that the chimeric formulae for pairwise epistasis and the chimeric formulae for higher-order epistasis correspond to the joint cumulants of the MVB, a concept from probability theory for measuring interactions between continuous variables[33]. However, the joint cumulant is known to *not* be an appropriate measure of higher-order interactions for binary random variables[34,35]. Accordingly, we argue that the chimeric epistasis formula are not appropriate for measuring higher-order epistasis between biallelic mutations. In this way, just like how the hero Bellerophon in the *Iliad* slayed the monstrous chimera, the multivariate Bernoulli distribution allows us to "slay" the chimeric epistasis formula.

We demonstrate that the mathematical issues with the chimeric epistasis formula lead to markedly different biological interpretations of perturbation experiments using haploid genomes. Analyzing multigene knockout data in yeast using the more appropriate multiplicative formula changes the sign of 12% of the 7957 trigenic interactions that Kuzmin et al.[24,26] reported using the chimeric formula. Many of these sign changes are concentrated on negative interactions, which are more functionally informative than positive interactions and are commonly used to measure functional redundancy between genes[25]. In particular, the multiplicative epistasis formula identifies nearly 500 negative interactions not reported by Kuzmin et al.[24,26] that are significantly enriched for several measures of functional redundancy, thus extending the trigenic interaction network by 25%.

We further demonstrate that the multiplicative and additive formulae yield markedly different interactions compared to the chimeric formula in two other applications: the identification of higher-order synergistic and antagonistic drug interactions in *Escherichia coli* and the identification of epistatic interactions between protein mutations in deep mutational scanning (DMS) experiments which is important for 3-D protein structure prediction[36,37], protein engineering[7], genome editing optimization[38], variant effect prediction[39], and other applications. Notably, we show that the discordance between the different formulae increases with interaction order: the additive formula shows significantly less antagonism between five-way interactions compared to the chimeric formula used in ref. 40, while for some proteins there is substantial (up to 60%) disagreement in the sign of interaction between the multiplicative and chimeric formulae.

## Results

### Pairwise epistasis: additive, multiplicative, and chimeric

Pairwise epistasis describes interactions between two genetic loci. We consider haploid genomes and assume that each locus is *biallelic*, i.e., each locus has two alleles labeled 0 and 1. Thus for a pair of loci there are four possible genotypes: the wild-type 00, the single mutants 01 and 10, and the double mutant 11. Accordingly, for a pair $(i, j)$ of loci there are four corresponding fitness values: the wild-type fitness $f_{\varnothing}$, corresponding to the wild-type genotype 00 with no mutations; the single-mutant fitnesses $f_i, f_j$, corresponding to the genotypes 01 and 10 with either locus $i$ or locus $j$ mutated, respectively; and the double-mutant fitness $f_{ij}$, corresponding to the genotype 11 with both loci mutated. Pairwise epistasis is measured by comparing the observed double-mutant fitness $f_{ij}$ to the expected fitness under a null model with no epistasis.

In practice, the fitness $f$ of a genotype cannot be directly measured. Instead, experiments typically measure traits that are expected to be highly correlated with fitness. For example, in populations of microbes or proteins, it is standard to estimate the fitness of a genotype with either its population growth rate or relative frequency, which are typically assumed to follow an additive or multiplicative scale, respectively ("Methods"). Accordingly, the two standard null models of

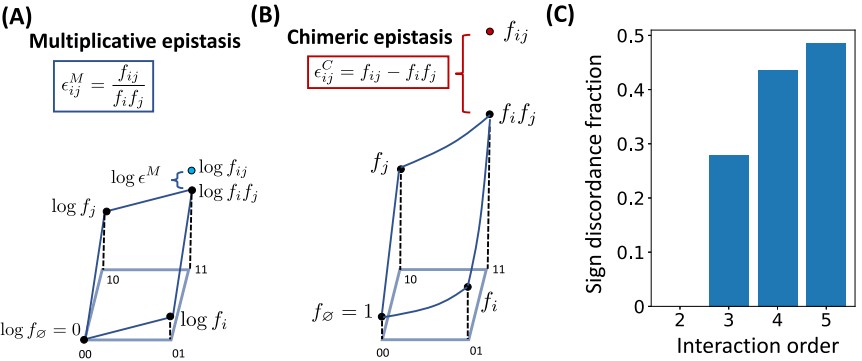

**Fig. 1 | Comparison of multiplicative and chimeric epistasis measures. A** For a pair $(i, j)$ of loci, the multiplicative epistasis measure $\epsilon_{ij}^M = \frac{f_{ij}}{f_i f_j}$ is the *ratio* between the observed fitness $f_{ij}$ of the double mutant and the expected fitness $f_i f_j$ under a multiplicative null model. Equivalently, the logarithm $\log \epsilon_{ij}^M$ of the epistasis measure is given by $\log \epsilon_{ij}^M = \log f_{ij} - \log f_i f_j$, or the *difference* between the observed and expected values in log-fitness space. **B** The chimeric epistasis measure $\epsilon_{ij}^C = f_{ij} - f_i f_j$ is the *difference* between the observed and expected fitness values of the double-mutant under a multiplicative fitness model. The chimeric measure $\epsilon_{ij}^C$ thus mixes scales by measuring deviations from multiplicativity on an additive scale. **C** The fraction of instances where the signs $\mathrm{sgn}(\log \epsilon^M)$ and $\mathrm{sgn}(\epsilon^C)$ of the multiplicative and chimeric fitness formula, respectively, disagree ("sign discordance fraction") for interaction orders $L = 2, ..., 5$, where fitness values $f_i, f_{ij}, ...$ are sampled uniformly at random from the interval $[0, 1]$. For two loci, the sign of the two measures always agree (see Proposition 1), but for more than two loci, there is substantial disagreement.

fitness for measuring epistasis are the *additive* model and the *multiplicative* model.

In the *additive* model, mutations are assumed to have an additive effect on fitness, and the pairwise epistasis measure $\epsilon_{ij}^A$ is equal to the difference between the observed and expected double-mutant fitness values:

$$\epsilon_{ij}^A = f_{ij} - (f_i + f_j), \tag{1}$$

under the assumption that fitness values are normalized such that the wild-type fitness $f_\varnothing = 0$. The sign of the interaction (i.e., positive vs. negative) is given by the sign $\mathrm{sgn}(\epsilon_{ij}^A)$ of the epistasis measure $\epsilon_{ij}^A$. The additive model was first posed by Fisher[41], who used the term "epistacy" to refer to any statistical deviation from additivity[6].

In the *multiplicative* fitness model, mutations have a multiplicative effect on fitness, and the multiplicative pairwise epistasis measure $\epsilon_{ij}^M$ is given by the ratio between the observed and expected double-mutant fitness values:

$$\epsilon_{ij}^M = \frac{f_{ij}}{f_i f_j}, \tag{2}$$

under the typical assumption that the wild-type fitness $f_\varnothing$ is equal to 1. The sign of the interaction is determined by whether the multiplicative measure $\epsilon_{ij}^M$ is greater than or less than 1. Moreover, if fitness values $f$ are multiplicative, then the *log*-fitness values $\log f$ are additive; thus, the sign of interaction under the multiplicative model is also given by the sign $\mathrm{sgn}(\log \epsilon_{ij}^M)$ of the log-epistasis measure $\log \epsilon_{ij}^M$. The additive and multiplicative epistasis measures are closely related to the linear/log-linear regression frameworks[5,12] and the Walsh coefficients[30–32,42,43] used in the genetics literature; see Methods for details.

Curiously, there is a third epistasis formula that is widely used for the multiplicative fitness model. Here, deviations from the multiplicative model are measured on an additive scale, resulting in the following *chimeric* formula for pairwise epistasis:

$$\epsilon_{ij}^C = f_{ij} - f_i f_j. \tag{3}$$

We refer to $\epsilon_{ij}^C$ as the chimeric epistasis measure because it measures deviations from a multiplicative null model on an additive scale and is thus a *chimera* of both the multiplicative and additive measurement scales. As in the additive model, the sign of the interaction is given by the sign $\mathrm{sgn}(\epsilon_{ij}^C)$ of the chimeric measure $\epsilon_{ij}^C$.

The chimeric epistasis measure $\epsilon_{ij}^C$ appears in the genetics literature (e.g., refs. [14–23]) and in the drug interaction literature (e.g., refs. [27,40,44,45]) because of its interpretation as a *residual*, i.e., the difference between the observed and expected values of a measurement. However, despite the simplicity of this explanation, residuals are typically only appropriate for additive models. For multiplicative models, it is standard to compute statistical interactions using the *ratio* between observed and expected measurements (as in equation (2)), rather than the difference[12]. Moreover, Wagner[13,46] notes that preserving the multiplicative measurement scale (by using the ratio) is required in order to guarantee meaningful notions of statistical and functional interactions.

While both the chimeric measure $\epsilon_{ij}^C$ and the multiplicative measure $\epsilon_{ij}^M$ are described as measuring deviations from a multiplicative fitness model, the two measures are not equal. In particular, the (log-) multiplicative epistasis measure $\log \epsilon_{ij}^M = \log f_{ij} - \log f_i f_j$ computes the difference between the observed and expected double-mutant fitness values on a logarithmic scale (Fig. 1A) while the chimeric epistasis measure $\epsilon_{ij}^C = f_{ij} - f_i f_j$ computes the difference directly (Fig. 1B). When the double-mutant fitness $f_{ij}$ and single-mutant fitness values $f_i, f_j$ are close to 1, we show that the chimeric measure $\epsilon_{ij}^C$ is approximately equal to the log-multiplicative measure $\log \epsilon_{ij}^M$ (Supplementary Note 1). However, if the fitness values are substantially different from 1, then the chimeric epistasis measure $\epsilon_{ij}^C$ may over- or under-state the strength of a pairwise interaction in a multiplicative fitness model as we demonstrate numerically (Supplementary Note 2).

Nevertheless, we prove ("Methods") that the chimeric measure $\epsilon_{ij}^C$ has the same *sign* of an interaction as the multiplicative measure $\epsilon_{ij}^M$ but not the same *magnitude*. Thus, using either the chimeric or multiplicative measures will not affect findings that depend on the *sign* of an epistatic interaction, and the sign is often the quantity of interest (e.g., negative epistasis is used to quantify functional redundancy[21]). However, the agreement between the multiplicative and chimeric measures on the sign of interaction is true only for pairwise epistasis and not higher-order epistasis, as we will show below.

### Higher-order epistasis
For higher-order epistasis, or interactions between three or more genetic loci, we find that the difference between the multiplicative and chimeric epistasis measures are more consequential. Under the multiplicative fitness model, the three-way epistasis measure $\epsilon_{ijk}^M$ between loci $i, j, k$ is given by the ratio between observed and expected triple-

**Table 1 | Correspondence between epistasis measures and the parametrizations of the multivariate Bernoulli distribution when the fitness values are proportional to the indicated quantities of the distribution**

|  | Parameters of multivariate Bernoulli distribution | Fitness values **f** proportional to |
|---|---|---|
| Additive epistasis measure $\epsilon^A$ | Natural parameters $\boldsymbol{\beta}$ | Log-probabilities log **p** |
| Multiplicative epistasis measure $\epsilon^M$ | Natural parameters $\boldsymbol{\beta}$ | Probabilities **p** |
| Chimeric epistasis measure $\epsilon^C$ | Joint cumulants $\kappa$ | Moments $\boldsymbol{\mu}$ |
| Walsh coefficients | Moments of $(1 - 2X_1, ..., 1 - 2X_L)$ | Probabilities **p** |

mutant fitness:

$$\epsilon^M_{ijk} = \frac{f_{ijk}}{f_i f_j f_k \epsilon^M_{ij} \epsilon^M_{ik} \epsilon^M_{jk}} = \frac{f_{ijk} f_i f_j f_k}{f_{ij} f_{ik} f_{jk}}. \quad (4)$$

Recent work in the yeast genetics[24,26] and antibiotic resistance[27] literature claim to use a multiplicative fitness model, but derive a different epistasis formula:

$$\epsilon^C_{ijk} = f_{ijk} - (f_i f_j f_k + \epsilon^C_{ij} f_k + \epsilon^C_{ik} f_j + \epsilon^C_{jk} f_i), \quad (5)$$

where $\epsilon^C_{ij}, \epsilon^C_{ik}, \epsilon^C_{jk}$ are the pairwise chimeric epistasis measures in (3). Note that as in the pairwise case, formula (5) mixes the additive and multiplicative scales in a complex manner. Thus, we refer to $\epsilon^C_{ijk}$ as the *chimeric* three-way epistasis measure.

As in the pairwise setting, the three-way chimeric measure $\epsilon^C_{ijk}$ in (5) is clearly different from the three-way multiplicative measure $\epsilon^M_{ijk}$ in (4). However, we show that these formula often differ in both the magnitude of epistasis (as in the pairwise setting) *and* in the sign of epistasis. Thus, one formula may indicative positive epistasis between three loci while another formula may indicate negative epistasis, and vice-versa. In simulations, we find that approximately 28% of triples have different signs between the two formulae (Fig. 1C).

Tekin et al.[27] extended the three-way chimeric epistasis formula (5) to compute a 4-way chimeric epistasis measure $\epsilon^C_{ijkl}$ and a 5-way chimeric epistasis measure $\epsilon^C_{ijklm}$. We find even more substantial differences in the sign of epistasis between these 4-way and 5-way chimeric epistasis measures and the 4-way and 5-way multiplicative epistasis measures (Equation (18) in "Methods"). In simulations, only approximately 57% and 52% of 4-way and 5-way interactions, respectively, have the same sign using the chimeric and multiplicative epistasis formulae (Fig. 1C).

This substantial disagreement between the chimeric and multiplicative epistasis measure motivates a deeper mathematical understanding of the various epistasis formulae, which we undertake in the next section.

**Unifying epistasis measurements with the multivariate Bernoulli distribution**

A genotype of biallelic mutations on $L$ loci can be represented as a binary string of length $L$, where 0 corresponds to the wild-type allele, and 1 corresponds to the mutant, or derived, allele. For example, the string 01100 represents the genotype of $L = 5$ loci with mutations in the second and third loci. The fitness values of all genotypes, often referred to as the fitness landscape, correspond to a function $f$ that maps a binary string $\mathbf{x} \in \{0, 1\}^L$ to its fitness $f_{\mathbf{x}}$.

A natural approach for studying a fitness landscape function $f$ is to view it as a *distribution* on the set $\{0, 1\}^L$ of binary strings, where the probability $p_{\mathbf{x}}$ of a binary string $\mathbf{x}$ is derived from its fitness $f_{\mathbf{x}}$. Such distributions are often used by protein structure models[39]. Moreover, many real-world fitness datasets – including the yeast fitness data and many of the protein datasets analyzed in this manuscript – measure

the fitness of a genotype $\mathbf{x}$ in terms of its relative frequency in a large population of genotypes, i.e., its probability $p_{\mathbf{x}}$.

Here, we model the fitness landscape using the *multivariate Bernoulli* (MVB) distribution[29,47] which describes *any* distribution on the set $\{0, 1\}^L$ of binary strings. Formally, a multivariate random variable $(X_1, ..., X_L)$ distributed according to a MVB is parametrized by the probabilities $p_{\mathbf{x}} = P((X_1, ..., X_L) = \mathbf{x})$ for each binary string $\mathbf{x} = (x_1, ..., x_L) \in \{0, 1\}^L$. We model the genotype $(X_1, ..., X_L)$ of an organism as a random variable distributed according to a MVB parametrized by the probabilities $\mathbf{p} = (p_{\mathbf{x}})_{\mathbf{x} \in \{0, 1\}^L}$.

We prove that the additive, multiplicative, and chimeric measures of epistasis – as well as the Walsh coefficients described in refs. 30–32,42,43 – correspond to different parametrizations of the MVB distribution (Table 1, "Methods"). We briefly describe these results below.

**Multiplicative and additive epistasis.** Suppose the fitness values $f_{\mathbf{x}} \in \mathbb{R}$ of each genotype $\mathbf{x} = (x_1, ..., x_L) \in \{0, 1\}^L$ are proportional to the corresponding probability $p_{\mathbf{x}}$ of a multivariate Bernoulli random variable $(X_1, ..., X_L)$, i.e., $f_{\mathbf{x}} = c \cdot p_{\mathbf{x}}$ for some $c > 0$. We prove that the (log-) multiplicative epistasis measures are equal to the *natural parameters* of the MVB. The natural parameters $\boldsymbol{\beta} = \{\beta_S\}_{S \subseteq \{1, ..., L\}}$ are another parameterization of the MVB that encodes conditional independence relations between the random variables $X_1, ..., X_L$; see refs. 29,48. We prove a similar result for the additive epistasis measure under the assumption that the fitness $f_{\mathbf{x}}$ is proportional to the log probability log $p_{\mathbf{x}}$. See Methods and Supplementary Note 3 for theorem statements and proofs.

Our theoretical results provide a connection between the multiplicative epistasis measure and interaction coefficients in a log-linear regression model. This is because for each subset $S$ of loci, the natural parameter $\beta_S$ corresponds to the interaction term for the subset $S$ in a log-linear regression model[29,47,48]. Such interaction terms are a standard approach for measuring epistasis in genetics, e.g., GWAS or QTL analyses for quantitative traits[2,5].

We also prove that the natural parameters $\boldsymbol{\beta}$ of the MVB are closely related to the two standard approaches for measuring pairwise SNP-SNP interactions in a case-control GWAS: logistic regression and conditional independence testing[49]. Specifically, we prove that the interaction term in a logistic regression is equal to a 3-way interaction term $\beta_{ijk}$ in an MVB, while the conditional independence test is equivalent to testing whether a 2-way interaction term $\beta_{ij}$ and a 3-way interaction term $\beta_{ijk}$ are both equal to zero. These interaction terms are equal to the corresponding log-multiplicative epistasis measures log $\epsilon^M$.

Thus, our results show that the additive and multiplicative epistasis measures are equivalent to computing interaction terms in regression models commonly used in genetics.

**Chimeric epistasis.** The connection between the epistasis formulae and the MVB distribution allows us to derive a mathematically rigorous definition of the chimeric epistasis formula. Specifically, suppose the fitness value $f_{\mathbf{x}}$ of each genotype $\mathbf{x} = (x_1, ..., x_L)$ is equal to a corresponding *moment* $E[X_1^{x_1} \cdots X_L^{x_L}]$ of the random variable $(X_1, ..., X_L)$. Then, we define the chimeric epistatic measure $\epsilon^C_{i_1 \cdots i_K}$ as the $K$-th order

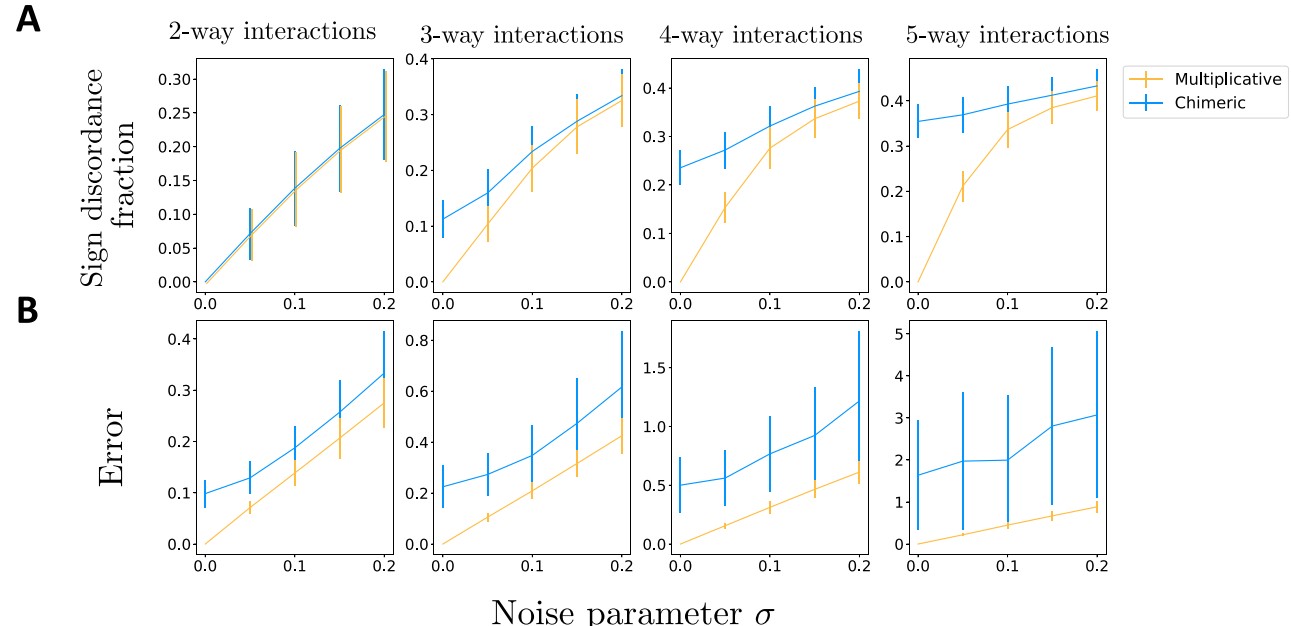

**Fig. 2 | Comparison of epistasis measures using simulated data from a multiplicative fitness model.** Fitness values **f** are simulated following a multiplicative fitness model with interaction parameters $\beta$, for different choices of the maximum interaction order $K$, and multiplicative Gaussian noise with standard deviation $\sigma$. **A** The fraction of $K$-way interactions where the sign of the log-multiplicative epistasis measure $\log \epsilon^M$ (orange) and the chimeric epistasis measure $\epsilon^C$ (blue) do not match the sign of the true interaction parameter $\beta$. **B** The average absolute difference ("error") $|\beta - \log \epsilon^M|$ and $|\beta - \epsilon^C|$ between the true interaction parameter $\beta$ and (orange) the log-multiplicative measure $\log \epsilon^M$ and (blue) the chimeric measure $\epsilon^C$, respectively. These quantities are computed for different values of the maximum interaction order $K$ and noise parameter $\sigma$ and are averaged across 200 simulated fitness values. Error bars indicate standard deviation across simulated instances.

*joint cumulant* $\kappa(X_{i_1}, \ldots, X_{i_K})$ of the random variables $X_{i_1}, \ldots, X_{i_K}$ (Table 1). Joint cumulants are a concept from probability theory that are used to quantify higher-order interactions between random variables[33]. See Methods for a formal definition.

We emphasize that prior literature on higher-order interactions do not provide a rigorous statistical interpretation of the chimeric epistasis measure. For example, Kuzmin et al.[24,26] does not explicitly state the connection between the joint cumulant and their three-way chimeric formula, while Tekin et al.[27] heuristically uses the joint cumulant formulae without specifying random variables or a probability distribution − thus obscuring any assumptions made by using joint cumulants to measure higher-order interactions.

Our explicit definition of the $K$-th order chimeric epistasis measure $\epsilon^C_{i_1 \cdots i_K}$ as the $K$-th order joint cumulant reveals two critical issues with the chimeric formula. First, the assumption that the fitness values $f$ are equivalent to the moments of an MVB random variable is not biologically reasonable for higher-order interactions between three or more loci. This is because the moments assumption implies that the fitness of a particular genotype depends on the probability of many other genotypes. For example, if we assume that the fitness values for $L = 4$ loci are moments of the MVB, then the fitness $f_{1100}$ of a double mutant is equal to the moment $E[X_1 X_2]$, which is equal to

$$E[X_1 X_2] = P(X_1 = 1, X_2 = 1) = p_{1100} + p_{1101} + p_{1110} + p_{1111}. \quad (6)$$

However, it is not clear why the fitness $f_{1100}$ of a *single* genotype, 1100, should equal the sum of the probabilities of *four different* genotypes, 1100, 1101, 1110, and 1111.

The second issue is that joint cumulants are not an appropriate measure of higher-order interactions between *binary* random variables. The differences between the joint cumulants $\kappa$ and natural parameters $\boldsymbol{\beta}$ have been previously investigated in the neuroscience literature, as both quantities have been used to quantify higher-order interactions in neuronal data. For example, Staude et al.[34] write that the

joint cumulants $\kappa$ and natural parameters $\boldsymbol{\beta}$ *"do not measure the same kind of dependence. While higher-order cumulant correlations [$\kappa$] indicate additive common components ... the [natural parameters $\beta$] directly change the probabilities of certain patterns multiplicatively"*. In particular, the natural parameters $\boldsymbol{\beta}$ measure *"to what extent the probability of certain binary patterns can be explained by the probabilities of its subpatterns"*[34]. Thus, for biallelic genotype data, the natural parameters $\boldsymbol{\beta}$ correspond exactly with the epistasis we aim to measure, i.e., how the fitness of a binary pattern can be explained by the fitness of its "subpatterns", while the joint cumulants $\kappa$ do not.

## Simulations using a multiplicative fitness model

We performed simulations to demonstrate the discrepancy between the multiplicative epistasis measure and the chimeric epistasis measure. Since both the multiplicative and chimeric measures use a multiplicative fitness model, we simulated fitness values **f** for $L = 10$ loci following a multiplicative fitness model with $K$-way interactions $\beta$ for different choices of interaction order $K$, and with multiplicative Gaussian noise with standard deviation $\sigma$ (Methods). We computed the $K$-way multiplicative measure $\epsilon^M_S$ and chimeric measure $\epsilon^C_S$ for each set $S \subseteq \{1, \ldots, L\}$ of loci of size $|S| = K$, and we compared these two measures to the true interaction measure $\beta_S$.

We first assessed whether the *sign* of the epistasis measures, i.e., $\text{sgn}(\log \epsilon^M_S)$ and $\text{sgn}(\epsilon^C_S)$, match the sign $\text{sgn}(\beta)$ of the corresponding interaction term $\beta_S$, since the sign of a measure indicates whether there is a positive or negative interaction between mutations in the loci $S$. We observed (Fig. 2A) that for pairwise interactions ($K = 2$), both the multiplicative measure $\epsilon^M$ and chimeric measure $\epsilon^C$ have the same sign as the true interaction measure $\beta$ for the same fraction of instances, which matches our theoretical result (Proposition 1, Methods). However, for higher-order interactions ($K > 2$), the chimeric measure $\epsilon^C$ has an incorrect sign more often than the multiplicative measure $\epsilon^M$ (Fig. 2A). In particular, for $K = 5$-way interactions, even with no noise (i.e., $\sigma = 0$), the chimeric measure has a different sign than the true

interaction parameter $\sigma$ for more than 30% of simulated instances. We also highlight that when there is no noise, i.e., $\sigma = 0$, the multiplicative measure always has the same sign as the true interaction parameter $\beta$, i.e., $\mathrm{sgn}(\log \epsilon^M) = \mathrm{sgn}(\beta)$, which agrees with Theorem 1.

We next compared how well the *magnitudes* of the multiplicative and chimeric epistasis measures agree with the magnitude of the true interaction parameters. We computed the average absolute difference ("error") $|\log \epsilon_S^M - \beta|$ and $|\epsilon_S^C - \beta|$ between the true interaction measure $\beta$ and the estimated multiplicative and chimeric epistasis measures, respectively, for all subsets $S$ of loci of size $|S| = K$. We found (Fig. 2B) that the multiplicative measure has a smaller error for all interaction orders $K$ and noise parameters $\sigma$. In particular, we observe that the multiplicative measure has a smaller error than the chimeric measure even for pairwise interactions ($K = 2$) – i.e., when both the multiplicative and chimeric measures have the same sign – and that the error of the chimeric measure $\epsilon^C$ increases with the interaction order $K$. The reason that the chimeric measure has much larger error than the multiplicative measure for pairwise interactions is that the chimeric measure $\epsilon_{ij}^C$ is approximately equal to the (log-)multiplicative measure *only when* $f_{ij} \approx 1$ and $f_i f_j \approx 1$, with the two measures being noticeably different otherwise (Supplementary Figs. 1 and 2 and Supplementary Note 1). We also emphasize that when there is no noise, i.e., $\sigma = 0$, the multiplicative measure has zero error, i.e., $\log \epsilon^M = \beta$, matching our theoretical results (Theorem 1, Methods). (Note that Theorem 1 does not apply when there is multiplicative Gaussian noise, i.e., $\sigma > 0$, as this noise will cause the fitness values to not follow a log-linear model.)

Thus, our results demonstrate that the multiplicative measure $\epsilon^M$ yields a more accurate measurement of pairwise and higher-order epistasis in a multiplicative fitness model compared to the chimeric measure $\epsilon^C$ which conflates additive and multiplicative factors.

## Simulations using the NK fitness model

We next compared the multiplicative and chimeric epistasis measures using the NK model, a classical model for simulating random fitness landscapes $\mathbf{f}$ with varying degrees of "ruggedness"[50]. The NK model has two parameters: the number $N$ of loci, which we call $L$ below; and $K$, a measure of the ruggedness of the fitness landscape $\mathbf{f}$, where the fitness landscape is smoothest at $K = 0$ and most rugged for $K = L - 1$. Each locus $\ell = 1, \dots, L$ interacts with $K$ random other loci, meaning that the fitness landscape contains at most $(K+1)$-way interactions. Since the NK model simulates fitness values under an additive model, we exponentiated the NK fitness values.

Each simulated fitness landscape $\mathbf{f}$ has an associated graph $G = (V, E)$ which describes a (simulated) *genetic interaction network*, where the vertices $V = \{1, \dots, L\}$ are the $L$ loci and the edges $E$ connect pairs of interacting loci[51]. For example, for $K = 0$, the graph $G$ has no

edges, indicating that there are no interactions between loci, while for $K = 1$ the graph $G$ has edges connecting loci with pairwise interactions. (For $K \geq 2$, one may also describe the interaction relationships with a *hypergraph* where hyperedges connect sets of interacting loci, e.g., ref. 51.)

We find that the chimeric measure falsely indicates the presence of higher-order interactions that are not present in the simulated fitness landscape $\mathbf{f}$ while the multiplicative measure does not. For example, when the fitness landscape $\mathbf{f}$ contains only *pairwise* interactions (i.e., $K = 1$), then the 3-way multiplicative epistasis measure $\epsilon_{ijk}^M = 0$ is equal to zero for all triples $(i, j, k)$ of loci. However, if the NK model graph $G$ contains a triangle $(i, j, k)$, then the 3-way chimeric measure $\epsilon_{ijk}^C \neq 0$ will be nonzero with high probability (Fig. 3A). Thus, the chimeric measure $\epsilon^C$ falsely indicates the presence of three-way interactions that do not exist in the simulated fitness landscape. (As this is sometimes a point of confusion: we note that triangles in a graph are sometimes referred to as higher-order structures[52]. However, as our simulation demonstrates, it is quite possible to have a triangle in a graph, i.e., three pairwise interactions, without having a genuine higher-order (3-way) interaction.) More generally, for any value $K > 0$ of the ruggedness parameter, the fitness landscape $\mathbf{f}$ only contains at most $(K+1)$-way interactions. The $(K+2)$-way multiplicative measure $\epsilon^M$ is always equal to zero, reflecting that there are no $(K+2)$-way interactions. However, we empirically observe that the $(K+2)$-way chimeric measure $\epsilon^C$ is often non-zero (Fig. 3B).

Thus, our analyses demonstrate how the chimeric measure $\epsilon^C$ will often erroneously identify higher-order interactions that are not present in the underlying fitness landscape.

## Three-way epistasis in budding yeast

We investigate the biological implications of using the chimeric epistasis measure instead of the multiplicative epistasis measure by reanalyzing two triple-gene-deletion studies in budding yeast by Kuzmin et al.[24,26]. These studies used triple-mutant synthetic genetic arrays (SGA)[53,54] to measure the fitness of single-, double-, and triple-mutant strains. The authors use a multiplicative fitness model since the SGA protocol models yeast colony sizes as a product of fitness, time, and experimental factors[23]. The Kuzmin et al. studies, refs. 24 and 26, measure fitness values for 195,666 and 256,861 gene triplets, respectively. They calculate the three-way chimeric epistasis measure $\epsilon_{ijk}^C$ and report 3196[24] and 2466[26] negative three-way epistatic interactions, respectively.

We calculated the multiplicative epistasis measure $\epsilon_{ijk}^M$ (formula (4)) and the chimeric epistasis measure $\epsilon_{ijk}^C$ (formula (5)) used by Kuzmin et al.[24,26] for the 189,340 gene triplets $(i, j, k)$ whose single-, double- and triple-mutant fitness values were available in the publicly

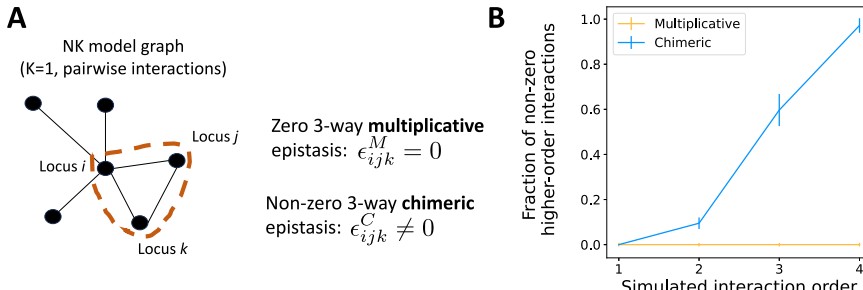

**Fig. 3 | Comparison of epistasis measures using simulated data from the NK fitness model. A** A fitness landscape $\mathbf{f}$ simulated following the NK fitness model with "ruggedness" parameter $K = 1$ contains only pairwise interactions. These interactions are represented with an interaction graph $G$. The 3-way multiplicative measure $\epsilon_{ijk}^M = 0$ equals zero for all loci triples $(i, j, k)$. However, if the triple $(i, j, k)$ forms a triangle in the graph $G$ (shown as a dashed red line), then the 3-way chimeric epistasis measure $\epsilon_{ijk}^C$ is non-zero and incorrectly indicates the presence of a higher-

order interaction. **B** The fraction of non-zero $(K+2)$-way interactions ("fraction of non-zero higher-order interactions") identified by the multiplicative measure $\epsilon^M$ (orange) and the chimeric measure $\epsilon^C$ (blue) across 100 fitness landscapes $\mathbf{f}$ simulated according to the NK fitness model with ruggedness parameter $K$, with error bars indicating standard deviation across simulated instances. The fitness landscape $\mathbf{f}$ contains at most $(K+1)$-way interactions, but the chimeric measure $\epsilon^C$ spuriously detects many non-zero $(K+2)$-way interactions.

**Table 2 | Comparison of signs of trigenic interactions in budding yeast calculated using the multiplicative epistasis measure and the chimeric epistasis measure on fitness data from Kuzmin et al.[26]**

| | | Chimeric measure $\epsilon_{ijk}^C$ | | |
|---|---|---|---|---|
| | | Positive | Ambiguous | Negative |
| Multiplicative measure $\epsilon_{ijk}^M$ | Positive | 1197 | 259 | 0 |
| | Ambiguous | 116 | 4291 | 91 |
| | Negative | 10 | 466 | 1527 |

Values not on the diagonal correspond to gene triplets having a different sign of epistasis using the multiplicative measure versus the chimeric measure (approximately 12% of triplets).

available data from refs. 24,26 and with a reported $p$-value of $p_{ijk} < 0.05$. Following Kuzmin et al.[24,26] we say a gene triplet $(i, j, k)$ has a *positive chimeric interaction* if $\epsilon_{ijk}^C > 0.08$; a *negative* chimeric interaction if $\epsilon_{ijk}^C < -0.08$; and an *ambiguous* chimeric interaction if $-0.08 < \epsilon_{ijk}^C < 0.08$. Accordingly, using the same quantile as the chimeric threshold of 0.08, we say that a gene triplet $(i, j, k)$ has a *positive (resp. ambiguous, negative) multiplicative interaction* if $\epsilon_{ijk}^M > 1.105$ (resp. $0.905 < \epsilon_{ijk}^M < 1.105$, $\epsilon_{ijk}^M < 0.905$). See Supplementary Note 4 and Supplementary Figs. 3 and 4 for specific details on data processing and reproducing the Kuzmin et al. results.

We observed considerable differences between the signs of the multiplicative epistatic measure versus the chimeric epistatic measure (Table 2). In particular, approximately 12% of gene triplets have a *different* interaction sign with the multiplicative measure compared to the chimeric measure. The difference between the two measures is especially pronounced for *negative* interactions, which are typically more functionally informative than positive interactions[23,24,26]. In particular, there were 476 gene triplets $(i, j, k)$ with a negative *multiplicative-only* interaction, or triplets with a negative multiplicative interaction but not a negative chimeric interaction (Fig. 4A). On the other hand, there were only 91 gene triplets with a negative *chimeric-only* interaction, or triplets with a negative chimeric interaction but not a negative multiplicative interaction (Fig. 4A); in fact, some of these 91 triplets even had *positive* multiplicative interaction (Fig. 4A). We also observe a qualitatively similar discrepancy between the two formula using the earlier fitness data from Kuzmin et al. (2018)[24]; on this data, we find that there were 746 gene triplets with a negative multiplicative-only interaction versus 177 triplets with a negative chimeric-only interaction (Supplementary Fig. 5). Our results were also qualitatively similar when we did not restrict to triplets with reported $p$-value $p_{ijk} < 0.05$ (Supplementary Fig. 6).

Negative trigenic interactions often contain genes whose proteins are partially redundant in their functions[25] and are enriched for other features that arise from biological models of functional redundancy, including shared expression patterns[55,56], shared protein-protein interactions[57], GO annotation, and amino acid divergence[56,57]. We observed (Fig. 4B) that gene triplets with negative multiplicative-only interactions − that is, gene triplets not identified by the chimeric formula used in Kuzmin et al. (2020)[26] − are significantly enriched for co-expression ($P = 0.017$, hypergeometric test), shared protein-protein interactions ($P < 1.5 \times 10^{-4}$, hypergeometric test), and similar GO annotations ($P < 2.1 \times 10^{-5}$, hypergeometric test). In contrast, gene triplets with a negative chimeric-only interaction are not significantly enriched for any of these features (Fig. 4B). In this way using the multiplicative measure extends the network of functionally redundant genes by almost 25% compared to the chimeric measure. We obtain a similar result when analyzing the fitness data from the earlier Kuzmin et al. (2018) study[24] (Supplementary Fig. 5) and also when we do not remove gene triplets with large reported $p$-values $p_{ijk}$ as computed by Kuzmin et al.[24,26] (Supplementary Fig. 6). These results demonstrate that using the appropriate three-way multiplicative formula for a

multiplicative fitness model leads to more biologically meaningful higher-order genetic interactions compared to using the chimeric epistasis formula that mixes additive and multiplicative scales in an statistically unsound manner.

In particular, trigenic interactions also reveal the functional redundancy of *paralogs*, or pairs of duplicated genes with overlapping functions, since two functionally similar genes tend to have a negative trigenic interaction with a third gene more often compared to gene pairs with non-overlapping functions[26]. Thus, we evaluated whether the gene triplets with negative multiplicative-only interactions involve functionally redundant gene pairs. We quantified the functional redundancy between two genes by calculating the number of negative trigenic interactions to which both genes belong, where we restricted our calculation to gene pairs involved in at least two negative multiplicative interactions. We found that many pairs of genes had additional multiplicative-only interactions (Fig. 4C). Thus the multiplicative measure identified additional functional redundancies not found using the chimeric measure. As additional validation, we note that Kuzmin et al.[26] quantify functional redundancy between two genes using a related quantity that they call the *trigenic interaction fraction* (see Supplementary Note 4 for more details). We observed that for most gene pairs, the trigenic interaction fraction is larger when computed using the multiplicative formula versus using the chimeric formula (Supplementary Fig. 7). This observation further supports the conclusion that the multiplicative formula uncovers additional functional redundancies between these paralogs that was not detected by the chimeric measure.

We expect paralogs with large increases in the number of multiplicative-only interactions to be functionally redundant. Of the 130 paralogs we analyzed, there are fifteen paralogs with at least 10 negative multiplicative-only interactions (highlighted in Fig. 4C). The three paralogs with the largest number of negative multiplicative-only interactions were *RPS24A-RPS25B*, *MSN2-MSN4*, and *ARE1-ARE2*. For these three paralogs, the multiplicative formula quadrupled the number of total trigenic interactions compared to the number of such interactions reported by Kuzmin et al.[26] using the chimeric formula. These three paralogs also appear to have redundant functions according to other patterns of sequence evolution: all three have highly correlated position-specific evolutionary rates (Table S12 in ref. 26) and two of them (*RPS24A-RPS25B* and *ARE1-ARE2*) have low sequence divergence rates (Supplementary Fig. 8). Moreover, negative genetic interactions have been previously documented for *MSN2-MSN4*[58], *ARE1-ARE2*[59], and *RPS25A-RPS25B*[21,60].

The paralogs with many multiplicative-only interactions are also enriched for shared PPIs or GO annotations with the genes they interact with (Fig. 4D). In particular, the paralogs *NUP53-ASM4*, which are components of the large nuclear pore complex[61], had 36 additional negative multiplicative-only interactions. These epistatic interactions are highly enriched for shared PPIs and GO annotations (Fig. 4D) and also involve members of the same protein complexes (Fig. 4E). One of the 36 additional genes that interact with *NUP53-ASM4* is *NUP145*, which also forms part of the nuclear pore[62]. Interestingly, while the gene triplet *NUP53-ASM4-NUP145* has a negative multiplicative interaction ($\epsilon^M = 0.684 < 1$), the same gene triplet was reported to have a *positive* chimeric interaction ($\epsilon^C = 0.25 > 0$; Kuzmin et al.[26]). Another example of one of the 36 additional interactions is *SAC3*, which encodes a nuclear pore-associated protein that functions in mRNA transport[63]. The gene triplet *NUP53-ASM4-SAC3* has a very negative multiplicative interaction ($\epsilon^M = 0.046 << 1$), but in the original study[26] was reported to have a slightly positive chimeric interaction ($\epsilon^C = 0.014 > 0$). Moreover, both *NUP145* and *SAC3* share at least one protein-protein interaction and GO category with *NUP53* and *ASM4*. These findings provide additional support to the hypothesis by Kuzmin et al.[26] that *NUP53* and *ASM4* have overlapping functions.

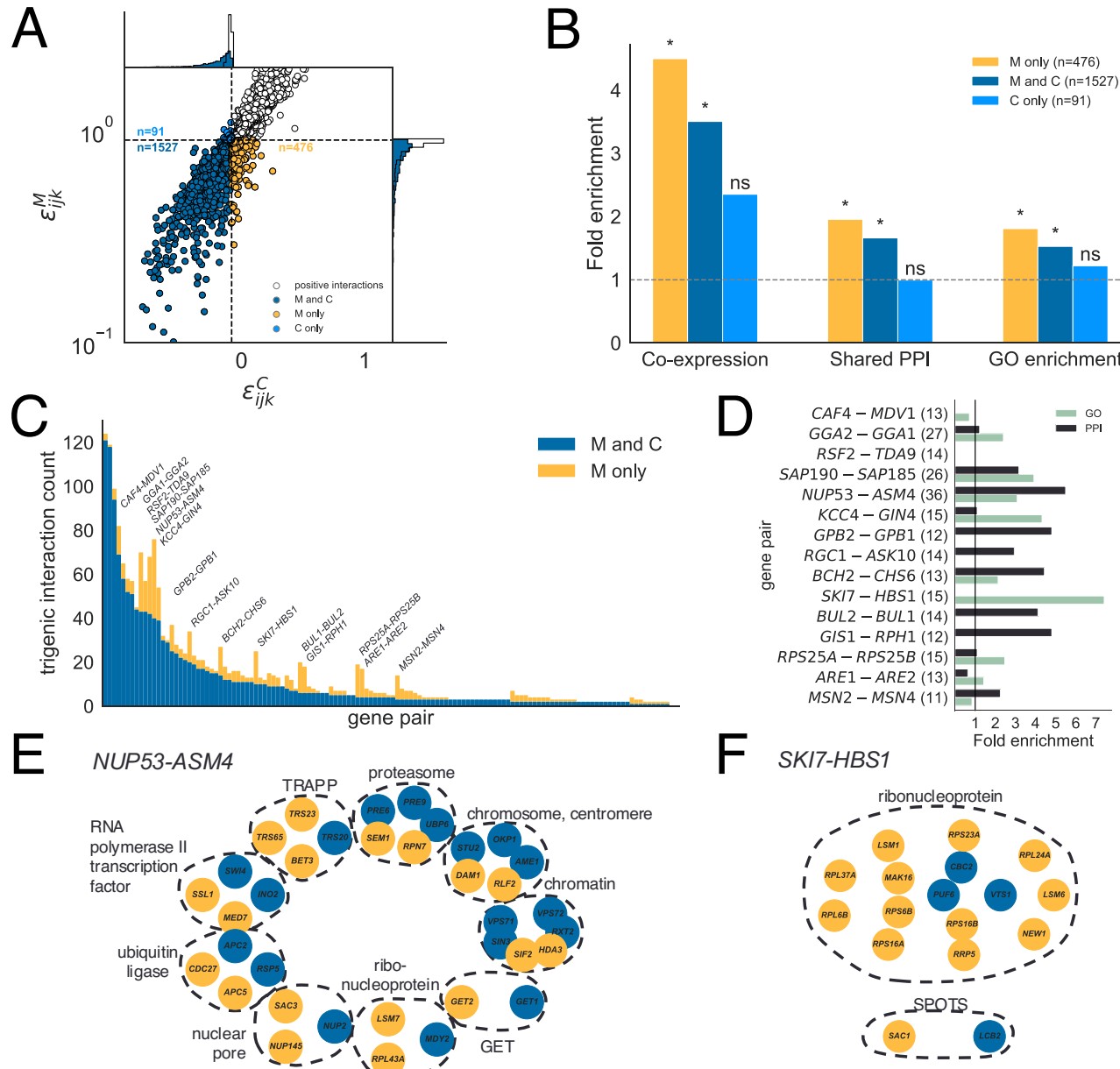

**Fig. 4 | Negative trigenic interactions in budding yeast calculated using the multiplicative and chimeric epistasis measures. A** Chimeric epistasis measure $\epsilon_{ijk}^C$ versus the multiplicative epistasis measure $\epsilon_{ijk}^M$ for gene triplets $(i,j,k)$ in Kuzmin et al.[26]. We highlight trigenic interactions that are negative only by the multiplicative measure ("M only"), only by the chimeric measure ("C only"), or by both measures ("M and C"). **B** Fold enrichment for co-expression patterns, shared protein-protein interactions (PPI), and shared GO annotations for negative trigenic interactions. Asterisk (*) denotes statistical significance ($P < 0.018$, hypergeometric

test, one-sided), while 'ns' indicates not significant ($P > 0.05$). **C** Number of negative trigenic interactions $(i, j, k)$ for every pair $(i, j)$ of genes with at least five negative trigenic interactions. **D** Fold enrichment for GO annotations and protein-protein interactions (PPI) for negative "M only" trigenic interactions that involve the gene pairs highlighted in (**C**). The numbers in parentheses are the number of "M only" interactions. **E/F** Genes that have a negative trigenic interaction with either *NUP53-ASM4* (**E**) or with *SKI7-HBS1* (**F**), organized into protein complexes and colored by whether the trigenic interaction is "M only" (gold) or "M and C" (blue).

Two other noteworthy paralogs are *SKI7* and *HBS1*; both genes recognize ribosomes stalled during translation and also initiate mRNA degradation. While some studies report that these paralogs have evolved distinct functions[64,65], other studies show that they retain some overlapping functions[66–68] and may bind to similar sites on the ribosome[68]. Kuzmin et al.[26] previously reported relatively few (13) trigenic interactions involving both *SKI7* and *HBS1* as corroboratory evidence for the functional divergence of these paralogs. However, by using the multiplicative epistasis formula, we find 15 additional trigenic interactions involving *SKI7* and *HBS1*. These 15 multiplicative-only interactions are highly enriched for shared GO terms (Fig. 4D). Moreover, 12 of the 15 multiplicative-only interactions involve functionally

similar genes that are all members of the ribonucleoprotein complex (Fig. 4F). Thus, the multiplicative epistasis measure finds evidence for additional functional redundancy between *SKI7* and *HBS1* that went undetected by the chimeric epistasis measure used in Kuzmin et al.[26].

In addition to the negative interactions just described, we also highlight an example of a biologically relevant *positive* trigenic interaction that is missed by the chimeric epistasis measure but detected by the multiplicative measure. The gene triplet *CIK1-VIK1-SUP35-td*, which consists of two paralogs, *CIK1* and *VIK1*, involved in mitosis[69], and the essential gene *SUP35-td*[70], has an ambiguous, negative chimeric interaction ($\epsilon_{ijk}^C = -0.03$) but has a very large, positive multiplicative interaction ($\epsilon_{ijk}^M = 75.337$). Examining the fitness values (Supplementary

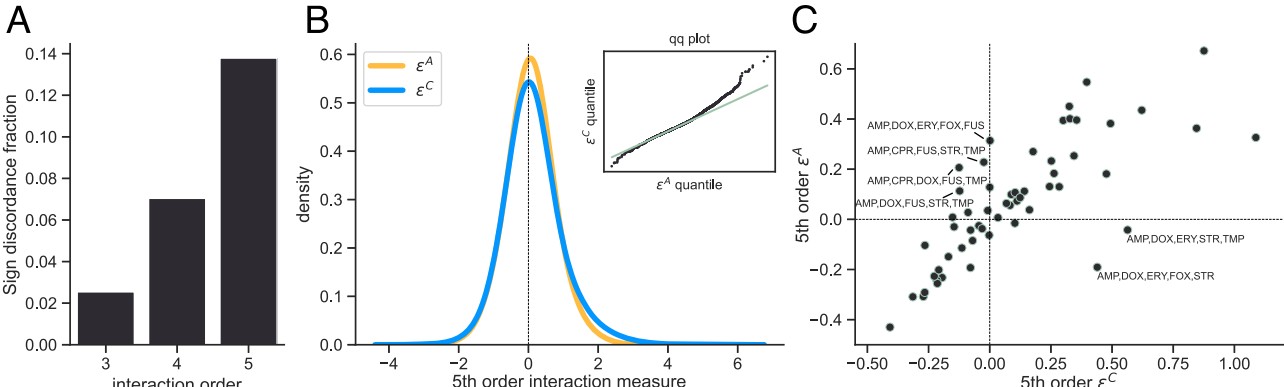

**Fig. 5 | Higher-order interactions between antibiotics in E. coli using drug response data from Tekin et al.[40]. A** Proportion of E. coli cultures where the sign (positive vs. negative) of the chimeric and additive measures disagree. The sign discordance fraction, or proportion of interactions where the sign of the two measures disagree, increases with the interaction order, consistent with the simulations shown in Fig. 1. **B** Distributions and Q-Q plots (insets) for the additive (orange) and chimeric (blue) measures for 5th-order interactions. **C** Scatter plot of median relative growth rates for each 5-way combination of antibiotics across concentration levels and replicates. Dashed horizontal and vertical lines indicate zero additive and chimeric epistasis measures, respectively.

Fig. 9) shows that the fitness of the *CIK1-VIK1-SUP35-td* triple mutant is more than 100 times larger than the fitness of the *CIK1-SUP35-td* double mutant. Moreover, positive interactions have been previously documented between pairs of these genes: *VIK1* deletion mutants suppress several phenotypes of *CIK1* deletion mutants, including a mitotic delay phenotype and a temperature-dependent fitness defect[69]; and a phenotypic suppression interaction exists between *CIK1* and *SUP35*, where deletion of *CIK1* reduces the ability of *SUP35-td* to form prions[69]. These previously identified positive pairwise interactions, together with the large triple-mutant fitness value, demonstrate that the gene triplet *CIK1-VIK1-SUP35-td* is more likely to have a positive interaction as indicated by the multiplicative measure, rather than a neutral interaction as indicated by the chimeric measure.

Overall, our results demonstrate not only the degree to which the multiplicative and chimeric formula may lead to distinct interpretations of fitness data, but also that genetic interactions measured using the multiplicative formula appear to be more consistent with other biological features compared to interactions measured using the chimeric formula.

### Higher-order interactions in drug responses

We next reanalyzed a drug response dataset[40] in which three-way, four-way, and five-way interactions between drug combinations were quantified using the chimeric formula. For these data, the authors exposed *Escherichia coli* cultures to between one and five antibiotics (out of eight total) at one of three different concentrations. They measured fitness as the difference in exponential growth rates between the culture exposed to antibiotics and a negative control with no antibiotics. The authors then used the chimeric epistasis measure $\epsilon^C$ to identify third-, fourth-, and fifth-order interactions between different combinations of antibiotics. We compared their results with the additive epistasis measure $\epsilon^A$. We used the additive measure $\epsilon^A$ because, under the standard assumption that antibiotic exposure multiplicatively affects the survival probability of individual cells[46], then antibiotic exposure will have an additive effect on the exponential growth rates of the population of cells[11,71].

The signs of the chimeric interaction measure $\epsilon^C$ and the additive interaction measure $\epsilon^A$ disagree for three-way, four-way, and five-way interactions, with the discrepancy between the two measures increasing with the interaction order (Fig. 5A), which is consistent with our earlier simulations (Fig. 1C). The discrepancy is largest for fifth-order interactions, with approximately 14% of fifth-order interactions having a different sign using the additive measure versus the chimeric measure (Fig. 5A).

The discrepancy between the additive and chimeric measures may lead to different conclusions on the type of interactions between antibiotics, i.e., whether a given combination of antibiotics is *synergistic* (more effective at killing bacteria when taken together versus taken individually, i.e., a *negative* interaction) or *antagonistic* (less effective together versus individually, i.e., a *positive* interaction). For fifth-order interactions, the chimeric measure $\epsilon^C$ was more positively skewed than the additive measure $\epsilon^A$ (Fig. 5B), with a Pearson skewness coefficient of 0.87 for the chimeric measure versus 0.17 for the additive measure. Thus, the chimeric measure is significantly more likely to identify antagonistic interactions than the additive measure ($P < 7 \times 10^{-43}$, paired t-test).

We then examined specific five-way combinations of antibiotics with different interaction signs following the procedure of ref. 27 and ref. 45. For each five-way combination of antibiotics we first calculated the median relative growth rate of E. coli across replicates and concentrations, and then used these median relative growth values to compute both the additive and chimeric measures (Fig. 5C). The interaction between the antibiotic combination Ampicillin (AMP), Doxycycline hyclate (DOX), Erythromycin (ERY), Streptomycin (STR), Trimethoprim (TMP) is highly antagonistic using the chimeric measure (i.e., $\epsilon^C = 0.56 > 0$) but synergistic using the additive measure (i.e., $\epsilon^A = -0.04 < 0$). A similar pattern also holds for the antibiotic combination consisting of AMP, DOX, ERY, STR, and Cefoxitin sodium salt (FOX). We emphasize that because we use the same fitness values as reported in ref. 40, the differences between the additive and chimeric measures arise *solely* from the use of the additive versus chimeric measures as opposed to variability arising from biological or technical replicates.

### Epistasis between protein mutations

We further demonstrate the difference between the multiplicative and chimeric epistasis measures using experimental fitness data of eleven different proteins[10,38,72–79]. These fitness values were measured using deep mutational scanning (DMS), a recent class of technologies which use high-throughput sequencing to measure the fitness of many variants of a protein. The published analyses of each of these datasets quantified epistasis using either an additive or multiplicative epistasis measure, depending on the fitness measurement being made. We reanalyzed each dataset using the chimeric measure to demonstrate the differences between the chimeric measure and the additive/multiplicative measures.

We found that the multiplicative and chimeric measures have substantial disagreement in quantifying higher-order epistasis within

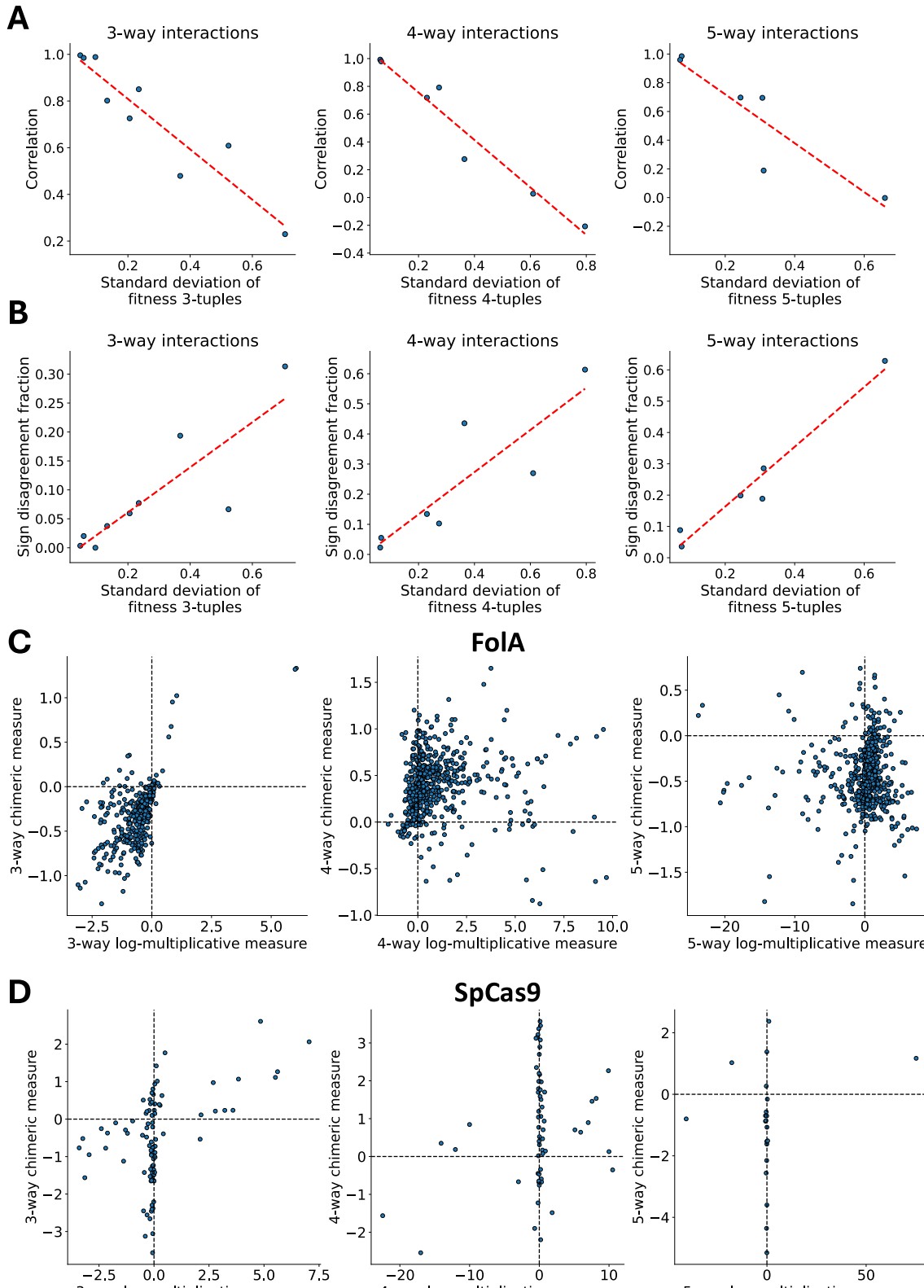

**Fig. 6 | Epistasis between protein mutations in eleven different proteins.**
**A**, **B** Standard deviation of fitness values across all (left) three-, (middle) four-, and (right) five-way tuples of mutations versus the average (**A**) correlation and (**B**) sign disagreement fraction of the log-multiplicative measure log $\epsilon^M$ versus the chimeric measure $\epsilon^C$. Line of best fit is shown as a dashed red line. **C**, **D** Log-multiplicative measure log $\epsilon^M$ versus chimeric measure $\epsilon^C$ for the (**C**) FolA[72] and (**D**) *Streptococcus pyogenes* Cas (SpCas9) nuclease[38] proteins. Dashed vertical and horizontal lines indicate zero log-multiplicative and chimeric epistasis measures, respectively.

several of the proteins. Furthermore, we observe that both the correlation between the measures and the sign disagreement fraction vary as a function of the standard deviation $s$ of the $2^K$ fitness values $\{f_{0\cdots0}, \ldots, f_{1\cdots1}\}$ across all $K$-tuples of mutation. Specifically, the correlation between the chimeric measure and the multiplicative measure *decreases* as a function of the fitness standard deviation $s$ (Fig. 6A), while the sign disagreement function *increases* as a function of the fitness standard deviation $s$. These results show the large difference between the multiplicative and chimeric measures for proteins with a large standard deviation in fitness values.

The FolA metabolic protein from E. coli has the second largest standard deviation $s$ in fitness of the eleven proteins that we analyzed. This data from ref. 72 includes the fitness of approximately 260, 000 mutations at nine single-nucleotide loci. The three-way multiplicative and chimeric measures for the FolA protein have correlation 0.6086, while the four- and five-way measures have correlation < 0.05 – i.e., the two measures are almost uncorrelated for four- and five-way interactions (Fig. 6C). There is also substantial sign disagreement between the multiplicative and chimeric measures, with over 60% sign disagreement for five-way interactions.

Another protein with large fitness standard deviation $s$ is the *Streptococcus pyogenes* Cas9 (SpCas9) nuclease, a widely used protein for genome editing across biology. The fitness landscape of SpCas9 was profiled in ref. 38, where fitness was measured by the editing efficiency of the SpCas9 protein. For the SpCas9 protein, the sign disagreement between the two epistasis measures is over 20% for three-, four-, and five-way interactions (Fig. 6D). The large sign disagreement between the two epistasis measures is likely because for many protein variants, the log-multiplicative measure $\log \epsilon^M$ is close to 0 while the chimeric measure $\epsilon^C$ varies substantially between $-4$ and 2.

Overall, our results demonstrate the extent to which one may infer substantially different higher-order epistasis between protein mutations – including different *signs* of epistasis – if the chimeric measure is used in place of the additive/multiplicative measures.

## Discussion

Higher-order interactions between genetic variants, drugs, and other perturbations play a large role in shaping the fitness landscape of an organism[1,80]. Yet despite the importance of these interactions, there are multiple different – and sometimes inconsistent – formulae used in the literature for measuring higher-order interactions, most notably for measuring higher-order epistasis between mutations. In particular, many researchers use a *chimeric* formula that quantifies epistasis as an additive deviation from a multiplicative null model and is thus a "chimera" of additive and multiplicative measurement scales.

In this work, we show that there is considerable disagreement between the chimeric epistasis measure and the additive and multiplicative measures. For higher-order interactions, the chimeric measure often has a different *sign* compared to the multiplicative measure (Fig. 1C). We demonstrate that this inconsistency is not purely a mathematical curiosity but also leads to markedly different biological conclusions in yeast genetics[24,26] (Fig. 4), antibiotic resistance[40,45] (Fig. 5), and protein epistasis (Fig. 6), raising potential questions about some reported higher-order epistatic interactions in the literature. Furthermore, we show that the different epistasis measures are equal to different parametrizations of the multivariate Bernoulli distribution (MVB)[29] (Table 1) and demonstrate that the chimeric epistasis measure is less statistically sound than the additive and multiplicative measures. Our connection between epistasis measures and parameters of the multivariate Bernoulli measure is general and unifies many different epistasis measures: the additive, multiplicative, and chimeric measures; and the Walsh coefficients[30–32,42,43]. Overall, our results demonstrate that the more appropriate multiplicative and additive formulae for higher-order epistasis yield more mathematically sound and

biologically meaningful results compared to the chimeric formula which improperly conflates measurement scales.

Historically, most work in epistasis has focused on pairwise interactions, where the chimeric and multiplicative measures agree on the interaction sign, and thus the differences between these two measures are not widely reported. However, even in the pairwise setting, the two measures have different magnitudes, which may still affect biological findings. For example, Costanzo et al.[21] recently built a large-scale pairwise interaction network for yeast using the chimeric epistasis measure, where they included an edge between two genes if the absolute value of the chimeric measure was greater than a certain threshold. From our results with the trigenic yeast network (Section 2.6), it is possible that the edges in the network would change if one used the more appropriate multiplicative measure instead, which may lead to the inference of different genetic interactions and thus the functional relationships and regulatory mechanisms identified by Costanzo et al.[21].

There are several future directions for our work. First, it would be useful to further investigate the relationship between the MVB and regression-based approaches for quantifying epistasis in GWAS[81,82]. For example, regression-based approaches often do not require that one has measured the fitness of all $2^L$ genotypes, which may make the estimation of the interaction parameters of the MVB more challenging. Moreover, these regression approaches may sometimes produce biased estimates of epistasis[83], and we imagine that the MVB would provide a useful statistical framework for characterizing such statistical biases. A second direction is to incorporate uncertainty of fitness measurements in the MVB, e.g., by using a Bayesian framework. Thirdly, one could generalize our theoretical results by relaxing the assumption that fitness is proportional to genotype probability, e.g., by incorporating genetic drift or other more evolutionarily realistic factors. Fourth, our statistical framework could be extended to model how higher-order interactions contribute to evolutionary trajectories in a fitness landscape[84]. Fifth, it would be quite interesting to investigate the connections between the MVB and the circuit formulae used to quantify the shape of a fitness landscape[19,42,43,85]. Finally, we note that our MVB framework provides an approach for doing formal model comparisons. Thus, an interesting and important future direction would be to derive a statistical test using the MVB to test whether an additive or multiplicative fitness model better fits data from different experimental technologies.

Ultimately, future studies on interactions in genetics, drug response, protein fitness landscapes, and other domains should take care to use the mathematically appropriate additive or multiplicative formula for measuring higher-order interactions, and not fall victim to the chimera.

## Methods
### Pairwise epistasis
We start with the simplest setting where the genotype consists of two loci, each with two alleles labeled 0 and 1, i.e., a haploid genome with a single biallelic mutation. Thus, there are four possible genotypes – the wild-type 00, the single mutants 01 and 10, and the double mutant 11 – with corresponding fitness values $f_{00}, f_{01}, f_{10}$, and $f_{11}$ (Fig. 1). There are two standard null models that relate genotype to fitness: the additive model and the multiplicative model.

**Additive fitness model.** In the first model, mutations are assumed to have an *additive* effect on fitness[2,19], e.g., in drug resistance[42,85] and protein binding[30,32]. The effect of a mutation is quantified by the difference in fitness when one locus is mutated; for example, $f_{11} - f_{10}$ measures the effect of a mutation in the second locus, where the genetic background is a mutation in the first locus. Interaction between mutations in the two loci, i.e., *pairwise epistasis*, is measured by the difference in the effect of a mutation in one locus across the two

possible genetic backgrounds (Supplementary Fig. 10A). The pairwise interaction measure $\epsilon^A$ is given by

$$\epsilon^A = (f_{11} - f_{10}) - (f_{01} - f_{00}). \tag{7}$$

Note that the definition (7) of the pairwise epistasis measure is invariant to the choice of which locus is mutated, i.e., $\epsilon^A = (f_{11} - f_{10}) - (f_{01} - f_{00}) = (f_{11} - f_{01}) - (f_{10} - f_{00})$. In practice, the fitness values are often normalized so that $f_{00} = 0$, i.e., the fitness $f_{00}$ of the wild-type is zero, resulting in the following commonly-used equation for pairwise epistasis under an additive fitness model:

$$\epsilon^A = f_{11} - (f_{01} + f_{10}). \tag{8}$$

Equivalently, the pairwise epistasis measure $\epsilon^A$ is the difference between the observed double-mutant fitness $f_{11}$ and the expected double-mutant fitness $f_{01} + f_{10}$ under a null model with no epistasis. As ref. 2 notes, this definition of pairwise epistasis is similar to Fisher's original definition of epistasis[41].

The *sign* $\text{sgn}(\epsilon^A)$ of the pairwise epistasis measure $\epsilon^A$ determines the type of epistatic interaction. If $\epsilon^A = 0$, then there is no interaction between the two loci and so the fitness $f_{11}$ of a double mutant is completely determined by the sum $f_{11} = f_{01} + f_{10}$ of the single mutant fitnesses $f_{01}, f_{10}$. If $\epsilon^A > 0$ then there is a *positive* interaction between the two loci, in the sense that the fitness $f_{11}$ of the double mutant is *larger* than the fitness if there was no pairwise interaction. Similarly, if $\epsilon^A < 0$ then there is a *negative* interaction between the two loci, in the sense that the fitness $f_{11}$ of the double mutant is *smaller* than the fitness if there was no pairwise interaction.

The pairwise epistasis measure $\epsilon^A$ is equivalent to two other notions of epistasis used in the genetics literature. First, the pairwise epistasis measure $\epsilon^A$ is equal to the pairwise interaction term in the standard linear *regression* framework for quantifying epistasis[12]. Specifically, if the fitness values $f_{00}, f_{01}, f_{10}, f_{11}$ follow a linear model of the form

$$f_{x_1 x_2} = \beta_0 + \beta_1 x_1 + \beta_2 x_2 + \beta_{12} x_1 x_2, \tag{9}$$

then the coefficient $\beta_{12}$ of the interaction term $x_1 x_2$ is equal to the pairwise epistasis measure $\epsilon^A$ in (7). Second, the epistasis measure $\epsilon^A$ is equal (up to a constant factor) to the 2nd-order Walsh coefficient that is often used to measure "background-averaged" epistasis[30–32,42,43].

**Multiplicative fitness model.** In this model, mutations are assumed to have a *multiplicative* effect on fitness, e.g., modeling cellular growth rates[2,14,21,23,24]. The multiplicative pairwise epistasis measure (Supplementary Fig. 10B) is given by

$$\epsilon^M = \frac{f_{11}}{f_{10}} \Big/ \frac{f_{01}}{f_{00}} = \frac{f_{11} f_{00}}{f_{10} f_{01}}. \tag{10}$$

As in the additive model, in practice the fitness values are typically normalized such that $f_{00} = 1$, resulting in the following equation for pairwise epistasis:

$$\epsilon^M = \frac{f_{11}}{f_{01} f_{10}}. \tag{11}$$

That is, the pairwise epistasis measure $\epsilon^M$ is the *ratio* between the double-mutant fitness $f_{11}$ and the product $f_{01} f_{10}$ of the single-mutant fitness values.

The multiplicative fitness model is closely related to the additive fitness model: if fitnesses $f$ are multiplicative, then the *log*-fitnesses $\log f$ are additive. Thus, the sign of the interaction is determined by the difference between the epistasis measure $\epsilon^M$ and 1, or equivalently the sign $\text{sgn}(\log \epsilon^M)$ of the $\log \epsilon^M$ of the epistasis measure $\epsilon^M$ (Fig. 1A). If

$\epsilon^M > 1$, i.e., $\log \epsilon^M > 0$, there is a *positive* interaction between the two loci; if $\epsilon^M = 1$, i.e., $\log \epsilon^M = 0$, then there is no interaction between the two loci; and if $\epsilon^M < 1$, i.e., $\log \epsilon^M < 0$, then there is a *negative* interaction between the two loci.

The multiplicative pairwise epistasis measure is closely related to the pairwise interaction term in the standard *log-linear* regression framework for epistasis[12]. Specifically, if the fitness values $f_{00}, f_{01}, f_{10}, f_{11}$ follow a log-linear regression model of the form

$$\log f_{x_1 x_2} = \beta_0 + \beta_1 x_1 + \beta_2 x_2 + \beta_{12} x_1 x_2, \tag{12}$$

then $\beta_{12} = \epsilon^M$.

**Chimeric formula.** Many studies in genetics use a multiplicative fitness model but do not measure pairwise epistasis with the multiplicative epistasis measure $\epsilon^M$. Instead, these papers use a multiplicative null model but measure deviations with an *additive* scale, yielding the following epistasis measurement:

$$\epsilon^C = f_{11} - f_{01} f_{10}. \tag{13}$$

We call $\epsilon_{ij}^C$ a *"chimeric"* measure as it measures deviations from a multiplicative null model on an additive scale, and is thus a *chimera* of both the multiplicative and additive measurement scales. The chimeric measure has been widely used in the genetics literature (e.g., refs. 2,14,23,86) and in the drug interaction literature (e.g., refs. 27,40,44–48). In these applications, similar to the additive measure, the sign of an interaction between two loci is determined by the $\text{sgn}(\epsilon^C)$ of the chimeric measure $\epsilon^C$: $\epsilon^C > 0$ corresponds to a positive interaction while $\epsilon^C < 0$ corresponds to a negative interaction.

Although it is often described in terms of a multiplicative fitness model, the chimeric epistasis measure $\epsilon^C$ is *not* equal to the multiplicative measure $\epsilon^M$. The chimeric epistasis measure $\epsilon^C$ in Equation (13) is similar to equation (11), but the deviation between the observed double-mutant fitness $f_{11}$ and the expected fitness $f_{01} f_{10}$ under a multiplicative null model is computed using subtraction instead of division. Equivalently, the (log-)multiplicative epistasis measure $\log \epsilon^M = \log f_{11} - \log f_{01} f_{10}$ computes the difference between the observed and expected logarithm of the fitness of the double mutant, while the chimeric epistasis measure $\epsilon^C = f_{11} - f_{01} f_{10}$ computes the difference directly (Fig. 1A). In this way, the chimeric epistasis measure may overstate or understate the strength of a pairwise interaction in a multiplicative fitness model (Fig. 1A); see Supplementary Note 2 for a numerical example highlighting this issue with the chimeric measure.

We note that the differences between the multiplicative epistasis measure $\epsilon_{ij}^M$ and chimeric epistasis measure $\epsilon_{ij}^C$ do not appear to be widely appreciated in either the applied or theoretical literature. Almost every study that uses the chimeric epistasis measure $\epsilon_{ij}^C$ does not consider the multiplicative measure $\epsilon_{ij}^M$. On the other hand, while many in the statistics literature draw a distinction between additive and multiplicative interaction effects (e.g., refs. 12,13), none of these papers discuss the chimeric interaction measure $\epsilon_{ij}^C$ that is frequently used in the genetics and drug interaction literature. An exception is Gao, Granka, and Feldman[87] who refer to the multiplicative formula (2) as a *"rescaling of [the chimeric] formula"*, but we take the stronger view that "rescaling" obscures consequential implications of the two formula.

Nevertheless, we show that the chimeric measure $\epsilon^C$ measures the same *sign* of interaction as the multiplicative measure $\epsilon^M$.

**Proposition 1.** Let $f_{01}, f_{10}, f_{11} \in \mathbb{R}$ be real numbers. Let $\epsilon^M = \frac{f_{11}}{f_{01} f_{10}}$ and $\epsilon^C = f_{11} - f_{01} f_{10}$. Then $\text{sgn}(\epsilon^C) = \text{sgn}(\log \epsilon^M)$.

**Proof.** $f_{11} - f_{01} f_{10} > 0 \iff \frac{f_{11}}{f_{01} f_{10}} > 1$.

**Choosing an appropriate null model.** The appropriate choice of null fitness model depends on the quantitative trait being used to approximate fitness. Cellular growth rate (i.e., the Malthusian parameter[88], which is often used as a measure of the fitness of microbial populations) is typically described with an additive fitness model; this is because in the population genetics literature, it is often assumed that mutations that independently effect survival and reproduction probabilities combine multiplicatively within individual cells[89,90], and so these mutations will combine additively in their effect on the growth rate of a continuously-growing clonal population[11,71]. However, multiplicative models are sometimes still used inappropriately in this setting[9]. On the other hand, the relative frequency of a microbial (or protein) population is typically modeled multiplicatively, as the growth rate of a population is proportional to the logarithm of its relative frequency. In general, Wagner[13,46] suggests that one should use the model that preserves the scale on which single-mutant fitness effects were measured (i.e., additive or fold differences from wild type).

## Higher-order epistasis

We next generalize our discussion to genotypes with $L \geq 2$ loci, where we demonstrate that the differences between the multiplicative measure and the chimeric measure become even more pronounced when analyzing *higher-order epistasis*, or interactions between three or more loci.

There are $2^L$ genotypes $x_1 \cdots x_L$, where $x_\ell \in \{0, 1\}$ indicates a mutation in locus $\ell$, with each genotype $x_1 \cdots x_L$ having a corresponding fitness value $f_{x_1 \cdots x_L}$, e.g., $f_{010}$ is the fitness of genotype 010 with a mutation in the second locus and no mutations in the first and third loci. However, because writing out the $2^L$ genotypes is infeasible for large $L$, we use the following notational shorthand. We use $f_i$ to refer to the fitness of the genotype with a single mutation in locus $i$, $f_{ij}$ to refer to the fitness of the genotype with mutations in loci $i, j$, and so on. For example, for $L = 3$ loci, $f_2$ corresponds to $f_{010}$ while $f_{12}$ corresponds to $f_{110}$. Without loss of generality, we assume the wild-type fitness $f_\varnothing$ is equal to 0 for the additive fitness model and equal to 1 for the multiplicative and chimeric fitness models. We also define $\epsilon_{ij}^A, \epsilon_{ij}^M, \epsilon_{ij}^C$ as the additive, multiplicative, and chimeric pairwise epistasis measure, respectively, between the $i$-th locus and the $j$-th locus, i.e., $\epsilon_{ij}^A = f_{ij} - f_i - f_j$, $\epsilon_{ij}^M = \frac{f_{ij}}{f_i f_j}$ and $\epsilon_{ij}^C = f_{ij} - f_i f_j$. For example, for $L = 3$ loci, $\epsilon_{12}^M$ corresponds to $\frac{f_{110}}{f_{100} f_{010}}$.

**Additive fitness model.** We start by quantifying three-way epistasis in the additive fitness model. When there is no pairwise epistasis, the fitness $f_{ijk}$ of a triple mutant is equal to $f_i + f_j + f_k$, i.e., the fitness from of each of the single mutants. When there is pairwise epistasis, then the triple mutant fitness $f_{ijk}$ also includes pairwise interaction measures, i.e.,

$$f_i + f_j + f_k + \epsilon_{ij}^A + \epsilon_{ik}^A + \epsilon_{jk}^A \quad (14)$$

Three-way epistasis is computed by measuring the difference between the observed triple-mutant fitness $f_{ijk}$ and the expected fitness in (14) when only pairwise interactions are included. Thus, the three-way additive epistasis measure $\epsilon_{ijk}^A$ is given by

$$\epsilon_{ijk}^A = f_{ijk} - \left[ f_i + f_j + f_k + \epsilon_{ij}^A + \epsilon_{ik}^A + \epsilon_{jk}^A \right] \\ = f_{ijk} - f_{ij} - f_{ik} - f_{jk} + f_i + f_j + f_k. \quad (15)$$

As in the pairwise case, the sign of the three-way epistatic measure $\epsilon_{ijk}^A$ determines the sign of the interaction: if $\epsilon_{ijk}^A > 0$, then there is a *positive* three-way interaction between loci $i, j, k$ – in the sense that the fitness $f_{ijk}$ of the triple mutant is larger than the expected fitness in (14)

when only pairwise interactions are present – while if $\epsilon_{ijk}^A < 0$, then there is a *negative* three-way interaction between loci $i, j, k$.

Our derivation of the three-way epistasis measure $\epsilon_{ijk}^A$ is easily extended to higher-order interactions. The additive $K$-way epistasis measure $\epsilon_{i_1 \cdots i_K}^A$ is defined recursively as

$$\epsilon_{i_1 \cdots i_K}^A = f_{i_1 \cdots i_K} - \left[ \left( \sum_{j=1}^{K} f_{i_j} \right) + \left( \sum_{1 \leq j_1 < j_2 \leq K} \epsilon_{i_{j_1} i_{j_2}}^M \right) + \cdots + \left( \sum_{1 \leq j_1 < \cdots < j_{K-1} \leq K} \epsilon_{i_{j_1} \cdots i_{j_{K-1}}}^A \right) \right]. \quad (16)$$

The $K$-way epistasis measures $\epsilon_{i_1 \cdots i_K}^A$ are proportional to two other measures of epistasis: (1) the $K$-th order Walsh coefficient used to quantify background-averaged epistasis among $K$ genetic loci[30–32] and (2) the $K$-th order interaction coefficients of a linear regression model, which we discuss in more detail in the following section.

**Multiplicative fitness model.** We derive formulae for epistasis in a multiplicative fitness model by using the equivalence between multiplicative fitness and additive log fitness. For example, the 3-way epistasis measure $\epsilon_{ijk}^M$ in the multiplicative model is given by

$$\epsilon_{ijk}^M = \frac{f_{ijk}}{f_i f_j f_k \epsilon_{ij}^M \epsilon_{ik}^M \epsilon_{jk}^M} = \frac{f_{ijk} f_i f_j f_k}{f_{ij} f_{ik} f_{jk}}. \quad (17)$$

As in the pairwise setting, the sign of interaction is determined by the difference between the multiplicative measure $\epsilon_{ijk}^M$ and 1, or equivalently by the $\text{sgn}(\log \epsilon_{ijk}^M)$ of the logarithm of the epistasis measure $\epsilon_{ijk}^M$.

Using (16), then the $K$-way epistasis measure $\epsilon_{i_1 \cdots i_K}^M$ in the multiplicative model is defined recursively by

$$\epsilon_{i_1 \cdots i_K}^M = \frac{f_{i_1 \cdots i_K}}{\left( \prod_{j=1}^{K} f_{i_j} \right) \left( \prod_{1 \leq j_1 < j_2 \leq K} \epsilon_{i_{j_1} i_{j_2}}^M \right) \cdots \left( \prod_{1 \leq j_1 < \cdots < j_{K-1} \leq K} \epsilon_{i_{j_1} \cdots i_{j_{K-1}}}^M \right)}. \quad (18)$$

Recent work in the genetics[24,26] and drug interaction[27] claim to measure three-way epistasis using a multiplicative fitness model. However, they do not measure three-way epistasis the multiplicative epistasis formula (17) but instead derive a chimeric formula using both additive and multiplicative measurement scales:

$$\epsilon_{ijk}^C = f_{ijk} - f_i f_j f_k - \epsilon_{ij}^C f_k - \epsilon_{ik}^C f_j - \epsilon_{jk}^C f_i. \quad (19)$$

We call $\epsilon_{ijk}^C$ the *chimeric* three-way epistasis measure. In these applications, the sign of the interaction is determined by the $\text{sgn}(\epsilon_{ijk}^C)$ of the chimeric measure $\epsilon_{ijk}^C$.

Despite the claim that the chimeric measure $\epsilon_{ijk}^C$ is derived from a multiplicative fitness model, it is clear by inspection that the three-way chimeric measure $\epsilon_{ijk}^C$ is not equal to the multiplicative three-way epistasis measure $\epsilon_{ijk}^M$. However, unlike in the pairwise setting, even the *signs* of these two measures disagree (Fig. 1B). We demonstrate in Supplementary Note 2 that even when $\epsilon_{ijk}^M = 1$ – that is, there is no three-way epistasis – the chimeric three-way epistasis measure $\epsilon_{ijk}^C$ may still indicate either positive or negative three-way epistasis.

Tekin et al.[27] extended the three-way chimeric epistasis formula (19) by heuristically deriving chimeric formulae for 4-way and 5-way epistasis. For example, their chimeric formula for 4-way epistasis is given by

$$\epsilon_{ijk\ell}^C = f_{ijk\ell} - f_i f_{jk\ell} - f_j f_{ik\ell} - f_k f_{ij\ell} - f_\ell f_{ijk} - f_{ij} f_{k\ell} - f_{ik} f_{j\ell} - f_{jk} f_{i\ell} \\ + 2 f_i f_j f_{k\ell} + 2 f_i f_k f_{jl} + 2 f_i f_\ell f_{jk} + 2 f_j f_k f_{i\ell} + 2 f_j f_\ell f_{ik} + 2 f_k f_\ell f_{ij} - 6 f_i f_j f_k f_\ell.$$

As in three-way epistasis, the sign of the 4-way and 5-way chimeric epistasis measures derived by Tekin et al.[27] do not match the signs of the corresponding multiplicative epistasis measures (Fig. 1C). This fundamental disagreement motivates a deeper mathematical

understanding of these epistasis measures, which we explore in the following section.

## Multivariate Bernoulli distribution

In the previous section, we defined quantitative measures of epistasis for two standard null models for fitness: the additive model and the multiplicative model. Nevertheless, some recent papers use a multiplicative fitness model but instead use an epistasis measure which is a *chimera* of both multiplicative and additive measurement scales. Here, we unify these different epistasis measures using the *multivariate Bernoulli* distribution from probability theory[29].

The multivariate Bernoulli distribution describes any distribution on $\{0, 1\}^L$, i.e., binary strings of length $L$, for $L \geq 2$. The multivariate Bernoulli distribution has three different parameterizations which are used throughout the literature[29,47]. We start by describing these parametrizations for the simplest such distribution: a *bivariate* Bernoulli distribution over binary strings of length $L = 2$.

**Bivariate Bernoulli distribution.** Suppose that $X = (X_1, X_2) \in \{0, 1\}^2$ is distributed according to a bivariate Bernoulli distribution. A distribution on $X$ is specified by the parameters $p_{00}, p_{01}, p_{10}, p_{11}$, where $p_{x_1 x_2} = P(X_1 = x_1, X_2 = x_2)$ is the probability of $(x_1, x_2)$. The parameters $\mathbf{p} = (p_{00}, p_{01}, p_{10}, p_{11})$ are sometimes called the *general* parameters[29]. Note that since $p_{00} + p_{01} + p_{10} + p_{11} = 1$, only three such parameters are needed to define the distribution.

The probability density function (PDF) $P(X_1, X_2)$ of $X = (X_1, X_2)$ has the form

$$
\begin{aligned}
P(X_1, X_2) &= p_{00}^{(1-X_1)(1-X_2)} p_{01}^{(1-X_1)X_2} p_{10}^{X_1(1-X_2)} p_{11}^{X_1 X_2} \\
&= \exp\left[\log p_{00} + \left(\log \frac{p_{10}}{p_{00}}\right)X_1 + \left(\log \frac{p_{01}}{p_{00}}\right)X_2 + \left(\log \frac{p_{11} p_{00}}{p_{10} p_{01}}\right)X_1 X_2\right].
\end{aligned}
\tag{20}
$$

In other words, the PDF $P(X_1, X_2)$ follows a log-linear model of the form

$$
\log P(X_1, X_2) = \beta_0 + \beta_1 X_1 + \beta_2 X_2 + \beta_{12} X_1 X_2
\tag{21}
$$

for constants $\beta_0, \beta_1, \beta_2, \beta_{12} \in \mathbb{R}$. There is a one-to-one correspondence between the general parameters $\mathbf{p} = (p_{00}, p_{01}, p_{10}, p_{11})$ and the constants $\boldsymbol{\beta} = (\beta_0, \beta_1, \beta_2, \beta_{12})$. Thus, a bivariate Bernoulli distribution is also parametrized by the parameters $\boldsymbol{\beta}$, also known as the *natural* parameters of the distribution[29]. As with the general parameters $\mathbf{p}$, we note that only three out of the four parameters $\beta_0$, $\beta_1$, $\beta_2$, $\beta_{12}$ are needed to fully specify a distribution. We also note that independence between the random variables $X_1$ and $X_2$ is described by the parameter $\beta_{12}$, where $X_1$ and $X_2$ are independent if and only if $\beta_{12} = 0$.

Equation (21) demonstrates that $X = (X_1, X_2)$ follows an *exponential family* distribution, a wide class of distributions that includes many common distributions including normal distributions or Poisson distributions. In particular, using the terminology of exponential families, equation (21) shows that the *sufficient statistics* of $X$ are $X_1, X_2$, and $X_1 X_2$, with corresponding *canonical* parameters $\beta_1, \beta_2$, and $\beta_{12}$[48,91]. As a result, the distribution $P(X)$ is uniquely defined by the *expected values* $E[X_1], E[X_2], E[X_1 X_2]$ of the sufficient statistics, sometimes called the *moments* or the *mean parameters* of the distribution[48]. Thus, we obtain a third parametrization of the distribution $P(X)$ using the moments $\mu_0 = 1$, $\mu_1 = E[X_1]$, $\mu_2 = E[X_2]$, $\mu_{12} = E[X_1 X_2]$. The elements of the vector $\boldsymbol{\mu} = (1, \mu_1, \mu_2, \mu_{12})$ of moments are sometimes called the *mean parameters* of the distribution.

**Multivariate Bernoulli distribution.** The three parametrizations we derived for the bivariate Bernoulli distribution extend to the multivariate Bernoulli distribution. Suppose that $(X_1, \ldots, X_L) \in \{0, 1\}^L$ is distributed according to a multivariate Bernoulli distribution. Then the

distribution $P(X)$ of the random variables $X$ is uniquely specified by one of the three following parametrizations.

1. **General parameters**: These are $2^L$ non-negative values $\mathbf{p} = (p_{x_1 \cdots x_L})_{(x_1, \ldots, x_L) \in \{0,1\}^L}$ satisfying

$$
p_{x_1 \cdots x_L} = P(X_\ell = x_\ell \text{ for } \ell = 1, \ldots, L).
\tag{22}
$$

For example if $L = 3$, then $p_{010} = P(X_1 = 0, X_2 = 1, X_3 = 0)$ and $p_{110} = P(X_1 = 1, X_2 = 1, X_3 = 0)$. Note that since $\sum_{(x_1, \ldots, x_L) \in \{0,1\}^L} p_{x_1 \cdots x_L} = 1$, only $2^L - 1$ values $p_{x_1 \cdots x_L}$ are necessary to define the distribution.

2. **Natural/canonical parameters**: These are $2^L$ real numbers $\boldsymbol{\beta} = (\beta_S)_{S \subseteq [L]} \in \mathbb{R}$ satisfying

$$
\log P(X_1, \ldots, X_L) = \sum_{S \subseteq [L]} \beta_S \cdot \prod_{i \in S} X_i.
\tag{23}
$$

Similar to the general parameters $p_i$, only $2^L - 1$ values $\beta_S$ are necessary to uniquely define the distribution. Typically, the parameter $\beta_\varnothing$, often called a normalizing constant or a *partition function* of the distribution, is left unspecified. As noted in the bivariate setting, equation (23) shows that the multivariate Bernoulli is an exponential family distribution with $2^L - 1$ sufficient statistics of the form $\prod_{i \in S} X_i$ for subsets $S$ with $|S| > 0$. Moreover, by rewriting (23) as

$$
\log p_{x_1 \cdots x_L} = \beta_\varnothing + \left(\sum_{i=1}^{L} \beta_i x_i\right) + \left(\sum_{1 \leq i_1 < i_2 \leq L} \beta_{i_1 i_2} \cdot x_{i_1} x_{i_2}\right) + \cdots + (\beta_{1 \cdots L} \cdot x_1 \cdots x_L),
\tag{24}
$$

we observe that the natural parameters $\boldsymbol{\beta}$ correspond to interaction coefficients in a log-linear regression model with response variables $\mathbf{p}$. For example, the natural parameter $\beta_{12}$ is the coefficient of the interaction term $x_1 x_2$.

3. **Moments/mean parameters**: These are $2^L$ real numbers $\boldsymbol{\mu} = (\mu_S)_{S \subseteq [L]}$ satisfying

$$
\mu_S = E\left[\prod_{s \in S} X_s\right].
\tag{25}
$$

For example if $L = 3$, then $\mu_{13} = E[X_1 X_3]$ while $\mu_{12} = E[X_1 X_2]$. The mean parameters $\{\mu_S\}_{|S| > 0}$ are sufficient statistics for the multivariate Bernoulli distribution, as seen in the exponential family form (23) of the multivariate Bernoulli distribution.

We note that all three parametrizations, as well as the fitness values $f$ and epistasis measures $\epsilon$, can be defined either in terms of subsets $S \subseteq [L]$ as with the natural parameters $\boldsymbol{\beta}$ and moments $\boldsymbol{\mu}$, or in terms of binary strings $x_1 \cdots x_L$ as with the general parameters $\mathbf{p}$. We use both definitions interchangeably, with the convention that a subset $S \subseteq [L]$ corresponds to the binary string $x_1 \cdots x_L$ with $x_i = 1_{\{i \in S\}}$.

Moreover, when written as vectors indexed by binary strings, the three parametrizations $\boldsymbol{\beta}, \boldsymbol{\mu}, \mathbf{p}$ of the multivariate Bernoulli are related to each through different linear transformations involving a matrix operation known as the *Kronecker product* (see Supplementary Note 3 for specific formulae). Interestingly, several papers quantify epistasis using the *Walsh-Hadamard transform* which is also defined in terms of Kronecker products[30–32]. This connection is not a coincidence; in the next section we show that the Walsh-Hadamard transform is closely related to the parametrizations of the multivariate Bernoulli.

## Unifying epistasis measures with the multivariate Bernoulli

The multivariate Bernoulli distribution provides an elegant means by which to describe the different epistasis formulae in the literature. We model the genotype $(X_1, \ldots, X_L) \in \{0, 1\}^L$ as a random variable

distributed according to a multivariate Bernoulli distribution. The parametrizations of the multivariate Bernoulli correspond to different features of the genotype, as we demonstrate next.

**Multiplicative and additive epistasis measures.** We start by relating the multiplicative epistasis formula (18) to the multivariate Bernoulli distribution. A careful reader may observe that the natural parameter $\beta_{12} = \log \frac{p_{11}p_{00}}{p_{10}p_{01}}$ in the bivariate Bernoulli distribution (21) bears close resemblance to the multiplicative epistasis measure in equation (10). Specifically, if the fitness $f_{x_1 x_2}$ of each genotype $(x_1, x_2) \in \{0, 1\}^2$ is proportional to the probability $p_{x_1 x_2}$ of that genotype in the multivariate Bernoulli, then the natural parameter $\beta_{12}$ is equal to the logarithm $\log \epsilon_{12}^M$ of the multiplicative epistasis measure $\epsilon_{12}^M$. Thus, for $L = 2$ loci, epistasis is measured by the natural parameters $\boldsymbol{\beta}$ of a bivariate Bernoulli distribution.

We prove that this observation is not specific to the bivariate Bernoulli distribution with $L = 2$ loci, and in fact generalizes to any number $L$ of loci. Specifically, we prove that if the fitness $f_{x_1 \cdots x_L}$ of genotype $(x_1, ..., x_L)$ is proportional to the probability $p_{x_1 \cdots x_L}$ of observing the genotype, then for each subset $S \subseteq [L]$ of loci, the natural parameter $\beta_S$ equals the logarithm $\log \epsilon_S^M$ of the corresponding multiplicative epistasis measure as defined in Equation (18).

**Theorem 1.** Let $f_{\mathbf{x}} \in \mathbb{R}$ be fitness values for genotypes $\mathbf{x} = (x_1, ..., x_L) \in \{0, 1\}^L$ such that $f_{\mathbf{x}} = c \cdot p_{\mathbf{x}}$ for some constant $c > 0$ and for some multivariate Bernoulli random variable $(X_1, ..., X_L)$ with general parameters $\mathbf{p} = (p_{\mathbf{x}})_{\mathbf{x} \in \{0,1\}^L}$. Then for all subsets $S \subseteq \{1, ..., L\}$ of loci, the log-multiplicative epistasis measure $\log \epsilon_S^M$ is equal to the interaction parameter $\beta_S$ of the random variable $(X_1, ..., X_L)$.

By using the equivalence between multiplicative fitness values and additive log-fitness values, we also derive a similar probabilistic interpretation of the additive epistasis formula. Specifically, if fitness $f_{x_1 \cdots x_L}$ is proportional to the *log*-probability $\log p_{x_1 \cdots x_L}$ of observing the genotype $(x_1, ..., x_L)$, then for each subset $S = \{i_1, ..., i_k\} \subseteq [L]$ of loci, the natural parameter $\beta_S$ equals the logarithm $\log \epsilon_{i_1, \cdots, i_k}^A$ of the corresponding additive epistasis measure as defined in Equation (16). We formalize this observation as the following Corollary of Theorem 1.

**Corollary 1.** Let $f_{\mathbf{x}} \in \mathbb{R}$ be fitness values for genotypes $\mathbf{x} = (x_1, ..., x_L) \in \{0, 1\}^L$ such that $f_{\mathbf{x}} = c \cdot \log p_{\mathbf{x}}$ for some constant $c > 0$ and for some multivariate Bernoulli random variable $(X_1, ..., X_L)$ with general parameters $\mathbf{p} = (p_{\mathbf{x}})_{\mathbf{x} \in \{0,1\}^L}$. Then for all subsets $S \subseteq \{1, ..., L\}$ of loci, the log-additive epistasis measure $\log \epsilon_S^A$ is equal to the interaction parameter $\beta_S$ of the random variable $(X_1, ..., X_L)$.

We note that Theorem 1 follows from Lemma 3.1 in ref. 29 which states a formula relating the general parameters $\mathbf{p}$ and the natural parameters $\boldsymbol{\beta}$ of a multivariate Bernoulli distribution. Theorem 1 follows by showing that the right-hand side of the formula in Lemma 3.1 is equal to the multiplicative epistasis measure.

The assumption that the probability $p_{x_1 \cdots x_L}$ of observing a genotype $(x_1, ..., x_L)$ is derived from its fitness $f_{x_1 \cdots x_L}$ is often used in generative models for estimating the fitness of protein structures from sequence data[39]. Moreover, many real-world fitness datasets – including the yeast fitness data and many of the protein datasets analyzed in the Results – measure the fitness of a genotype $\mathbf{x}$ in terms of its relative frequency in a large population of genotypes, i.e., its probability $p_{\mathbf{x}}$.

We also note that the statistical problem of estimating the natural parameters $\boldsymbol{\beta}$ or mean parameters $\boldsymbol{\mu}$ of a multivariate Bernoulli distribution from samples $(X_1, ..., X_L)$ of the distribution is computationally hard[48]. The reason why we are able to use relatively simple formulae (16), (18) to compute the natural parameters $\boldsymbol{\beta}$ is because in this setting, we have both samples $(X_1, ..., X_L)$ *and* their corresponding probabilities $P(X_1, ..., X_L)$, i.e., the fitness values $f$.

**Relationship with (log-)linear regression.** Under the assumption that the fitness values $f_{x_1 \cdots x_L}$ are proportional to the genotype probabilities $p_{x_1 \cdots x_L}$, then (24) is a log-linear regression model of the form

$$\log f_{x_1 \cdots x_L} = \beta_\varnothing + \left( \sum_{i=1}^L \beta_i x_i \right) + \left( \sum_{1 \le i_1 < i_2 \le L} \beta_{i_1 i_2} \cdot x_{i_1} x_{i_2} \right) + \cdots + (\beta_{1 \cdots L} \cdot x_1 \cdots x_L). \quad (26)$$

Thus, Theorem 1 shows that computing the multiplicative epistasis measure $\epsilon^M$ is equivalent to computing the interaction parameters of the log-linear regression in (26). The interaction parameters of regression are a standard approach for quantifying epistasis in GWAS and QTL analyses[5]. Equation (26) is sometimes also called the "Taylor series expansion" of a fitness landscape[92].

Similarly, Corollary 1 demonstrates the equivalence between the additive epistatis measure $\epsilon^A$ and the coefficients of a linear regression model with response variables equal to the fitness values. Specifically, under the assumption that the fitness values $f_{x_1 \cdots x_L}$ are proportional to the logarithm $\log p_{x_1 \cdots x_L}$ of the genotype probabilities, then computing the additive epistasis measures $\epsilon^A$ is equivalent to computing the interaction parameters $\boldsymbol{\beta}$ of the following linear regression model

$$f_{x_1 \cdots x_L} = \beta_\varnothing + \left( \sum_{i=1}^L \beta_i x_i \right) + \left( \sum_{1 \le i_1 < i_2 \le L} \beta_{i_1 i_2} \cdot x_{i_1} x_{i_2} \right) + \cdots + (\beta_{1 \cdots L} \cdot x_1 \cdots x_L). \quad (27)$$

In this way, Theorem 1 and Corollary 1 provide a connection between the multiplicative and additive epistasis measures and the interaction coefficients of log-linear and linear regression models, respectively.

**Relationship with case-control GWAS.** We also prove that the natural parameters $\boldsymbol{\beta}$ of an MVB with three variables are closely related to the two standard approaches for measuring pairwise SNP-SNP interactions in a case-control GWAS: logistic regression and conditional independence testing[49]. Specifically, suppose we are given genotype $(X_1, X_2) \in \{0, 1\}^2$ and (binary) disease status $D \in \{0, 1\}$. Then the joint random variable $(X_1, X_2, D)$ follows an MVB distribution, where the log-probability $\log P(X_1, X_2, D)$ is given by the following expression in terms of the natural parameters $\boldsymbol{\beta}$:

$$\log P(X_1, X_2, D) = \beta_0 + \beta_1 X_1 + \beta_2 X_2 + \beta_d D + \beta_{12} X_1 X_2 + \beta_{1d} X_1 D + \beta_{2d} X_2 D + \beta_{12d} X_1 X_2 D. \quad (28)$$

We note that there is a natural approach for representing GWAS data from diploid genomes (with {0, 1, 2}-valued allelic states) using binary random variables $X_1, X_2$, as described in ref. 93.

We show that the logistic regression approach for measuring pairwise interactions is equivalent to computing $\beta_{12d}$, while the conditional independence test is equivalent to testing the null hypothesis $H_0$: $\beta_{12} = \beta_{12d} = 0$. See Supplementary Note 5 for details.

**Chimeric epistasis measure.** The multivariate Bernoulli also provides a way of rigorously defining the pairwise and higher-order chimeric epistasis measures using *joint cumulants*. Joint cumulants are a concept from probability theory used to quantify higher-order interactions between random variables. For example, the 2nd order joint cumulant $\kappa(X, Y)$ of two random variables $X, Y$ is given by

$$\kappa(X, Y) = E[XY] - E[X]E[Y], \quad (29)$$

and is equal to the covariance $\text{Cov}(X, Y)$. The 3rd order joint cumulant $\kappa(X, Y, Z)$ of three random variables is given by

$$\kappa(X, Y, Z) = E[XYZ] - \kappa(X, Y)E[Z] - \kappa(X, Z)E[Y] - \kappa(Y, Z)E[X]. \quad (30)$$

Under the assumption that the fitness $f_{x_1 \cdots x_L}$ of a genotype $(X_1, \ldots, X_L)$ is equal to the corresponding *moment* $\mu_{x_1, \ldots, x_L}$, we define the $K$-way chimeric epistatic measure $\epsilon^C_{i_1 \cdots i_K}$ as the $K$-th order *joint cumulant* $\kappa(X_{i_1}, \ldots, X_{i_K})$ of the random variables $X_{i_1}, \ldots, X_{i_K}$.

**Definition 1.** Let $f_{\mathbf{x}} \in \mathbb{R}$ be fitness values for genotypes $\mathbf{x} = (x_1, \ldots, x_L) \in \{0, 1\}^L$ such that $f_{x_1 \cdots x_L} = c \cdot \mu_{x_1, \ldots, x_L}$ for some constant $c > 0$ and for some multivariate Bernoulli random variable $(X_1, \ldots, X_L)$ with moments $\mu_{x_1, \ldots, x_L} = E[X_1^{x_1} \cdots X_L^{x_L}]$. The *chimeric epistasis measure* $\epsilon^C_{i_1 \cdots i_K}$ is the joint cumulant $\kappa(X_{i_1}, \ldots, X_{i_K})$ of the random variables $X_{i_1}, \ldots, X_{i_K}$.

Our definition of the $K$-th order chimeric epistasis measure $\epsilon^C_{i_1 \cdots i_K}$ as the $K$-th order joint cumulant formalizes the heuristic derivation of the chimeric measure in previous literature. Almost every paper that uses the chimeric epistasis measures $\epsilon^C$ does not even mention the joint cumulant. Two notable exceptions are refs. [40,45], which use the joint cumulants to derive formulae for 3-way, 4-way, and 5-way interactions between drugs. However, refs. [40,45] do not rigorously define a probability distribution nor the random variables whose joint cumulant they compute.

At the same time, our formal definition of the chimeric epistasis measure $\epsilon^C$ reveals two critical issues with the chimeric formula. First, the assumption that the fitness values $f$ are equivalent to the moments of an MVB random variable is not biologically reasonable for higher-order interactions between three or more loci. This assumption implies that the fitness of a particular genotype depends on the probability of many other genotypes. For example, making this assumption for $L = 4$ loci, the fitness $f_{1100}$ of a double mutant is equal to the moment $E[X_1 X_2]$, which is equal to

$$E[X_1 X_2] = P(X_1 = 1, X_2 = 1) = p_{1100} + p_{1101} + p_{1110} + p_{1111}. \quad (31)$$

However, it is not clear why the fitness $f_{1100}$ of a *single* genotype, 1100, should equal depend on the probabilities of *four different* genotypes, 1100, 1101, 1110, and 1111.

The second issue is that joint cumulants are not necessarily an appropriate measure of higher-order interactions between *binary* random variables. The differences between the joint cumulants and natural parameters $\boldsymbol{\beta}$ have been previously investigated in the neuroscience literature, as both quantities have been used to quantify higher-order interactions in neuronal data. For example, Staude et al.[34,35] write that the joint cumulants $\kappa$ and natural parameters $\boldsymbol{\beta}$ measure mathematically distinct types of higher-order interactions, and that each quantity may be appropriate for different applications. In particular, Staude et al.[34,35] note that the joint cumulants measure higher-order interactions between random variables in terms of "additive common components", while the natural parameters $\boldsymbol{\beta}$ measure *"to what extent the probability of certain binary patterns can be explained by the probabilities of its sub-patterns"*. It follows that for binary mutation data, the natural parameters $\boldsymbol{\beta}$ correspond exactly with the epistasis we aim to measure, i.e., how the fitness of a binary pattern can be explained by the fitness of its "sub-patterns", while the joint cumulants do not.

**Walsh coefficients and background-averaged epistasis.** The multivariate Bernoulli distribution also provides a probabilistic interpretation of the *Walsh coefficients* that are used to measure "background-averaged" epistasis[30–32,42,43]. The Walsh coefficients $\mathbf{u} = [u_{x_1 \cdots x_L}] \in \mathbb{R}^{2^L}$,

i.e., a vector indexed by binary strings, are defined by

$$\mathbf{u} = \boldsymbol{\Psi} \mathbf{f} \quad (32)$$

where $\boldsymbol{\Psi} = \begin{pmatrix} 1 & 1 \\ 1 & -1 \end{pmatrix}^{\otimes L} \in \mathbb{R}^{2^L \times 2^L}$ is a *Hadamard* matrix[32] and $\mathbf{f} = [f_{x_1 \cdots x_L}] \in \mathbb{R}^{2^L}$ is the vector of fitness values indexed by binary strings. Equation (32) is known as the *Walsh-Hadamard* transformation, sometimes also called the *Walsh* or *Fourier-Walsh* transform; see refs. [30,32] for more details.

We prove that if the fitness values $\mathbf{f}$ are equal to probabilities $\mathbf{p}$ of a multivariate Bernoulli random variable $(X_1, \ldots, X_L)$, then the Walsh coefficients $\mathbf{u}$ are equal to the *moments* of $(1 - 2X_1, \ldots, 1 - 2X_L) \in \{-1, 1\}^L$, i.e., a linear transformation of the random variable $(X_1, \cdots, X_L)$ such that it takes values in $\{-1, 1\}^L$ instead of $\{0, 1\}^L$.

**Theorem 2.** Let $(X_1, \ldots, X_L) \in \{0, 1\}^L$ be distributed according to a multivariate Bernoulli distribution with general parameters $\mathbf{f}$, and define $Y_\ell = 1 - 2X_\ell \in \{-1, 1\}$ for $\ell = 1, \ldots, L$. Define $\mathbf{u} = [u_{x_1 \cdots x_L}] \in \mathbb{R}^{2^L}$ as in (32). Then $u_{x_1 \cdots x_L} = E[Y_1^{x_1} \cdots Y_L^{x_L}]$.

Theorem 2 gives a probabilistic interpretation of the Walsh coefficients $\mathbf{u}$. Interestingly, the Walsh coefficients $\mathbf{u}$ assume an *additive* fitness model[30,32] while Theorem 2 requires that the fitness values $\mathbf{f}$ are equal to the probabilities $\mathbf{p}$, an assumption corresponding to the *multiplicative* fitness model (Table 1).

**Relationship to theoretical genetics models.** We note that some previous works in theoretical genetics by Barton and Turelli (e.g., ref. [94]) also model the genotype with an MVB. However, their approach is substantially different from ours. Barton and Turelli model linkage disequilibrium between $k$ loci $X_{i_1}, \ldots, X_{i_k}$ using the $k$-way central moment $\tilde{\mu}(X_{i_1}, \ldots, X_{i_k}) = E\left[\prod_{j=1}^{k}(X_{i_j} - E[X_{i_j}])\right]$ of the genotype distribution $P(X_1, \ldots, X_n)$. Barton and Turelli model epistasis using coefficients that are not related to the genotype distribution $P(X_1, \ldots, X_n)$. In contrast, we model epistasis with the natural parameters $\boldsymbol{\beta}$ of the genotype distribution $P(X_1, \ldots, X_n)$, as described in Multiplicative and additive epistasis measures.

Interestingly, Barton and Turelli's 3-way linkage disequilibrium term, i.e., $\tilde{\mu}(X_{i_1}, X_{i_2}, X_{i_3})$, is equal to the 3-way joint cumulant $\kappa(X_{i_1}, X_{i_2}, X_{i_3})$ implicitly used by Kuzmin et al.[24,26] to measure 3-way epistasis. This equivalence is because the $k$-way central moment $\tilde{\mu}(X_{i_1}, \ldots, X_{i_k})$ is equal to the $k$-way joint cumulant $\kappa(X_{i_1}, \ldots, X_{i_k})$ for $k = 1, 2, 3$. However, for $k \geq 4$, the $k$-way linkage disequilibrium term used by Barton and Turelli is not equal to the $k$-way joint cumulant.

## Simulating fitness values

We simulate fitness values $f_{\mathbf{x}}$ for genotypes $\mathbf{x} = (x_1, \ldots, x_L)$ with $L = 10$ loci and $K$-way interactions using the following two different approaches. For both models, we divide all of the fitness values $\mathbf{f}$ by $f_{\varnothing}$ so that $f_{\varnothing} = 1$.

**Multiplicative fitness model.** We draw interaction parameters $\beta_S \sim \text{Uni}(-0.5, 0.5)$ for each subset $S \subseteq \{1, \ldots, L\}$ of loci with size $|S| \leq K$. We set the fitness $f_{\mathbf{x}}$ of genotype $\mathbf{x} = (x_1, \ldots, x_L)$ as

$$\log f_{\mathbf{x}} = \left( \sum_{\substack{S \subseteq \{1, \ldots, L\} \\ |S| \leq K}} \beta_S \left( \prod_{i \in S} x_i \right) \right) + \epsilon_{\mathbf{x}} \quad (33)$$

where $\epsilon_{\mathbf{x}} \sim N(0, \sigma^2)$ are independent and identically distributed Gaussian random variables with mean zero and variance $\sigma^2$. We note that

our noise model $\epsilon$ differs from the widely-used Rough Mount Fuji fitness model[95], where the noise terms are the source of epistasis[96–98].

**NK model.** We simulate fitness values **f** according to the NK model with the code used by ref. 31. Because ref. 31 uses an additive fitness model, we exponentiate the fitness values from the NK model.

**Epistasis between protein mutations**

The analysis in Epistasis between protein mutations was performed using publicly available DMS data for the following proteins/RNA molecules:

- the *E.coli* metabolic protein FolA[72], where the fitness of a genotype is measured by the logarithm of its relative frequency in a large population;
- the *Streptococcus pyogenes* Cas9 (SpCas9) nuclease[38], where the fitness of a genotype is measured by the logarithm of its relative frequency in a large population;
- the immunoglobulin-binding protein G domain B1 (GB1) expressed in Streptococcal bacteria[73,74], where the fitness of a genotype is measured by its relative frequency in a large population;
- the Omicron BA.1 variant of the SARS-CoV-2 virus[75], where the fitness of a genotype is measured by the logarithm of its binding affinity relative to the Wuhan Hu-1 strain;
- the *Entacmaea quadricolor* fluorescent protein (eqFP611)[76], where the fitness of a genotype is measured by (a normalized version of) its relative frequency in a large population;
- the *Aequorea victoria* green fluorescent protein (avGFP)[77], where the fitness of a genotype is measured by the logarithm of its fluorescence;
- the green fluorescent proteins (GFPs) from ref. 10, where the fitness of a genotype is measured by its fluorescence;
- yeast tRNA[78], where the fitness of a genotype is measured by the logarithm of its relative frequency in a large population; and
- the *Chlamydomonas reinhardtii* flavin mononucleotide (FMN)-based fluorescent protein (CreiLOV)[79], where the fitness of a genotype is measured by the logarithm of its fluorescence.

We aim to use these protein fitness landscapes to directly compare the multiplicative and chimeric epistasis measures. However, the quantitative trait used to measure fitness – and thus the appropriate scale for measuring epistasis (additive or multiplicative) – varies across the different proteins. In particular, the original publications for the FolA protein, SpCas9 protein, eqFP611 protein, COVID spike protein, avGFP protein, yeast tRNA, and CreiLOV protein assume that fitness is measured using an additive scale (e.g., by measuring fitness as the logarithm of the relative frequency of a genotype). Since our aim is to demonstrate the disagreement between the multiplicative and chimeric measures, we first transform the fitness measurements to a multiplicative scale by exponentiating the fitness values **f**, i.e., $f \rightarrow e^f$, before computing the multiplicative measure. This allows us to directly compare the multiplicative epistasis measure $\epsilon^M$ with the chimeric epistasis measure $\epsilon^C$, which implicitly assumes fitness values are measured using a multiplicative scale.

For each protein and each interaction order $K$, we compute the multiplicative (resp. chimeric) measure $\epsilon^M$ (resp. $\epsilon^C$) across all $K$-tuples of mutational events. We note that for some proteins, the fitness of multiple mutations at a given locus is measured (e.g., all 3 possible base-pair substitutions at a locus, or all 19 possible amino acid substitutions); for these proteins, we consider each possible mutation at a given genetic locus as a separate mutational event. Furthermore, we only compute the epistasis measure for a given $K$ tuple of mutational events if all fitness values $f$ for the $2^K$ genotypes are greater than a threshold $\epsilon$, which we set to $\epsilon = 0.01$ following the convention from ref. 73.

**Reporting summary**

Further information on research design is available in the Nature Portfolio Reporting Summary linked to this article.

## Data availability

The datasets used in this study were obtained through publicly available repositories. The synthetic genetic array (SGA) data used to analyze three-way epistasis in yeast was obtained from https://doi.org/10.5061/dryad.g79cnp5m9 and https://boonelab.ccbr.utoronto.ca/supplement/kuzmin2018/supplement.html. The drug response data used to analyze higher-order interactions between drug combinations was obtained from the supplementary information of ref. 40. The deep mutational scanning (DMS) data used to analyze higher-order interactions between protein mutations was obtained from: https://doi.org/10.5281/zenodo.8228919(FolA); https://github.com/AWHKU/RunMLDE_SpCas9/tree/main(SpCas9); https://github.com/J-SNACKKB/FLIP/tree/main/splits/gb1(GB1); https://github.com/desai-lab/compensatory_epistasis_omicron/tree/main/Titeseq/results/Kds(Omicron BA.1 variant of SARS-CoV-2); the supplementary information of ref. 76 (eqFP611); https://doi.org/10.6084/m9.figshare.3102154(avGFP); https://github.com/aequorea238/Orthologous_GFP_Fitness_Peaks(GFPs); the supplementary information of ref. 78 (tRNA); and the supporting information of ref. 79 (CreiLOV). Source data are provided in this paper.

## Code availability

The code for our analyses is located in our public GitHub repository and is available here: 10.5281/zenodo.14426370. Our code requires the following Python packages: `Numpy` ($\geq 1.23.4$), `Matplotlib` ($\geq 3.8.0$), `Pandas` ($\geq 2.1.1$), `Scipy` ($\geq 1.11.2$), `Seaborn` ($\geq 0.12.2$).

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

## Acknowledgements

We thank Daniel Weinreich, Sriram Sankararaman, and Boyang Fu for their helpful comments and suggestions. U.C. was supported by a National Science Foundation Graduate Research Fellowship and the Siebel Scholars program. B.J.A. gratefully acknowledges financial support from the Schmidt DataX Fund at Princeton University, made possible through a major gift from the Schmidt Futures Foundation. This research is also supported by National Cancer Institute (NCI) grants U24CA248453 and U24CA264027 to B.J.R.

## Author contributions

Conceptualization: B.J.R., U.C., and B.A.; methodology: U.C. and B.J.R.; software: U.C. and B.A.; validation: B.A.; formal analysis: U.C., B.A., and B.J.R.; investigation: U.C., B.A., and B.J.R.; data curation: U.C. and B.A.; writing—original draft: U.C., B.A., and B.J.R.; writing—review and editing: U.C., B.A., and B.J.R.; visualization: U.C. and B.A.; supervision: B.J.R.

## Competing interests
The authors declare no competing interests.
