## [Transparent Peer Review file · Nature Communications]

Beware the chimera when quantifying higher-order epistasis

Corresponding Author: Professor Benjamin Raphael

Version 0:

Reviewer comments:

Reviewer #1

(Remarks to the Author)
See PDF

(Remarks on code availability)

Reviewer #2

(Remarks to the Author)

This paper makes a useful point about the illogicality of measuring epistatic effects between mutations by applying an additive measure to a multiplicative model of the null expectation of a trait in the absence of interactions (their 'chimera'). (Epistasis in the sense used here is equivalent to interactions between main effects in the analysis of variance, and this concept introduced by RA Fisher, the inventor of ANOVA, in 1918 - oddly, they do not cite Fisher until ref. 130.)

The paper is, however, rather poorly presented, with a lot of repetition, and the quite simple message gets lost in a maze of detail. It could usefully be made much shorter and clearer. In addition, there are a number of instances in which cited work has been incorrectly described, which needs to be corrected (see Specific Comments). The number of references could be cut considerably, as many of them are reviews with overlapping contents.

They also never mention that they exclusively consider haploid genomes; with diploidy, it is much harder to describe epistasis, as interactions between additive genetic and dominance effects need to be modelled (see any textbook of quantitative genetics). The results are thus of more limited application than they seem to think. The discussion of GWAS etc in humans (p.15) is thus not very relevant.

Importantly, the authors have not brought out a basic point about fitness measures. They are largely focussed on microbial systems, where genetically manipulated strains reproducing asexually are competed against each other. In such systems, the fitness of a strain is generally (but not always) measured by its growth rate, i.e. the rate of change of the logarithm of its number of individuals, or a proxy for it. In contrast, fitness in higher organisms is frequently measured by the expected lifetime reproductive success of a genotype or phenotype, and it is usually assumed that independent effects on survival and probability of reproduction combine multiplicatively, although the justification for this is weak for traits like fecundity.

A multiplicative model is, however, inappropriate for microbial growth rate fitness measures; for strain differences in growth rates, one can use additivity as the baseline model for estimating epistatic parameters. This very basic point seems to have escaped the attention of most workers in the field, despite J-L Chevin having pointed out in 2011 the difference between fitness measures based on discrete generation population genetics models and growth-rate measures, and the resulting errors in interpreting microbial data (Biology Letters 7:210). For example, in ref. 23, a log transform of the ratio of growth rates of mutant to wild-type strains of *E. coli* was used to test for epistasis. It is, of course, illogical to use a log of a log for this purpose. Papkou et al. (ref. 113, SM, pp.9-10) have recently drawn attention to this issue. Population growth rate has a long history as fitness measure. Fisher (1930, The Genetical Theory of Natural Selection, Chap. 2) introduced it (his "Malthusian parameter") as a fitness measure; there are, however, problems in using this for higher organisms with complex life histories (Charlesworth 1994, Evolution in Age-Structured Populations).

The situation is thus quite simple far when microbial growth rates are used as the measure of fitness; just look for deviations

from additivity, for which the statistical machinery is well-worked out (e.g. ref. 41). On p.34, where they describe how they reanalyze a number of datasets, they mention that some studies used additive measures and others multiplicative measures, but don't explain why this is the case. It's not clear why they felt they needed to transform from additive to multiplicative and then compare with chimeric; they could simply make their point by using examples where the authors applied chimeric to multiplicative (e.g. refs. 2 and 42, which use colony size). This would spare the reader much unnecessary detail. For some of the cases, such as fluorescence, the choice of scale is somewhat arbitrary, since the relation between trait and fitness is unknown.

It would also help to make it clearer why there has been such emphasis on multiplicativity in the fitness and population genetics literature. It partly comes about from the fact that it is reasonable to assume that independent effects on survival probability combine multiplicatively, so that single-locus results can be generalised easily to multiple loci (e.g. Haldane 1937 *Am Nat* 71:337), and partly from the fact that the effects of selection on traits subject to directional selection in discrete generation models suggest that multiplicative fitness effects do not generate non-random associations between loci (e.g. Shnol & Kondrashov 1993 *Genetics* 134:995). However, it has long been known that multi-locus systems require additivity of fitness effects for there to be an absence of such associations among polymorphisms maintained by selection when recombination is rare (reviewed by Ewens 2004 *Mathematical Population Genetics*), so that the biological relevance of multiplicativity versus additivity is context dependent.

Finally, I found their justification of the use of the multivariate Bernoulli distribution to analyze fitness epistasis unconvincing. On p.30, they simply say that the components of the probability vector "might be derived from the frequency of observing the genotype .. in a large population". This is incorrect; the frequency of a genotype in a population is only very loosely related to its fitness – see any textbook of population genetics. A justification of what they are doing in the context of microbial experiments that use colony size as a fitness proxy is as follows. The size at time t of a colony with genotype i and growth-rate r_i , relative to its initial size, is $\exp(r_i t)$; call this w_i . Normalise this by the sum of the w_i ; you then get a vector whose components sum to one, which formally behaves like a vector of probabilities (but are not true probabilities). This is similar to what they say on p.31 (after Corollary 1), but removes the requirement of equal initial numbers.

But this is not really needed to get to their Equation 26; if you assume take the log transform of w_i , you are back to a quantity that can be treated in terms of the standard linear model of main effects and interactions. Equation 26 is thus almost trivially obvious as a general form, as is the fact that they end up with the standard Walsh coefficients after the log transform.

To sum up, they have a useful point about some erroneous analyses of fitness epistasis in microbial growth experiments, where a log transform should have been used to get to the correct scale, but this gets hidden in a mass of unnecessary detail.

Specific comments

p.1 I.1 Epistasis is a property of any trait, not just fitness.

I.7, 10 Synergistic vs antagonistic and negative versus positive are essentially different words for the same things; it's confusing to have both of them in the same abstract. Also, neither of them have been defined.

I.15 up Wright never used the term "fitness landscape", although he did invent the concept of a surface of population mean fitness as a function of genotype frequencies, and peaks of fitness associated with individual genotypes.

I.11 up and following The reader will be confused by their defining epistasis in terms of fitness, and then citing material that applies it to other traits, e.g. ref. 3 does not mention fitness. They need to correct this; in fact, they could substitute 'trait' for fitness in much of what follows.

p.2 §.2 It would help to define what is meant by negative vs positive epistasis at this point. The references are not entirely accurate, e.g. ref 23 simply measured the (lack of) departure relation between their (incorrect) measure of relative fitness and number of accumulated mutations, rather than specific interactions between mutations.

§3 The statement about recombination is incorrect; these papers deal with the effect of epistasis on selection for recombination modifiers, not the "quantification of recombination".

p.3 I.13-14 up This is a purely haploid model; diploidy is much more complex (see General Comments).

I.8 up They need to be more precise about what is meant by fitness (see General Comments); 'mean reproductive success' is not very meaningful for single-celled organisms reproducing by binary cell division, and rate of population growth (Malthusian parameter) is an appropriate measure. This paper is concerned entirely with data on such organisms (although this is never made explicit), so the authors might as well stick to this measure.

p.4-5 The material on this page could be presented much more concisely.

p.4 I.1 Ref. 64 is not the latest (1996) edition of this book.

§5 I.5 Ref 23 does not use the chimeric method of estimating pairwise epistasis; I haven't checked the other references.

p.5 Table 1 The Walsh coefficients apply to any quantitative trait, not one confined to (0, 1).

p.6 §2 I.7 This is incorrect; the frequency is proportional to the exponent of the product of time and growth rate; this, of course, assumes asexual reproduction, which is not specified.

p.33 They are correct that the MVB type model can be used to represent multi-locus linkage disequilibrium, but the use of log-linear models for this has a long history (e.g. Smouse 1974 Genetics 76: 557), so there is nothing especially new here.

(Remarks on code availability)

Reviewer #3

(Remarks to the Author)

My comments on the noteworthiness and potential significance of the paper are largely the same as those of the reviewers in the previous round. The questions raised by the previous reviewers have been addressed convincingly by the authors. The work is sound and has been made broadly accessible through a number of specific applications and illustrations.

I have a few minor comments.

1. While reading section 2.4.1, I was initially confused by the fact there is discordance between the multiplicative epistasis measure and the β_S (as seen in Figure 2, for example), since Theorem 1 (Methods) shows that the two are identical. I had to dig into Methods to understand that the "noise" term is actually there *in addition* to the interaction terms, and is not considered as a source of "true" interaction. (This may be especially confusing to those who are familiar with the widely-used Rough Mount Fuji model, where the noise term is the only source of epistasis.) This also explained why there is a discordance for 2-way epistasis in the first subfigure of Figure 2A. It would be good if the authors clarified this in section 2.4.1.

2. I would suggest that the authors mention in the Abstract their finding that the chimeric formula falsely detects higher order epistasis when there is none (Section 2.4.2). This is a clear and striking demonstration and has greatly improved the manuscript.

3. The expansion in Eq. (26) is identical to the so-called "Taylor" series expansion of fitness landscapes, see for example (Weinberger, "Fourier and Taylor series on fitness landscapes." Biological cybernetics, 1991). The connection of the multiplicative measure to the Taylor expansion is therefore more direct than to the Fourier-Walsh decomposition.

4. On page 33, authors say that "Theorem 2 also provides a connection between the multivariate Bernoulli distribution and the circuit formulae... as the circuit formulae for a full genotype space are linear combinations of the Walsh coefficients;". This connection is rather circuitous (please forgive the pun) and adds nothing substantive.

5. Typo on page 3: "there is up to substantial (up to 60%)".

6. On pages 6-7, the authors say that "prior literature on higher-order interactions does not rigorously define the chimeric epistasis measure...", and specifically point to Refs [2,42,43]. The complaint that there is lack of "rigor" is a bit uncharitable since the definition of chimeric epistasis has been clearly provided in the references, and the use of cumulants in [43] has been done with sufficient background to make the idea clear. Perhaps the most that can be said that the connections were made incompletely.

7. Typo near the bottom of page 29: I believe " $E[X_{12}]$ " should be " $E[X_1 X_2]$ ".

(Remarks on code availability)

Version 1:

Reviewer comments:

Reviewer #1

(Remarks to the Author)

(Remarks on code availability)

Reviewer #2

(Remarks to the Author)

The authors have responded conscientiously to my comments. I apologise for the delay in this review, but I have had a heavy load of similar work.

The paper is suitable for publication with a few minor corrections. It is a useful clarification of the issues surrounding the measurement of fitness in microbial experimental populations.

Introduction

I.3 "fitness effects of alleles"

p.2 §2 I.1 "notably" seems redundant.

§6 I.1 "leverage" is a horrible piece of corporate-speak; it just mean "use".

References

The use of full author names seems unusual in the reference list.

(Remarks on code availability)

Reviewer #3

(Remarks to the Author)

The authors have addressed all my comments satisfactorily.

(Remarks on code availability)

Reviewer response for manuscript NG-A62915

Uthsav Chitra*, Brian J. Arnold*, and Benjamin J. Raphael#
Department of Computer Science, Princeton University, Princeton, NJ 08540

* Equal contribution

Correspondence: braphael@princeton.edu

We thank the reviewers for their thorough and helpful comments. Below we include the reviewer text (black) and our response (blue text). In the revised manuscript, we have highlighted all major text changes in blue.

For all reviewers: we added a new analysis on **measuring epistasis between protein mutations using deep mutational scanning (DMS) data of 9 different proteins**. We show that the chimeric and multiplicative measures disagree substantially for several proteins including the *folA* protein (recently profiled by Papkou et al., *Science* 2023) and the SpCas9 genome editing protein. See Section 2.7 of manuscript and Figure 6 (below).

Figure 6: Comparison of multiplicative and chimeric measures for measuring epistasis between protein mutations in nine different proteins. (A-B) Standard deviation of fitness values across all (left) three-, (middle) four-, and (right) five-way tuples of mutations versus the average (A) correlation and (B) sign disagreement fraction of the log-multiplicative measure $\log e^M$ versus the chimeric measure e^C . (C-D) Log-multiplicative measure $\log e^M$ versus chimeric measure e^C for the (C) *folA* [111] and (D) *Streptococcus pyogenes* Cas9 (SpCas9) nuclease [112] proteins.

Reviewer #1:

Remarks to the Author:

A) The manuscript treats two different topics. The first topic concerns a formula for epistasis that has been used in publications. The authors call it "the chimeric formula". The authors argue that the formula is flawed and should not be used at all. The second topic is an attempt to relate the multivariate Bernoulli distribution to various epistasis concepts from the literature, including multiplicative and additive epistasis (defined as conventional), Walsh coefficients, Markov bases and circuits.

B-E) The first topic (regarding the chimeric formula) is well argued with strong empirical evidence for that the formula can be misleading.

We thank the reviewer for their positive feedback on a major component of our manuscript.

The second topic is rather difficult to evaluate because of a poor presentation with missing information, typos in mathematical formulas (already on Page 2), and incorrect claims.

We corrected the typo in the multiplicative formula on Page 2. Furthermore, we have removed the incorrect information on the circuits, as we describe below.

For example, the main text refers to information about circuits in Table 1, but Table 1 does not include circuits. Definitions are incomplete or missing (instead there are references to publications) throughout the manuscript.

A connection between the multivariate Bernoulli distribution and circuits would be especially interesting. However, this is what I found when I tried to read the manuscript text on circuits:

1) The manuscript gives an incorrect definition of circuits.

2) The manuscript gives an example of a "circuit", which is in fact not a circuit, but agrees with the authors' incorrect definition.

3) The manuscript makes the following claim

"[47] claims that Markov basis and circuits are able to capture certain kinds of epistasis that the additive epistasis measures "epsilon-A" cannot capture" The comment sounds almost sarcastic. However, it is easy to verify that circuits capture other gene interactions (see also below).

4) The manuscript makes the following claim: "the linear forms of the circuits and Markov bases are hard to interpret biologically, as even [47] notes". However, the main topic of the publication [47] is exactly to explain biological interpretations of a selected set of circuits, including what the authors call double-double tests (a test for curvature on greater distances than for the standard epistasis concept).

F) The manuscript needs to be much easier to read. I strongly recommend including all important definitions (rather than referring to other papers). The text on Markov bases could be deleted in my view, if the authors can produce a correct text on circuits. The exact connection between the multivariate Bernoulli distribution and circuits should be explicitly stated and proved.

(Alternatively, if the authors do not have such a result, there is no reason to discuss circuits).

I hope to see an improved version of the manuscript at some point, the project is clearly interesting.

We thank the reviewer for the careful reading of our manuscript, particularly on the connection to circuits. After carefully re-reading Beerenwinkel et al. ([26], [47]), we agree with the reviewer that we provide an inaccurate definition of circuits in Section 4.4.4 and we understate their significance.

Unfortunately, we do not have a general theory relating the circuits and the multivariate Bernoulli distribution. The most we can state is that our result for Walsh coefficients (Theorem 2) is somewhat related to the circuit formulas, as the circuits for a full genotype space are a linear combination of the Walsh coefficients (Beerenwinkel et al., 2007; Crona et al., 2017).

Thus, we have made the following changes to the manuscript to correct our description of circuits and address the reviewer's comments:

- We updated the title to remove the word “unified”, reflecting that our theory does not describe the circuits formulas.
- We removed the reference to the circuit formulas in the Introduction and the results section.
- We removed Section 4.4.4 of the Methods (Markov bases and circuits), as the reviewer suggested.
- We added the following sentence to the Methods section on Walsh coefficients and background-averaged epistasis:

Methods (Section 4.4.3): *“Theorem 2 also provides a connection between the multivariate Bernoulli distribution and the circuit formulas used to quantify the geometry of a fitness landscape [59, 29, 60, 69], as the circuit formulas for a full genotype space are linear combinations of the Walsh coefficients; see [29] for details.”*

- We added the following sentence to the Discussion section:

Discussion: *“Fourth, it would be quite interesting to investigate whether there exist deeper connections between the MVB and the circuit formulas used to quantify the shape of a fitness landscape [59, 29, 60, 69].”*

Finally, **we updated the title of the manuscript** to remove the word “unified”, since our model does not address the circuit formulas, as noted by the reviewer.

VERY SHORT ABOUT CIRCUITS FOR THE AUTHORS (not really part of the review):

If the null is additive fitness, then the circuits measure deviations from additivity. For instance, this is a circuit:

$$c=w_{111}-w_{101}-w_{011}+w_{001}.$$

Notice that $c=0$ if fitness is additive (all circuits have that property!). If $c>0$ we can conclude that mutations at the two left loci have a positive interaction provided that the rightmost locus is mutated. Such phenomena are not rare in nature, an interaction can be conditionally positive. (It could be the case that

$$d=w_{110}-w_{100}-w_{010}+w_{000} \text{ is zero even if } c>0).$$

Here is another circuit: $c'=2w_{111}+w_{000}-w_{110}-w_{101}-w_{011}$

Again $c'=0$ if fitness is additive. If $c'>0$ then the triple mutant has higher fitness than one would expect from looking at the wild-type and double mutants only.

The complete set of circuits for three loci consists of 20 elements. They describe ALL INTERACTIONS (other interactions are redundant in the sense that they can be described in terms of the 20 circuits).

How do we know that circuits are meaningful? By using circuits one can describe exactly under what circumstances genetic recombination (gene-shuffling) is unable to increase fitness. That's a very precise observation that holds for any number of loci. Such a population is called a "fittest population" (see Beerwinkel et al. 2007 for more background and proofs).

We thank the reviewer for the informative introduction to the circuit formulas and their relationship to the fittest populations of a fitness landscape.

Reviewer #2:

Remarks to the Author:

The interaction between genetic elements, both within and between genes is a topic of increasing importance in the study of molecular function and human disease. The authors investigate a critical element of this work: how do best measure these interactions (epistasis) in the first place. As they note there are several different ways of doing this, and they propose a “best” way, show how it is universally applicable, and then illustrate its use using a large yeast dataset. Overall, the paper makes some very important points and is a strong contribution to the literature.

We thank the reviewer for their positive feedback on the strength of our contributions.

It is fairly technical in nature and would not be a typical paper for Nature Genetics. It may be best suited for a journal that regularly has mathematical proofs as part of core results.

We agree that our original manuscript was quite technical. To make our manuscript more accessible to a broad audience, we have: (1) moved the mathematical details to the Methods/Appendix, and (2) added two new simulation analyses that empirically demonstrate the benefits of our model (see details below).

Furthermore, to improve the applicability of our manuscript, we have added a new analysis on measuring epistasis between protein mutations using deep mutational scanning (DMS) data. See Section 2.7 in the manuscript and the new Figure 6 at the top of this document.

Additionally, the authors are very free with their criticisms of epistasis measures, but do not fully touch on the potential utility of different measures. That is, epistasis has been measured in the way that it is because it is important for understanding fundamental biological processes. Because of the different contexts in which these measures have been applied, a bit more attention to this work and whether conclusions contained in this work would be different with different measures, is warranted.

1. The primary use of quantitative measures of epistasis have been in evolutionary genetics. Indeed, this paper focuses specifically on fitness, which is appropriate, as RA Fisher long ago made the point that other traits could be transformed to a variety of scales, many of which would have no epistasis, but regardless, an additive scale is fine for most purposes here and has many advantages. But fitness has only one scale. However, the authors need to be clear that the proper definition of fitness is the measure that accurately predicts allele frequency change in the next generation. That is usually a measure of population growth rate, which in turn makes measure of reproduction and/or cell division the right phenotypic measure – but not always.

We agree with the important distinction raised by the reviewer between (1) fitness, or the mean reproductive success of a genotype in the next generation, and (2) the phenotypic traits (e.g. growth rate) used in practice to estimate fitness and compute epistasis. In particular, the scale

of the true fitness values may not agree with the scale of the measured phenotypic traits that are used as a proxy for fitness.

In order to make this distinction clearer, we added the following text to the second paragraph of the Results section:

Results (Section 2.1): “In practice, the fitness f of a genotype, i.e. the mean reproductive success of the genotype, cannot be directly measured. Instead, experiments typically measure traits that are expected to be highly correlated with fitness, e.g. cellular reproductive or growth rate, and are assumed to follow either an additive or multiplicative scale [58]. Accordingly, the two standard null models of fitness for measuring epistasis are the additive model and the multiplicative model.”

Epistasis matters in evolutionary genetics for two primary reasons. First, epistatic interactions can change the adaptive landscape such that some evolutionary paths are not possible. The authors go to great effort to show that their measure of epistasis leads to different measures/predictions of higher order epistasis in their empirical examples. But just because it is different does not mean that it is correct for its purpose (although I suspect that it is). The authors should show that their measure predicts the correlation between sign epistasis and adaptive valleys in their multi-allelic models before that can assert that different is better.

We thank the reviewer for these helpful suggestions. We agree that it is necessary to empirically demonstrate that the multiplicative epistasis measure more accurately describes the adaptive fitness landscape, including adaptive valleys and peaks.

Thus, we have performed a new **simulation analysis** where we have **ground truth for the epistatic interactions**. Specifically, we simulate fitness values f according to a multiplicative fitness model with multiplicative Gaussian noise $N(0, \sigma^2)$ and with interaction parameters β for different maximum interaction orders. We independently draw the interaction parameters $\beta \sim Uni(-0.5, 0.5)$. The interaction parameters β describe the epistatic interactions, i.e. the adaptive peaks and valleys of the fitness landscape.

We used a multiplicative fitness model since (1) both the multiplicative measure and the chimeric measure assume a multiplicative fitness model (e.g. Kuzmin et al., *Science* 2018, 2020) and (2) fitness measures such as cellular growth rate and reproduction are often modeled multiplicatively (e.g. Costanzo et al., *Science* 2016, or Kuzmin et al., *Science* 2018).

Across all noise parameters and interaction orders >2 , we show that the multiplicative epistasis measure ϵ^M better predicts the *sign* (first row, Figure 2 below) of the true epistasis parameters β compared to the chimeric measure ϵ^C . Furthermore, for all interaction orders – even pairwise interactions – the multiplicative measure ϵ^M better estimates the *magnitude* $|\beta|$ of interaction parameters across all noise parameters (second row, Figure 2 below). These simulations demonstrate that the multiplicative epistasis measure ϵ^M yields a different and **more accurate**

measurement of both pairwise and higher-order epistasis compared to the chimeric measure. See Section 2.4.1 and Figure 2 below.

Figure 2: Fitness values f are simulated following a multiplicative fitness model with interaction parameters β , for different choices of the maximum interaction order K , and multiplicative Gaussian noise with standard deviation σ . (A) The fraction of K -way interactions where the sign of the log-multiplicative epistasis measure $\log \epsilon^M$ (orange) and the chimeric epistasis measure ϵ^C (blue) do not match the sign of the true interaction parameter β . (B) The average absolute difference (“error”) $|\beta - \log \epsilon^M|$ and $|\beta - \epsilon^C|$ between the true interaction parameter β and (orange) the log-multiplicative measure $\log \epsilon^M$ and (blue) the chimeric measure ϵ^C , respectively. These quantities are plot for different values of the maximum interaction order K and noise parameter σ and are averaged across 100 simulated fitness values.

2. The second most important use of epistasis in evolutionary genetics is that there is a fundamental relationship between epistasis and recombination in determining evolutionary outcomes. The chimeric model that the authors criticize has been used extensively in this context. While the authors cite papers that use this measure, they do not seem to appreciate why this measure is useful in this context. Nor do they address whether their measure is an adequate substitute for this use. This is literally sixty years of work, so it should not be flippantly brushed aside. I suggest that the authors look specifically at the work of Sarah Otto on positive and negative epistasis and its relationship to the evolution of sex and recombination. If the purely multiplicative measure is not appropriate for evolutionary genetics, this this paper will simply be one more addition to the confusion in the literature that constantly crops up when someone feels that measures that have existed for 100 years don’t work for them.

We thank the reviewer for pointing us to this line of work in theoretical population genetics on the relationship between epistasis and the evolutionary origins of recombination (e.g. Eshel and Feldman 1970; Barton 1995; Otto and Feldman 1997; Kouyos, Otto, and Bonhoeffer 2006).

We emphasize that the central results of this literature only depend on the **sign** $sgn(\epsilon^C)$ of the pairwise chimeric measure ϵ^C . Specifically, several studies (e.g. Barton 1995) show that under specific mathematical models, recombination is beneficial when there is *negative* chimeric

epistasis, i.e. $\epsilon^C < 0$. Since the sign $\text{sgn}(\epsilon^C)$ of the chimeric measure is equal to the sign $\text{sgn}(\log \epsilon^M)$ of the multiplicative measure – as we show in Proposition 1 – **the main results of these studies are unchanged if one uses the pairwise *multiplicative* measure ϵ^M rather than the pairwise chimeric measure**, as these results only depend on the epistasis *sign*.

That said, we note that the exact formulas that relate epistasis, selection, and recombination will change if one defines epistasis using the multiplicative measure ϵ^M rather than the chimeric measure.

We updated the manuscript in several places (below) to discuss the connection between recombination and the pairwise epistasis measures.

Introduction: *“the sign of an epistatic interaction is often the quantity of interest in genetics studies, e.g. negative epistatic interactions are used to quantify functional redundancy [2, 35, 36] and recombination [37, 38, 39, 40].”*

Results (Section 2.1): *“Thus, using either the chimeric or multiplicative measures will not affect results that depend on the sign of an epistatic interaction, such as the relationship between increased recombination and negative epistasis [37, 40].”*

Discussion: *“As another example, the formulae derived in the theoretical population genetics literature [38, 39, 40] that relate recombination, selection, and the pairwise chimeric epistasis will change if pairwise epistasis is instead measured using the multiplicative measure.”*

That being said, I do wonder if the chimeric measure might be a first order approximation to the multiplicative measure via a Taylor expansion or something similar. If so, then the chimeric measure may be a useful approximation around small deviation, while the multiplicative measure might be a more general solution. I don't know, but either the authors need to explore this or they need to tone down the dismissive nature of their narrative, which does not reflect a nuanced understanding of the issues at hand.

We thank the reviewer for this suggestion. After digging in further, we derived the following result: for pairwise (2-way) interactions, the chimeric measure approximates the log-multiplicative measure when the double-mutant and single-mutant fitness values are close to 1. Specifically, if $f_{ij} \approx 1$, $f_i f_j \approx 1$, then $\epsilon_{ij}^C \approx \log \epsilon_{ij}^M$. We show this approximation using the Taylor approximation $\log c \approx c - 1$ for $c \approx 1$ (Appendix G).

However, this approximation does **not** hold when the fitness values are not close to 1. We demonstrate this empirically using our simulations (see response to point 1), where we show that the absolute difference $|\epsilon_{ij}^C - \log \epsilon_{ij}^M|$ between the chimeric measure ϵ_{ij}^C and the

log-multiplicative measure $\log \epsilon^M$ is large when either f_{ij} , f_i , or f_j are far from 1. See Figure S2 from the revised manuscript below.

Figure S2: Double mutant fitness f_{ij} versus product $f_i f_j$ of single mutant fitness values for each pair (i, j) of loci across all simulated instances of the multiplicative fitness model in Section 2.4 (pairwise interactions with noise parameter $\sigma = 0$). Points are colored by the difference $|\epsilon_{ij}^C - \beta|$ between the chimeric epistasis measure ϵ_{ij}^C and true interaction parameter β .

Moreover, for *higher-order* interactions (>2-way), the chimeric measure does **not** approximate the log-multiplicative measure using a Taylor series expansion. In particular, the chimeric measure **does not even have the same sign** as the log-multiplicative measure. We empirically demonstrate this with our simulations (point 1 response) in Figure S3 below and Appendix G, where we observe that the Pearson correlation $\rho(\epsilon_{ij}^C, \log \epsilon^M_{ij})$ between the log-multiplicative measure $\log \epsilon^M_{ij}$ and the chimeric measure ϵ^C_{ij} decreases with the interaction order. (Note that we restrict the chimeric measure axis as it is skewed by some outliers with large chimeric measure value)

Figure S3: Pearson correlation $\rho(\epsilon^C, \log \epsilon^M)$ between chimeric measure ϵ^C and multiplicative measure ϵ^M for different interaction orders K and each K -tuple of loci across all simulated instances of the multiplicative fitness model in Section 2.4 (pairwise interactions with noise parameter $\sigma = 0$). The x - and y -axis ranges are $[\min \epsilon^M, \max \epsilon^M]$.

The empirical finding that the chimeric and multiplicative measures are not highly correlated for large interaction orders agrees with the central message of our manuscript, which is that the chimeric formula inaccurately measures *higher-order* interactions.

Nevertheless, we did not previously appreciate that the *pairwise* chimeric measure is approximately equal to the pairwise log-multiplicative measure for fitness values close to 1, as the reviewer suggested. To address this point, we added the following sentence to Results above Proposition 1:

Results Section 2.1: “We note that when the double-mutant fitness f_{ij} and single-mutant fitness values f_i, f_j are close to 1, the chimeric measure ϵ_{ij}^C is approximately equal to the log-multiplicative measure $\log \epsilon_{ij}^M$ (Supplementary Text).”

Moreover, in the Appendix, we added Figures S2 and S3 (above) and the following text which derives the Taylor approximation between the pairwise chimeric measure and log-multiplicative measure.

G Pairwise and higher-order epistasis measure comparison

The pairwise chimeric measure ϵ_{ij}^C approximates the pairwise multiplicative epistasis measure ϵ_{ij}^M under certain conditions. Specifically, if the double mutant fitness f_{ij} and the product $f_i f_j$ of the single-mutant fitness values are both close to 1, i.e. $f_{ij} \approx 1$ and $f_i f_j \approx 1$, then the pairwise chimeric epistasis measure ϵ_{ij}^C is approximately equal to the pairwise log-multiplicative measure $\log \epsilon_{ij}^M$. To see this, note that

$$\log \epsilon_{ij}^M = \log f_{ij} - \log f_i f_j \approx (f_{ij} - 1) - (f_i f_j - 1) = f_{ij} - f_i f_j = \epsilon_{ij}^C, \quad (55)$$

where we use the approximation that $\log c \approx c - 1$ if $c \approx 1$.

We empirically assessed (Figure S2) how closely the pairwise chimeric epistasis measure ϵ_{ij}^C approximates the interaction parameter β_{ij} (which is equal to the pairwise log-multiplicative measure $\log \epsilon_{ij}^M$), using the simulated fitness values \mathbf{f} from the multiplicative fitness model (Section 2.4, pairwise interactions $K = 2$ with noise parameter $\sigma = 0$). We observe that the chimeric measure has small error $|\epsilon_{ij}^C - \beta_{ij}|$ when both the double mutant fitness f_{ij} and product $f_i f_j$ of single mutant fitness values are close to 1. However, the error gets much larger when either f_{ij} or $f_i f_j$ are not close to 1, which agrees with when the approximation in (55) is valid. Moreover, the multiplicative and chimeric measures diverge substantially for higher-order interactions, with the correlation between the two measures approaching zero as the interaction order increases (Figure S3).

This analysis demonstrates that for pairwise interactions, the chimeric epistasis measure ϵ_{ij}^C is an accurate measure of interactions in a multiplicative fitness model in certain settings.

Finally, we updated the title of the manuscript to better emphasize the focus on higher order epistasis (where the multiplicative and chimeric measures differ in sign) and to be less dismissive of the multiple use-cases for the different epistasis formulas, as noted by the reviewer.

3. The third major use of epistasis is in the context of understanding the structure of genetic networks. This would seem to be the emphasis of the yeast analysis presented in the paper. This is a very difficult issue to be objective about in terms of which measure makes the right functional prediction. Similar to the adaptive peaks question, this would be best done in a graph-theoretic context by seeing which measures actually best reflect the functional structure of a genetic network. What does 4th order epistasis mean and/or predict in this context. Without a suitable null model or positive control to compare to, all such analyses are subjective narratives rather rigorous demonstrations. Again, the latter has great value, but the presentation in the paper suggests that something has been shown to be objectively useful when it is difficult to know that that is the case for certain.

We agree with the reviewer on the importance of an objective assessment of the different epistasis measures that incorporates the structure of a genetic network. Thus, in addition to the simulations described above (point 1), we performed a **second simulation analysis** using fitness values derived from a **synthetic genetic interaction network**.

Specifically, we simulate fitness values using the classical NK model developed by Stuart Kauffman and others in the 1980s. The NK model is a standard model for simulating random fitness landscapes (>700 citations) and is used in many analyses (e.g. as described by de Visser and Krug, *Nat. Rev. Genet.* 2014).

The NK model is parametrized by (1) the number N of loci, which we call L in the text and in below, and (2) a “ruggedness” parameter K , which describes how smooth/rugged the fitness landscape is, with $K=0$ being smooth (only a single adaptive peak) and $K=L-1$ being the most rugged (many adaptive peaks/valleys). Each loci interacts with K other loci, so that the NK model describes at most $(K+1)$ -way interactions.

Importantly, the NK model also simulates a **genetic interaction network** G , where two loci are connected by an edge if they have a pairwise or higher-order interaction. (For $K>1$, i.e. higher-order interactions, the NK model also simulates a genetic interaction *hypergraph*, where hyperedges connect groups of interacting loci, e.g. genetic pathways.)

For $K=1$, there are only pairwise (2-way) interactions in the NK model, and there are no higher-order interactions. The absence of higher-order interactions is reflected by the 3-way multiplicative measure ϵ^M , which equals zero for all triples (i,j,k) , i.e. $\epsilon^M_{ijk} = 0$. However, if there is a triangle (i,j,k) in the pairwise interaction network G , then the chimeric epistasis measure is non-zero, i.e. $\epsilon^C_{ijk} \neq 0$ (with probability 1). Thus, the chimeric measure will infer a spurious 3-way interaction that is not present in the fitness landscape. See Figure 3A below.

(As this is sometimes a point of confusion: we note that triangles in a graph are sometimes referred to as higher-order structures, e.g. Benson et al., *Science* 2016. However, as our NK simulation demonstrates, it is quite possible to have a triangle in a graph, i.e. three pairwise interactions, without having a genuine higher-order (3-way) interaction.)

More generally, for ruggedness parameter K , there are only $(K+1)$ -way interactions in the NK model. While the $(K+2)$ -way multiplicative measure is always zero, i.e. $\epsilon^M = 0$, the $(K+2)$ -way chimeric measure will often be non-zero, i.e. $\epsilon^C \neq 0$, and thus infer spurious higher-order interactions. In this way, the chimeric measure will incorrectly infer that the fitness landscape is more rugged (contains more adaptive peaks/valleys) than it actually is. See Figure 3B below.

We added the results of these simulations to Section 2.4.2 and Figure 3 (shown below).

Figure 3: **(A)** A fitness landscape f simulated following the NK fitness model with “ruggedness” parameter $K = 1$ contains only pairwise interactions. These interactions are represented with an interaction graph G . The 3-way multiplicative measure $\epsilon_{ijk}^M = 0$ equals zero for all loci triples (i, j, k) . However, if the triple (i, j, k) forms a triangle in the graph G , then the 3-way chimeric epistasis measure ϵ_{ijk}^C is non-zero, and incorrectly indicates the presence of a higher-order interaction. **(B)** The fraction of non-zero $(K + 2)$ -way interactions (“higher-order interactions”) identified by the multiplicative measure ϵ^M (orange) and the chimeric measure ϵ^C (blue) across 100 fitness landscapes f simulated according to the NK fitness model with ruggedness parameter K . The fitness landscape f contains at most $(K + 1)$ -way interactions, but the chimeric measure ϵ^C spuriously detects many non-zero $(K + 2)$ -way interactions.

4. A great strength of the paper is that the epistasis measure is put into a statistical framework using the multivariate Bernoulli distribution. While a very good way to go, using a set theory approach of this type is not unique in the field for these measures (and much more). I suggest that the authors look at the work by Nick Barton and Michael Turelli (e.g., <https://doi.org/10.1093/genetics/127.1.229> and many following). It would be useful to see how much of this ground has been covered by them in the more general context and how much is specific to this application to epistasis.

We thank the reviewer for pointing us to this line of work. Similar to our manuscript, the works by Barton and Turelli (e.g. B & T 1991, B & T 2004, T & B 2006) model the binary genotype (X_1, \dots, X_n) as a random variable. In particular, Barton and Turelli implicitly model the genotype distribution $p(X_1, \dots, X_n)$ as a *multivariate Bernoulli* distribution, although they do not explicitly use this terminology.

Nevertheless, Barton and Turelli’s approach is fundamentally different from ours. We model *epistasis* using the parameters of the multivariate Bernoulli distribution $p(X_1, \dots, X_n)$. In contrast, Barton and Turelli model *linkage disequilibrium* using the parameters of the distribution $p(X_1, \dots, X_n)$, and they model epistasis using coefficients that are **not related** to the distribution $p(X_1, \dots, X_n)$.

- Interestingly, Barton and Turelli’s 3-way linkage disequilibrium formula is equal to the 3-way cumulant formula used by Kuzmin et al. This is because Barton and Turelli define k -way linkage disequilibrium as the k -th order central moment (e.g. the 3rd order central

moment of X, Y, Z is $E[(X-E[X])(Y-E[Y])(Z-E[Z])]$, which is equal to the k -th order joint cumulant used in Kuzmin et al. for $k=1,2,3$. However, for $k \geq 4$, the Barton and Turelli's k -way linkage disequilibrium formula is not equal to the k -way joint cumulant formula.

We added the following text to Methods (Section 4.4.4) discussing this line of work:

4.4.4 Relationship to theoretical genetics models

We note that some previous works in theoretical genetics by Barton and Turelli (e.g. [134, 135, 136]) also model the genotype with a MVB. However, their approach is substantially different from ours. Barton and Turelli model linkage disequilibrium between k loci X_{i_1}, \dots, X_{i_k} using the k -way central moment $\tilde{\mu}(X_{i_1}, \dots, X_{i_k}) = E \left[\prod_{j=1}^k (X_{i_j} - E[X_{i_j}]) \right]$ of the genotype distribution $P(X_1, \dots, X_n)$. Barton and Turelli model epistasis using coefficients that are not related to the genotype distribution $P(X_1, \dots, X_n)$. In contrast, we model epistasis with the natural parameters β of the genotype distribution $P(X_1, \dots, X_n)$, as described in Section 4.4.1.

Interestingly, Barton and Turelli's 3-way linkage disequilibrium term, i.e. $\tilde{\mu}(X_{i_1}, X_{i_2}, X_{i_3})$, is equal to the 3-way joint cumulant $\kappa(X_{i_1}, X_{i_2}, X_{i_3})$ implicitly used by Kuzmin et al. [2, 44] to measure 3-way epistasis (see Section 4.4.2). This equivalence is because the k -way central moment $\tilde{\mu}(X_{i_1}, \dots, X_{i_k})$ is equal to the k -way joint cumulant $\kappa(X_{i_1}, \dots, X_{i_k})$ for $k = 1, 2, 3$. However, for $k \geq 4$, the k -way linkage disequilibrium term used by Barton and Turelli is not equal to the k -way joint cumulant.

Reviewer #3:

Remarks to the Author:

This theoretical and technical paper is about the analysis of higher-order interactions in the context of epistasis and fitness, in particular for studies in model organisms where multiple genes can be knocked out simultaneously. The authors claim that the usual way in which such experiments are analysed is not optimal and they propose that data should be analysed using either additive or multiplicative methods. They provide new theory that reconciles previous approaches. From re-analysis of published data, the authors show that the sign of higher-order interactions can change and claim that such sign changes are important in the interpretation of the interactions.

Interactions are per definition scale dependent, so presumably the correct model to fit is one that best reflects the distribution of effect sizes on absolute or relative fitness. If relative fitness (RF) effects are small, so that $RF_i = 1 + f_i$, then it would seem that $\log(f_i)$ model is essentially the same as the chimeric model because $\log(RF_i) \sim f_i$. Is that the reason why researchers have used the chimeric model in the past?

This is a good observation. After looking into this, we found that for pairwise (2-way) interactions, if the fitness values f_i, f_j, f_{ij} are close to 1, then the chimeric measure ϵ_{ij}^C is approximately equal to the (log-)multiplicative measure $\log \epsilon_{ij}^M$. This may be why researchers have used the pairwise chimeric measure in the past, as the reviewer points out.

However, the approximation does not hold when the fitness values f_i, f_j, f_{ij} are not close to 1, as we observe visually from simulations (see Figure S2 below and response to R2, point 2).

Figure S2: Double mutant fitness f_{ij} versus product $f_i f_j$ of single mutant fitness values for each pair (i, j) of loci across all simulated instances of the multiplicative fitness model in Section 2.4 (pairwise interactions with noise parameter $\sigma = 0$). Points are colored by the difference $|\epsilon_{ij}^C - \beta|$ between the chimeric epistasis measure ϵ_{ij}^C and true interaction parameter β .

Moreover, for *higher-order* interactions (>2-way), there is no approximation between the chimeric and multiplicative measures, with the Pearson correlation between the two measures decreasing with the interaction order (see Figure S3, shown below). In particular, the chimeric measure **does not even have the same sign** as the log-multiplicative measure.

Figure S3: Pearson correlation $\rho(\epsilon^C, \log \epsilon^M)$ between chimeric measure ϵ^C and multiplicative measure ϵ^M for different interaction orders K and each K -tuple of loci across all simulated instances of the multiplicative fitness model in Section 2.4 (pairwise interactions with noise parameter $\sigma = 0$). The x - and y -axis ranges are $[\min \epsilon^M, \max \epsilon^M]$.

Nevertheless, we did not previously appreciate that under certain conditions, the *pairwise* chimeric measure is approximately equal to the pairwise log-multiplicative measure, as the reviewer pointed out. We added the following sentence to the manuscript to describe this point.

Results Section 2.1: “We note that when the double-mutant fitness f_{ij} and single-mutant fitness values f_i, f_j are close to 1, the chimeric measure ϵ_{ij}^C is approximately equal to the log-multiplicative measure $\log \epsilon_{ij}^M$ (Supplementary Text).”

Moreover, in the Appendix, we added Figures S2 and S3 (shown above) and the following text which derives the Taylor approximation between the pairwise chimeric measure and log-multiplicative measure.

G Pairwise and higher-order epistasis measure comparison

The pairwise chimeric measure ϵ_{ij}^C approximates the pairwise multiplicative epistasis measure ϵ_{ij}^M under certain conditions. Specifically, if the double mutant fitness f_{ij} and the product $f_i f_j$ of the single-mutant fitness values are both close to 1, i.e. $f_{ij} \approx 1$ and $f_i f_j \approx 1$, then the pairwise chimeric epistasis measure ϵ_{ij}^C is approximately equal to the pairwise log-multiplicative measure $\log \epsilon_{ij}^M$. To see this, note that

$$\log \epsilon_{ij}^M = \log f_{ij} - \log f_i f_j \approx (f_{ij} - 1) - (f_i f_j - 1) = f_{ij} - f_i f_j = \epsilon_{ij}^C, \quad (55)$$

where we use the approximation that $\log c \approx c - 1$ if $c \approx 1$.

We empirically assessed (Figure S2) how closely the pairwise chimeric epistasis measure ϵ_{ij}^C approximates the interaction parameter β_{ij} (which is equal to the pairwise log-multiplicative measure $\log \epsilon_{ij}^M$), using the simulated fitness values \mathbf{f} from the multiplicative fitness model (Section 2.4, pairwise interactions $K = 2$ with noise parameter $\sigma = 0$). We observe that the chimeric measure has small error $|\epsilon_{ij}^C - \beta_{ij}|$ when both the double mutant fitness f_{ij} and product $f_i f_j$ of single mutant fitness values are close to 1. However, the error gets much larger when either f_{ij} or $f_i f_j$ are not close to 1, which agrees with when the approximation in (55) is valid. Moreover, the multiplicative and chimeric measures diverge substantially for higher-order interactions, with the correlation between the two measures approaching zero as the interaction order increases (Figure S3).

This analysis demonstrates that for pairwise interactions, the chimeric epistasis measure ϵ_{ij}^C is an accurate measure of interactions in a multiplicative fitness model in certain settings.

The authors propose that either additive or multiplicative models should be used and re-analyse data for each, using yeast and E-coli experimental data, respectively. In these examples, the authors claim that the choice of additive or multiplicative model is natural. Can the authors comment on whether the choice of which model to fit for other experiments is always obvious?

This is an excellent question. When fitness is measured experimentally, there is often a clear choice for which model — and thus which epistasis measure — to use based on the experimental technology and the scale used to measure fitness values. For example, many genetics experiments (including the yeast experiments we analyze) use a synthetic genetic array (SGA, Tong et al. *Science* 2001) technology which models fitness values as a multiplicative function of colony size and other covariates, thus indicating a multiplicative fitness model/multiplicative epistasis measure should be used to measure epistasis.

Still, there may be some situations where the choice of fitness model is not obvious. There are some papers solely dedicated to studying which epistasis formulas best fit experimental data (e.g. Mani et al, PNAS 2008). One benefit of our work is that our statistical MVB framework provides a way to do formal model comparisons. Thus, an interesting and important future direction would be to use the MVB to derive a statistical test for whether the additive or multiplicative model better fits a given dataset.

We added the following text to the Discussion describing this future direction.

Discussion: “Fifth, for certain technologies, it may be desirable to test whether an additive or multiplicative model better fits experimental fitness data. Our MVB framework provides an approach for doing formal model comparisons. Thus, an interesting and important future

direction would be to derive a statistical test using the MVB to test which fitness model better fits data.”

We also emphasize that in our **new analysis on epistasis between protein mutations** (Section 2.7 in the manuscript), each of the 9 deep mutational scanning (DMS) datasets that we analyzed clearly states whether fitness was measured additively or multiplicatively.

Despite statements in the paper and citations to published papers, there is no evidence that higher-order epistasis is relevant for the analysis or interpretation of GWAS for complex traits in outbred populations (such as humans). The papers that are cited are all review, perspective or advocacy papers and not papers that report primary discoveries on epistasis in humans. Higher-order interactions are not expected to explain individual differences in the population for complex traits (<https://pubmed.ncbi.nlm.nih.gov/24990992/>) and to my knowledge there are no humans with homozygous loss-of-variant mutations in multiple genes.

We believe that the reviewer’s claim that *“[h]igher-order interactions are not expected to explain individual differences in the population for complex traits”* is too strong. There are **many** differing opinions on the importance of complex trait epistasis. For example, while the paper the reviewer linked states that epistasis is not a likely source of missing heritability, Zuk et al. (PNAS 2012) argue that *“the failure to detect epistasis does not rule out the presence of genetic interactions sufficient to cause substantial phantom heritability.”* Moreover, Huang and Mackay (PLoS Genet 2016) wrote an entire paper describing a major logical flaw in the paper linked by the reviewer, namely that variance component analysis (which Maki-Tanila and Hill use) does not necessarily reveal the architecture of complex traits.

Moreover, several recent papers have identified significant complex trait epistasis in humans, including epistatic interactions between pathways (Sheppard et al. PNAS 2021; Tang et al., AJHG 2023) and individual SNPs (Fu et al., Nat Comms 2023; Fu et al., bioRxiv 2023). Tang et al. demonstrate that, compared to purely additive models, structured models of complex trait epistasis improve phenotype prediction and biological pathway recovery in humans.

- Interestingly, Turchin et al. (bioRxiv 2020) found that non-European populations have more complex trait epistasis compared to European cohorts, providing a potential reason why complex trait epistasis has only recently been discovered.

We added the following text to the Discussion describing the relevance of higher-order epistasis in human GWAS.

Discussion: *“The relevance of higher-order epistasis in human GWAS remains a topic of substantial debate. For example, there are many opinions on whether epistasis is a frequent source of missing heritability for human traits, e.g. [118] argues that epistasis does not contribute to heritability while [119, 120] argue the opposite. We note that several recent papers have identified complex trait epistasis in humans, including pairwise and higher-order epistatic interactions between pathways [121, 122] and individual SNPs [123, 124], suggesting that epistasis is relevant for human genetics.”*

Even if there were, the mutations would be in different genes and so the multi-locus effect on “fitness” (which is not easily measured in humans although proxies such as relative lifetime reproductive success exist) for a specific set of “knockouts” cannot be estimated.

We emphasize that our model is not restricted to gene knockouts (as analyzed by Kuzmin et al), and can be used to quantify any kind of interaction, including interactions between SNPs or drugs. In particular, our model can measure interactions between *any* type of biallelic mutation, including mutations that do not result in a loss of function (i.e. almost all SNPs) and mutations in the same gene, as measured in observational genetics studies.

Importantly, **we derive fundamental connections between our model and the two standard approaches for measuring pairwise SNP-SNP interactions in a case-control GWAS:** (1) logistic regression and (2) a conditional independence test (e.g. see Cordell, *Nat Rev Genet* 2009 and Wei et al, *Nat Rev Genet* 2014). Specifically, we prove that the interaction term in logistic regression is equal to the 3-way interaction term β_{ijk} in a multivariate Bernoulli distribution, while the conditional independence test is equivalent to testing whether a 2-way interaction term β_{ij} and a 3-way interaction term β_{ijk} are both equal to zero.

- These interaction terms are equal to the corresponding log-multiplicative epistasis measure $\log \epsilon^M$ with fitness values equal to empirical genotype frequencies.

To our knowledge, the relationship between the multivariate Bernoulli, logistic regression, and conditional independence testing has not been described previously in either the genetics or statistics literature. Thus, **we have added new mathematical results to the manuscript describing these relationships.** See Supplementary Text F for details.

We note that for measuring SNP interactions in GWAS, it is critical to have accurate measures of epistasis, as one may expect relatively small epistasis between SNPs (since most SNPs have small effect sizes). Our new mathematical results demonstrate that our log-multiplicative measure $\log \epsilon^M$ is the appropriate measure for measuring SNP-SNP interactions, and the chimeric measure ϵ^C should not be used to measure such interactions. For example, a recent paper (Reese et al, *Statistics in Bioscience* 2022) uses the joint cumulant to measure SNP-SNP interactions in case-control GWAS of a binary trait. Our work shows that this approach is not sound.

Finally, we emphasize that there is a long history of measuring epistasis using gene knockouts in model organisms, e.g. as reviewed by Mackay (*Nat. Rev. Genet.* 2014). These studies have led to many important biological insights including how genes are organized into metabolic and signaling pathways; the scale of genetic redundancy in eukaryotic genomes; and improved understanding and prediction of the evolution of antibiotic resistance in drug treatments. Because of the importance of these applications, the measurement of higher-order epistasis with gene knockouts in model organisms has been studied by *many* papers in *Nature Communications* and other journals with similar readership (e.g. Poelwijk et al., *Nat Comm*

2019; Jones et al., *Nat Comm* 2014; Wang et al, *Nat Comm* 2014; Zhou and McCandlish, *Nat Comm* 2020; Szappanos *Nat Genet* 2011; Forsberg et al., *Nat Genet* 2017; Celaj et al., *Cell Systems* 2020; Lik Ang et al, *Cell Genomics* 2023). Thus, we believe that our work is of interest to the readers of *Nature Communications*.

Figure 1C: Does the sign discordance fraction asymptote to 0.5? And if so is that because the probability that the chimeric epistatic residual is negative goes to 1 when the interaction order is large?

This is a good observation. Indeed, the sign discordance fraction does seem to converge to 0.5 as the interaction order gets larger. The reason is because, for large interaction orders, the chimeric measure and multiplicative measure have very low correlation. For example, we demonstrate using simulations (see Figure S3 below) that the Pearson correlation between the chimeric and multiplicative measures decreases with the interaction order.

Figure S3: Pearson correlation $\rho(\epsilon^C, \log \epsilon^M)$ between chimeric measure ϵ^C and multiplicative measure ϵ^M for different interaction orders K and each K -tuple of loci across all simulated instances of the multiplicative fitness model in Section 2.4 (pairwise interactions with noise parameter $\sigma = 0$). The x - and y -axis ranges are $[\min \epsilon^M, \max \epsilon^M]$.

We thank the reviewers for their thorough and helpful comments. Below we include the reviewer text (black) and our response (blue text). In the revised manuscript, we have highlighted all major text changes in blue.

Reviewer 1:

The problems pointed out my previous review have been taken care of. The revised manuscript is considerably more readable. I also appreciate that the authors have provided a little bit more context in the revised manuscript, including the remark that the chimeric formula ϵ^C is close to ϵ^M for fitness values close to 1, and that ϵ^C_{ij} and ϵ^M_{ij} agree about the sign of pairwise epistasis. The main part of the manuscript has improved quite a bit.

We thank the reviewer for their positive feedback on our revised manuscript.

However, a second and more careful reading of the manuscript revealed more problems. As much as the criticism of using the chimeric formulas as a default is convincing (the manuscript demonstrates that the higher order chimeric formulas has produced misleading results in empirical studies), the authors have not provided convincing arguments for that the chimeric formula should be avoided regardless of context. In particular, the manuscript criticizes some publications where I am not aware of any errors or poor methodology.

We agree with the reviewer that we have overstated our critiques on the usage of the chimeric measure in the selection/recombination literature. In response, we have removed these critiques from the manuscript; see our response to point (2) in 1.1 below for more details.

Another problem concerns the multivariate Bernoulli distribution (MBD). From the abstract: "We resolve these inconsistencies by deriving fundamental connections between the different epistasis formulae and the parameters of the multivariate Bernoulli distribution." Theorem 1 in the manuscript depends on the assumption that the probability of observing a genotype is proportional to its fitness. However, the assumption is almost never realistic for an evolutionary process. Because of that assumption, one can derive Theorem 1 (since the proportionality constants cancel out) which explains the claimed connection to the MBD.

First, we revised the referenced and subsequent sentences in the abstract based on reviewer comments to reduce confusion about the connections to the MBD as follows:

Abstract:

interactions with positive interactions, and vice versa. We resolve these inconsistencies by deriving mathematical relationships between the different epistasis formulae and different parametrizations of the *multivariate Bernoulli distribution*. Further, we argue that the parametrization corresponding to the chimeric formula does not appropriately model interactions between the Bernoulli random variables. In simulations, we show that the additive/multiplicative epistasis formulae are more accurate than the chimeric formula, and that the chimeric formula may falsely detect higher-order epistasis that is not present in the data. Furthermore, using multi-gene knockout data in yeast, multi-way

Second, we agree with the reviewer that our assumption that fitness is proportional to the probability $P(\text{genotype})$ of a genotype may be unrealistic for evolutionary processes. However, there are several reasons that we make the assumption.

(1) Our motivation in writing these Theorems down is that the equations for the different MBD parameters exactly match the functional form of the additive, multiplicative, chimeric measures as well as the Walsh coefficients, as we show in Table 1. For example, the pairwise (log-)natural parameter $\log(\beta_{12})$ of the MBD has the form

$$\log(\beta_{12}) = (p_{11} p_{00}) / (p_{01} p_{10}),$$

while the pairwise multiplicative measure has the form

$$\varepsilon_{ij}^M = (f_{11} f_{00}) / (f_{01} f_{10}).$$

Looking at these two equations, it is natural to equate the two under the assumption that the fitness values f are proportional to MBD probabilities p . Theorem 1 shows this equivalence for pairwise and higher-order interactions.

(2) Many of the datasets that we analyze measure the fitness of a genotype as the *relative frequency* of the genotype in a large population of many different genotypes. These include the Kuzmin et al. yeast data, which we analyze in Section 2.4, and many of the protein datasets which we analyze in Section 2.6. In other words, these datasets measure the fitness of a genotype as being (proportional to) the probability $P(\text{genotype})$ of a genotype appearing in a large population. Thus, making the mathematical assumption that “fitness is proportional to $P(\text{genotype})$ ” seems reasonable in order to understand real-world datasets which measure fitness under this assumption.

(3) We emphasize that Theorem 1, which relates the MBD to the multiplicative / additive epistasis measures, is not intended to be a *biological* motivation for these epistasis measures. Instead, Theorem 1 provides an **alternative, statistical interpretation** for the multiplicative / additive epistasis measures – i.e. as parameters of the MBD – which we have not seen written out previously in the literature. One could certainly derive a different epistasis measure by removing the assumption that fitness is proportional to $P(\text{genotype})$; e.g. by incorporating genetic drift or other factors.

We have updated the manuscript (see below) to emphasize that the MBD provides a statistical interpretation of epistasis, as opposed to describing an interpretation implicit in other derivations.

Introduction:

We resolve the mathematical and biological inconsistencies between the different epistasis formulae by deriving connections between epistasis and the parameters of the multivariate Bernoulli distribution (MVB), a probability distribution on binary random variables [29]. In particular, we show that a wide array of approaches for quantifying epistasis – including the additive, multiplicative, and chimeric formulae, as well as the regression models commonly used in GWAS and QTL analyses [5, 2] and the Walsh coefficients for measuring background-averaged epistasis [30, 31, 32] – are equivalent to computing different parameterizations of the MVB, showing that the MVB provides a unifying statistical framework for the different epistasis measures. To our knowledge, the statistical interpretations of the various epistasis formulae that we derive have not been previously described in the literature.

Methods Section 2.3:

Thus, our results show that the additive and multiplicative epistasis measures are equivalent to computing interaction terms in regression models commonly used in genetics.

Moreover, we have added a statement to the Discussion noting that one future direction is to make our assumption that fitness is proportional to $P(\text{genotype})$ more evolutionarily realistic.

Discussion:

tainty of fitness measurements in the MVB, e.g. by using a Bayesian framework. Thirdly, one could generalize our theoretical results by relaxing the assumption that fitness is proportional to genotype probability, e.g. by incorporating genetic drift or other more evolutionarily realistic factors. Fourth, our statistical framework could be extended to

Finally, we also updated the Results Section 2.3 and Methods to remove the claim about a population evolving after one unit of time. Instead, we now note that many real-world experimental fitness datasets measure fitness in terms of the relative frequency of a genotype.

Results Section 2.3:

A natural approach for studying a fitness landscape function f is to view it as a *distribution* on the set $\{0, 1\}^L$ of binary strings, where the probability $p_{\mathbf{x}}$ of a binary string \mathbf{x} is derived from its fitness $f_{\mathbf{x}}$. Such distributions are often used by protein structure models [50]. Moreover, many real-world fitness datasets – including the yeast fitness data analyzed in Section 2.5 and many of the protein datasets analyzed in Section 2.7 – measure the fitness of a genotype \mathbf{x} in terms of its relative frequency in a large population of genotypes, i.e. its probability $p_{\mathbf{x}}$.

Methods:

The assumption that the probability $p_{x_1 \dots x_L}$ of observing a genotype (x_1, \dots, x_L) is derived from its fitness $f_{x_1 \dots x_L}$ is often used in generative models for estimating the fitness of protein structures from sequence data [50]. Moreover, many real-world fitness datasets – including the yeast fitness data analyzed in Section 2.5 and many of the protein datasets analyzed in Section 2.7 – measure the fitness of a genotype \mathbf{x} in terms of its relative frequency in a large population of genotypes, i.e. its probability $p_{\mathbf{x}}$.

From a quick look, the connections between the MBD and GWAS studies seem more promising, but the current presentation in the manuscript is rather brief (a more thorough discussion about the MBD and GWAS studies would be interesting).

We agree that a further study on the relationship between the MBD and GWAS approaches for measuring epistasis would be quite interesting. We have added this as a future direction in the Discussion.

Discussion

There are several future directions for our work. First, it would be useful to further investigate the relationship between the MVB and regression-based approaches for quantifying epistasis in GWAS, e.g. [85, 86]. For exam-

The main part of the manuscript, i.e., the criticism of using the chimeric formula ϵ^C as a default, is convincing. The authors show that chimeric epistasis and multiplicative epistasis disagree for empirical studies, where most readers would agree that multiplicative epistasis is more natural. In particular, higher order chimeric epistasis seems close to bizarre. The value of the criticism is independent of connections to any particular theory (such as the MBD). I would strongly support a publication of the main part of the manuscript (after a minor revision).

We thank the reviewer for their positive comments on our empirical study comparing the chimeric and multiplicative epistasis measures.

The discussion about the MBD and epistasis is problematic for the reasons stated, and also because of some controversial opinions. Not everyone will agree that a) the soundness of an epistasis measure depends on how it relates, or fails to relate, to the MBD, b) that the MBD is implicitly assumed in work on Walsh coefficients (see e.g. Crona and Greene, 2024), or c) that $\epsilon^C_{\{ij\}}$ should be avoided because it mixes additive and multiplicative scales. I think the controversial opinions could distract from the main part of the manuscript. A somewhat drastic solution would be to split the manuscript and postpone the discussion about the MBD and epistasis to a second manuscript.

Regarding points a) and b): we did not aim to argue that an epistasis measure *must* be related to the MBD for the measure to be useful, nor that the MBD is implicit in work on Walsh coefficients/additive multiplicative measures. As the reviewer notes, papers such as Crona and Greene (2024) present derivations of the epistasis measures that do not use the MBD.

Rather, our goal was to argue that several of the commonly used epistasis measures are equivalent to estimating the parameters of the MBD, but using different parameterizations of the MBD. Thus, the MBD provides an *alternative*, statistical interpretation of the different epistasis measures. See the third response above for specific changes we made to the manuscript.

For c), we have weakened our claim that the pairwise chimeric measure is not appropriate for all situations by removing our critiques of the pairwise chimeric measure in the recombination / selection literature; see response to points (1), (2) in 1.1 below.

Finally, we do argue that the mixing of scales in the chimeric epistasis measure is undesirable for high-order interactions. We provide evidence for our argument by reanalyzing the Kuzmin et al. yeast data and the Tekin et al. drug response data (where the higher-order chimeric measure was used in the original publications) as well as deep mutational scanning (DMS) datasets of several proteins (where the additive/multiplicative measure was used in the original publications). We welcome criticisms of this argument and our supporting evidence in

subsequent works, but do not feel that a potentially controversial argument should preclude publication of our work. On the contrary, we would argue that works such as Kuzmin et al. and Tekin et al. are “controversial” as they use the chimeric model in place of the standard multiplicative model.

1. DETAILED COMMENTS

1.1. Epistasis and recombination.

(1) From the manuscript (Page 4): “The chimeric epistasis measure $\epsilon^{AC}_{\{ij\}}$ is widely used because of its interpretation as a residual, i.e., the difference between the observed and expected values of a measurement. However, despite the simplicity of this explanation, it is not statistically sound, as residuals are only appropriate for additive models.”

Comment: If the authors claim that the measure $\epsilon^{AC}_{\{ij\}}$ is never sound, I disagree. If $\epsilon^{AC}_{\{ij\}}$ can be used for say computations of selection coefficients, then there is no reason to avoid it.

We agree with the reviewer that the *pairwise* chimeric measure may have some application, e.g. in selection formulas as the reviewer notes below. Thus, we have weakened our statement to remove that residuals are never sound for multiplicative models, and we note that residuals are *typically* only defined for additive models; see quoted text below.

Results, Section 2.1:

i.e. the difference between the observed and expected values of a measurement. However, despite the simplicity of this explanation, residuals are *typically* only appropriate for additive models. For multiplicative models, it is standard

(2) From the manuscript (Page 16):

”As another example, the formulae derived in the theoretical population genetics literature [38, 39, 40] that relate recombination, selection, and the pairwise chimeric epistasis will change if pairwise epistasis is instead measured using the multiplicative measure.”

Comment: The criticism needs to be clarified. Do the authors claim that the selection formula described in Otto and Lenormand (2002) is incorrect (see below)? In that case the authors should provide a counterexample. Or is the claim rather that the theory on selection modifiers could be phrased in a better way if one avoids the chimeric epistasis formula? In that case the authors should provide an alternative text.

Unless the criticism can be clarified, I do not understand why the authors would want to include it in the manuscript.

Recombination modifiers are discussed in Otto and Lenormand (2002) with reference to work by Nick Barton. In brief, the chimeric formula (denoted ϵ below) appears in a discussion about the selection for a newly arisen recombination modifier M that alters the recombination rate ρ by an amount δ_ρ . The system have alleles A/a at locus A and B/b at locus B , and the selection coefficients for a and b are denoted s_a and s_b . D denotes the linkage disequilibrium: $D =$

frequency(AB) \times frequency(ab)–frequency(Ab) \times frequency(aB). According to the theory, the selection on the recombination modifier under some given assumptions equals:

$$\frac{\delta\rho}{\rho_{MAB}}D(\lambda - \varepsilon), \text{ where}$$

$$\lambda = -(s_a \times s_b) \left[\frac{1}{\rho_{MA}} + \frac{1}{\rho_{MB}} - 1 \right].$$

(There is in fact a typo in the original article but the expression for λ here is correct.)

As the reviewer notes, some papers in the theoretical epistasis literature derive selection formulas in terms of the pairwise chimeric measure ϵ_{ij}^c . Our intention was that some of the selection formulas (e.g. the Barton formula referenced above by the reviewer) could possibly be rephrased in terms of the multiplicative formula rather than using the chimeric epistasis measure. However, this claim is hypothetical and we have not attempted such a derivation. Thus, we have removed all statements in the Introduction and Discussion on the pairwise chimeric measure and its relation to the selection/recombination literature.

1.2. Epistasis and the multivariate Bernoulli distribution.

(1) From the manuscript (Page 6):

”Moreover, modeling fitness as a probability has a natural biological interpretation: if the growth rate of an organism with genotype \mathbf{x} is given by its fitness $f_{\mathbf{x}}$, and if there are initially an equal number of organisms of each of the 2^L genotypes $\mathbf{x} \in \{0, 1\}^L$ then after one unit of time the frequency $p_{\mathbf{x}}$ of each genotype \mathbf{x} will be proportional to its fitness $f_{\mathbf{x}}$... ”

Comment: That is right. However, the assumption about ”equal number of organisms” is very special. Also if the assumption does hold, after a second time unit the genotype frequencies are no longer equal, and consequently the proportionality is lost.

As far as I can see, the assumption about ”equal number of organisms” is not realistic for any evolutionary process. Neither does the assumption work as an approximation. For instance, if

$$f_{00} = 1, f_{10} = 1.1, f_{01} = 0.9, f_{11} = 1.2$$

then the genotype 01 will be exceedingly rare throughout an evolutionary process from the wild-type 00 to 11 (in the absence of recombination).

(2) Theorem 1 assumes that the probability of observing a genotypes is proportional to its fitness, and similarly for other results that relate the MBD and epistasis. I find the assumptions problematic for evolutionary processes.

We agree with the reviewer that our justification for fitness being proportional to $P(\text{genotype})$, which makes an argument about a population with ”an equal number of organisms” evolving after ”one unit of time”, may not describe an evolutionarily realistic process. We have updated the manuscript to remove our claim about a population evolving after one unit of time. Instead,

we now note that many real-world experimental fitness datasets measure fitness in terms of the relative frequency of a genotype. See the third response above for details.

(3) The proof of Theorem 1 is almost identical to a [standard] derivation of the natural parameters for the MBD (the proportionality constants cancel out in the derivation), i.e., the connection between the MBD and ϵ^M is not surprising but rather a direct consequence of a [questionable] assumption.

Following the reviewer's comment, we further investigated the connection between our Theorem and the results in the multivariate Bernoulli paper by Dai et al (2013). We have added a statement below Theorem 1 noting a connection between our Theorem and Lemma 3.1 in Dai et al. (2013).

Methods

We note that Theorem 1 follows from Lemma 3.1 in [29] which states a formula relating the general parameters \mathbf{p} and the natural parameters β of a multivariate Bernoulli distribution. Theorem 1 follows by showing that the right-hand side of the formula in Lemma 3.1 is equal to the multiplicative epistasis measure.

(4) From the manuscript (Page 2): "We demonstrate that this connection to the multivariate Bernoulli is implicit in several other approaches for quantifying epistasis, including the regression models commonly used in GWAS and eQTL analyses [3, 4, 48] and the Walsh coefficients for measuring background-averaged epistasis [41, 49, 50]. To our knowledge, the connections we derive between the MVB and the various epistasis formulae have not been previously described."

Comment: The hypothesis about implicit connections between the MBD and measures of epistasis has problems. Consider the class of functions $f : \{0,1\}^L \rightarrow \mathbb{R}$, where the default expectations is multiplicative. Such functions appear in many contexts (not necessarily related to probability distributions). For such functions, it is natural to consider additive deviations on the log scale, such as $F_{11} + F_{00} - F_{10} - F_{01}$. With pairwise deviations as a starting points, one can derive Walsh coefficients directly (as deviations from additive expectations that also take lower order deviations into account, see e.g. Crona and Greene (preprint 2024), i.e., the MBD has no role in the derivation.

For an alternative perspective on the MDB and epistasis: similar assumptions, such as a multiplicative default expectations [standard for both fitness and probabilities] generate similar formulas, in particular Walsh coefficients appear naturally in both cases.

While the Walsh coefficients are also connected to the MBD (Table 1), we agree with the reviewer that one can derive the Walsh coefficients and epistasis terms without referring to the MBD. We appreciate the reference to Crona and Greene (2024) for some examples of such derivations.

We have removed all claims that the MBD is “implicit” in the different epistasis measures. Instead, we write that the MBD provides a *statistical interpretation* of the epistasis measures. See our third response above for details.

The most controversial opinions/hypotheses in the manuscript (if I understand the text right) are:

(1) The value (or soundness) of an epistasis measure in evolutionary biology depends on how it appears, or fails to appear, in the MBD.

(2) The MBD is implicitly assumed in epistasis and closely related concepts that have been applied in evolutionary biology.

(3) An epistasis measure that mixes additive and multiplicative scales should be Avoided.

Points (1) and (2) are not the key claims in our manuscript, and we have adjusted the manuscript to not avoid confusion on these points. Specifically, as noted above: (1) we do not write that the MBD is “implicit” in epistasis measures, and instead write that the MBD provides a new, unifying statistical interpretation of different epistasis measure, and (2) we have removed our critiques of the pairwise chimeric measure in the recombination/selection literature.

Point (3) is a key claim in our manuscript, and as we noted above, we are comfortable with this being further debated in the literature.

In any case, a more thorough discussion about the MBD and GWAS studies could be interesting.

A couple of minor things:

(1) There is a typo in the second formula on Page 2.

We have fixed the typo.

(2) The article Dai et al. (2013) could be cited.

We thank the reviewer for pointing out our error. We mistakenly cited a different paper for the multivariate Bernoulli instead of Dai et al (2013). We have updated our multivariate Bernoulli reference to Dai et al.

Reviewer #2 (Remarks to the Author)

We thank the reviewer for their helpful comments that have significantly enhanced our manuscript.

This paper makes a useful point about the illogicality of measuring epistatic effects between mutations by applying an additive measure to a multiplicative model of the null expectation of a trait in the absence of interactions (their ‘chimera’).

We thank the reviewer for finding our manuscript useful.

(Epistasis in the sense used here is equivalent to interactions between main effects in the analysis of variance, and this concept introduced by RA Fisher, the inventor of ANOVA, in 1918 - oddly, they do not cite Fisher until ref. 130.)

We thank the reviewer for noting the history of epistasis as introduced by Fisher. We have added a citation to Fisher 1918 in the Results, Section 2.1, as shown below.

Results Section 2.1:

In the *additive* model, mutations are assumed to have an additive effect on fitness, and the pairwise epistasis measure ϵ_{ij}^A is equal to the difference between the observed and expected double-mutant fitness values:

$$\epsilon_{ij}^A = f_{ij} - (f_i + f_j), \quad (1)$$

under the assumption that fitness values are normalized such that the wild-type fitness $f_{\emptyset} = 0$. The sign of the interaction (i.e. positive vs. negative) is given by the sign $\text{sgn}(\epsilon_{ij}^A)$ of the epistasis measure ϵ_{ij}^A . The additive model was first posed by Fisher [37], who used the term “epistacy” to refer to any statistical deviation from additivity [6].

The paper is, however, rather poorly presented, with a lot of repetition, and the quite simple message gets lost in a maze of detail. It could usefully be made much shorter and clearer. In addition, there are a number of instances in which cited work has been incorrectly described, which needs to be corrected (see Specific Comments). The number of references could be cut considerably, as many of them are reviews with overlapping contents.

We have tightened our presentation of our work as suggested by the reviewer.

- We shortened Sections 2.1 and 2.2 by half a page, moving most of the details to the Methods.
- We have removed 45 citations, including reviews with overlapping content, and now have 100 references in the manuscript.

They also never mention that they exclusively consider haploid genomes; with diploidy, it is much harder to describe epistasis, as interactions between additive genetic and dominance effects need to be modelled (see any textbook of quantitative genetics). The results are thus of more limited application than they seem to think.

We have updated the Introduction (paragraph 2), Results (Section 2.1), and Methods (paragraph 1) to explicitly state that we study the fitness of **haploid** genomes.

Introduction:

While epistasis is a property of any quantitative trait, many studies have measured epistatic interactions using experimental fitness data from **haploid genomes** (reviewed in [6, 7, 8]). Most of these studies measure *pairwise*

Results:

2.1 Pairwise epistasis: additive, multiplicative, and chimeric

Pairwise epistasis describes interactions between two genetic loci. We **consider haploid genomes** and assume that each locus is *biallelic*, i.e. each locus has two alleles labeled 0 and 1. Thus for a pair of loci there are four possible

We emphasize that our model of haploid genomes is sufficient for several applications including drug interactions and epistasis in proteins.

The discussion of GWAS etc in humans (p.15) is thus not very relevant.

We have removed the GWAS discussion from the Discussion, and instead mention it as a future direction.

Discussion:

There are several future directions for our work. First, it would be useful to further investigate the relationship between the MVB and regression-based approaches **for quantifying epistasis in GWAS**, e.g. [85, 86]. For exam-

Importantly, the authors have not brought out a basic point about fitness measures. They are largely focussed on microbial systems, where genetically manipulated strains reproducing asexually are competed against each other. In such systems, the fitness of a strain is generally (but not always) measured by its growth rate, i.e. the rate of change of the logarithm of its number of individuals, or a proxy for it. In contrast, fitness in higher organisms is frequently measured by the expected life-time reproductive success of a genotype or phenotype, and it is usually assumed that independent effects on survival and probability of reproduction combine multiplicatively, although the justification for this is weak for traits like fecundity.

A multiplicative model is, however, inappropriate for microbial growth rate fitness measures; for strain differences in growth rates, one can use additivity as the baseline model for estimating epistatic parameters. This very basic point seems to have escaped the attention of most workers in the field, despite J-L Chevin having pointed out in 2011 the difference between fitness measures based on discrete generation population genetics models and growth-rate measures, and the resulting errors in interpreting microbial data (Biology Letters 7:210). For example, in ref. 23, a log transform of the ratio of growth rates of mutant to wild-type strains of *E. coli* was used to test for epistasis. It is, of course, illogical to use a log of a log for this purpose. Papkou et al. (ref. 113, SM, pp.9-10) have recently drawn attention to this issue. Population growth rate has a long history as fitness measure. Fisher (1930, *The Genetical*

Theory of Natural Selection, Chap. 2) introduced it (his “Malthusian parameter”) as a fitness measure; there are, however, problems in using this for higher organisms with complex life histories (Charlesworth 1994, Evolution in Age-Structured Populations).

As we noted in the previous answer, we now emphasize that our manuscript focuses on studies using haploid genomes, e.g. microbial systems. We agree with the reviewer that a multiplicative model is inappropriate when fitness is measured using the growth rate of a clonal population, and we appreciate the Chevin 2011 reference which we were not aware of. We note that if fitness is measured using the *relative frequency* of the population (e.g. the colony size of a microbial population) – rather than the growth rate – then a multiplicative model is appropriate, since the growth rate is proportional to the logarithm of the relative frequency (e.g. as noted by Russ and Kishony 2018).

To address these confusions, we added a new section to the Methods where we discuss how to choose the different fitness model (additive vs. multiplicative) depending on the trait being measured. We include the Chevin 2011 reference, and we are happy to take any further suggestions from the reviewer.

Methods:

Choosing an appropriate null model. The appropriate choice of null fitness model depends on the quantitative trait being used to approximate fitness. Cellular growth rate (i.e. the Malthusian parameter [92], which is often used as a measure of fitness of microbial populations) is typically described with an additive fitness model; this is because in the population genetics literature, it is often assumed that mutations that independently effect survival and reproduction probabilities combine multiplicatively within individual cells [93, 94], and so these mutations will combine additively in their effect on the growth rate of a continuously-growing clonal population [75, 11]. However, multiplicative models are sometimes still used inappropriately in this setting [9]. On the other hand, the relative frequency of a microbial (or protein) population is typically modeled multiplicatively, as growth rate of a population is proportional to the logarithm of its relative frequency. In general, Wagner suggests that one should use the model that preserves the scale on which single-mutant fitness effects were measured (i.e. additive or fold differences from wild type; [49, 13]).

The situation is thus quite simple far when microbial growth rates are used as the measure of fitness; just look for deviations from additivity, for which the statistical machinery is well-worked out (e.g. ref. 41). On p.34, where they describe how they reanalyze a number of datasets, they mention that some studies used additive measures and others multiplicative measures, but don't explain why this is the case. It's not clear why they felt they needed to transform from additive to multiplicative and then compare with chimeric; they could simply make their point by using examples where the authors applied chimeric to multiplicative (e.g. refs. 2 and 42, which use colony size). This would spare the reader much unnecessary detail. For some of the cases, such as fluorescence, the choice of scale is somewhat arbitrary, since the relation between trait and fitness is unknown.

As the reviewer notes, we did not state how the fitness values were measured in the protein fitness datasets, nor which proteins had fitness values measured on an additive scale versus on a multiplicative scale. Thus, we have added this information to Methods (Section 4.6):

4.6 Epistasis between protein mutations

The analysis in Section 2.7 was performed using publicly available DMS data for the following proteins/RNA molecules:

- the *E. coli* metabolic protein folA [76], where the fitness of a genotype is measured by the logarithm of its relative frequency in a large population;
- the *Streptococcus pyogenes* Cas9 (SpCas9) nuclease [38], where the fitness of a genotype is measured by the logarithm of its relative frequency in a large population;
- the immunoglobulin-binding protein G domain B1 (GB1), expressed in Streptococcal bacteria [77, 78], where the fitness of a genotype is measured by its relative frequency in a large population;
- the Omicron BA.1 variant of the SARS-CoV-2 virus [79], where the fitness of a genotype is measured by the logarithm of its binding affinity relative to the Wuhan Hu-1 strain;
- the *Entacmaea quadricolor* fluorescent protein eqFP611 [80], where the fitness of a genotype is measured by (a normalized version of) its relative frequency in a large population;
- the *Aequorea victoria* green fluorescent protein avGFP [81], where the fitness of a genotype is measured by the logarithm of its fluorescence;
- the green fluorescent proteins (GFPs) from [10], where the fitness of a genotype is measured by its fluorescence;
- yeast tRNA [82], where the fitness of a genotype is measured by the logarithm of its relative frequency in a large population; and
- the *Chlamydomonas reinhardtii* flavin mononucleotide (FMN)-based fluorescent protein CreiLOV [83], where the fitness of a genotype is measured by the logarithm of its fluorescence.

More broadly, our aim in analyzing these deep mutational scanning (DMS) protein datasets was to demonstrate that the chimeric measure yields substantially different results than the multiplicative measure on other real fitness values besides the Kuzmin et al. (Section 2.5) and antibiotic resistance (Section 2.6) datasets. While none of the original DMS dataset publications used the chimeric measure – and instead in our view correctly measured epistasis using an additive or multiplicative measure – we feel that showing the differences between the chimeric and multiplicative/additive measures on many datasets strengthens the argument of the deficiencies of the chimeric measure.

We added the following paragraph to Methods Section 2.7 to emphasize that we transform all fitness DMS datasets to a multiplicative scale to demonstrate the disagreement between the multiplicative measure versus chimeric measure.

Methods Section 2.6:

We aim to use these protein fitness landscapes to directly compare the multiplicative and chimeric epistasis measures. However, the quantitative trait used to measure fitness – and thus the appropriate scale for measuring epistasis (additive or multiplicative) – varies across the different proteins. In particular, the original publications for the *folA* protein, SpCas9 protein, eqFP611 protein, COVID spike protein, avGFP protein, yeast tRNA, and CreiLOV protein assume that fitness is measured using an additive scale (e.g. by measuring fitness as the logarithm of the relative frequency of a genotype). Since our aim is to demonstrate the disagreement between the multiplicative and chimeric measures, we first transform the fitness measurements to a multiplicative scale by exponentiating the fitness values f , i.e. $f \rightarrow e^f$, before computing the multiplicative measure. This allows us to directly compare the multiplicative epistasis measure ϵ^M with the chimeric epistasis measure ϵ^C , which implicitly assumes fitness values are measured using a multiplicative scale.

It would also help to make it clearer why there has been such emphasis on multiplicativity in the fitness and population genetics literature. It partly comes about from the fact that it is reasonable to assume that independent effects on survival probability combine multiplicatively, so that single-locus results can be generalised easily to multiple loci (e.g. Haldane 1937 *Am Nat* 71:337), and partly from the fact that the effects of selection on traits subject to directional selection in discrete generation models suggest that multiplicative fitness effects do not generate non-random associations between loci (e.g. Shnol & Kondrashov 1993 *Genetics* 134:995). However, it has long been known that multi-locus systems require additivity of fitness effects for there to be an absence of such associations among polymorphisms maintained by selection when recombination is rare (reviewed by Ewens 2004 *Mathematical Population Genetics*), so that the biological relevance of multiplicativity versus additivity is context dependent.

We thank the reviewer for the helpful exposition and references. As we noted above, we have added a section to Methods that reviews which null fitness model should be used depending on the fitness measurement, and we include citations to the references the reviewer notes (Haldane 1937, Shnol & Kondrashov 1993).

Methods

Choosing an appropriate null model. The appropriate choice of null fitness model depends on the quantitative trait being used to approximate fitness. Cellular growth rate (i.e. the Malthusian parameter [92], which is often used as a measure of fitness of microbial populations) is typically described with an additive fitness model; this is because in the population genetics literature, it is often assumed that mutations that independently effect survival and reproduction probabilities combine multiplicatively within individual cells [93, 94], and so these mutations will combine additively in their effect on the growth rate of a continuously-growing clonal population [75, 11]. However, multiplicative models are sometimes still used inappropriately in this setting [9]. On the other hand, the relative frequency of a microbial (or protein) population is typically modeled multiplicatively, as growth rate of a population is proportional to the logarithm of its relative frequency. In general, Wagner suggests that one should use the model that preserves the scale on which single-mutant fitness effects were measured (i.e. additive or fold differences from wild type; [49, 13]).

Finally, I found their justification of the use of the multivariate Bernoulli distribution to analyze fitness epistasis unconvincing. On p.30, they simply say that the components of the probability vector “might be derived from the frequency of observing the genotype .. in a large population”. This is incorrect; the frequency of a genotype in a population is only very loosely related to its fitness – see any textbook of population genetics. A justification of what they are doing in the context of microbial experiments that use colony size as a fitness proxy is as follows. The size at

time t of a colony with genotype i and growth-rate r_i , relative to its initial size, is $\exp(r_i t)$; call this w_i . Normalise this by the sum of the w_i ; you then get a vector whose components sum to one, which formally behaves like a vector of probabilities (but are not true probabilities). This is similar to what they say on p.31 (after Corollary 1), but removes the requirement of equal initial numbers.

We emphasize that our motivation for using the multivariate Bernoulli is mathematical rather than biological. Namely, the equations for the different MBD parameters exactly match the functional form of the additive, multiplicative, chimeric measures and Walsh coefficients as we show in Table 1. For example, the pairwise (log-)natural parameter $\log(\beta_{12})$ of the MBD has the form

$$\log(\beta_{12}) = (p_{11} p_{00}) / (p_{01} p_{10}),$$

while the pairwise multiplicative measure has the form

$$\epsilon_{ij}^M = (f_{11} f_{00}) / (f_{01} f_{10})$$

Looking at these two equations, it is natural to equate the two under the assumption that the fitness values f are proportional to MBD probabilities p . Theorem 1 shows this equivalence for pairwise and higher-order interactions.

We also note that many of the datasets that we analyze measure the fitness of a genotype as the *relative frequency* of the genotype in a large population of many different genotypes. These include the Kuzmin et al. yeast data, which we analyze in Section 2.4, and many of the protein datasets which we analyze in Section 2.6. In other words, these datasets measure the fitness of a genotype as being (proportional to) the probability $P(\text{genotype})$ of a genotype appearing in a large population. Thus, making the mathematical assumption that “fitness is proportional to $P(\text{genotype})$ ” seems reasonable in order to understand real-world datasets which measure fitness under this assumption.

That said, we agree with the reviewer that in an evolving population, the frequency of a genotype is not solely derived from its fitness. We have removed this claim from Methods page 27 (what the reviewer refers to as p.30). Moreover, we have updated the Results and Methods to justify our fitness assumption by noting that many real-world experimental fitness datasets measure fitness in terms of the relative frequency of a genotype (e.g. in terms of growth rate or colony size), as shown below.

Results Section 2.3:

A natural approach for studying a fitness landscape function f is to view it as a *distribution* on the set $\{0, 1\}^L$ of binary strings, where the probability p_x of a binary string x is derived from its fitness f_x . Such distributions are often used by protein structure models [50]. Moreover, many real-world fitness datasets – including the yeast fitness data analyzed in Section 2.5 and many of the protein datasets analyzed in Section 2.7 – measure the fitness of a genotype x in terms of its relative frequency in a large population of genotypes, i.e. its probability p_x .

Methods:

The assumption that the probability $p_{x_1 \dots x_L}$ of observing a genotype (x_1, \dots, x_L) is derived from its fitness $f_{x_1 \dots x_L}$ is often used in generative models for estimating the fitness of protein structures from sequence data [50]. Moreover, many real-world fitness datasets – including the yeast fitness data analyzed in Section 2.5 and many of the protein datasets analyzed in Section 2.7 – measure the fitness of a genotype \mathbf{x} in terms of its relative frequency in a large population of genotypes, i.e. its probability $p_{\mathbf{x}}$.

It would be interesting to derive epistasis measures under a more biologically realistic assumption on the fitness values, e.g. by incorporating genetic drift. We have added this as a future direction in the Discussion.

Discussion:

tainty of fitness measurements in the MVB, e.g. by using a Bayesian framework. Thirdly, one could generalize our theoretical results by relaxing the assumption that fitness is proportional to genotype probability, e.g. by incorporating genetic drift or other more evolutionarily realistic factors. Fourth, our statistical framework could be extended to

But this is not really needed to get to their Equation 26; if you assume take the log transform of w_i , you are back to a quantity that can be treated in terms of the standard linear model of main effects and interactions. Equation 26 is thus almost trivially obvious as a general form, as is the fact that they end up with the standard Walsh coefficients after the log transform.

First, we do not claim to be deriving a new model; as the reviewer correctly notes, (26) is the standard (linear) additive effect model. Instead, Theorem 1 provides an **alternative, statistical interpretation** of the multiplicative / additive epistasis measures in terms of the additive effect model in (26). To our knowledge, this statistical interpretation of the epistasis measures has not been previously described.

Second, we emphasize that our results relating the Walsh coefficients to the parameters of the multivariate Bernoulli distribution (Theorem 2) have not been previously described in the genetics literature. To the best of our knowledge, none of the existing papers on measuring epistasis with Walsh coefficients (e.g. Weinreich et al., Curr Opin Genet Dev 2013) mention the moments of a multivariate Bernoulli distribution, as we describe in Theorem 2. If the reviewer knows of a paper that describes our results, we are happy to update the manuscript.

To address the reviewer's comments, we have updated the manuscript (see below) to emphasize that the MBD provides a statistical interpretation of epistasis:

Introduction:

We resolve the mathematical and biological inconsistencies between the different epistasis formulae by deriving connections between epistasis and the parameters of the multivariate Bernoulli distribution (MVB), a probability distribution on binary random variables [29]. In particular, we show that a wide array of approaches for quantifying epistasis – including the additive, multiplicative, and chimeric formulae, as well as the regression models commonly used in GWAS and QTL analyses [5, 2] and the Walsh coefficients for measuring background-averaged epistasis [30, 31, 32] – are equivalent to computing different parameterizations of the MVB, showing that the MVB provides a unifying statistical framework for the different epistasis measures. To our knowledge, the statistical interpretations of the various epistasis formulae that we derive have not been previously described in the literature.

Methods Section 2.3:

Thus, our results show that the additive and multiplicative epistasis measures are equivalent to computing interaction terms in regression models commonly used in genetics.

To sum up, they have a useful point about some erroneous analyses of fitness epistasis in microbial growth experiments, where a log transform should have been used to get to the correct scale, but this gets hidden in a mass of unnecessary detail.

We thank the reviewer for their positive feedback on one of the main messages of our manuscript. We emphasize that the point of our manuscript is not that existing studies – which use the chimeric measure – are using the *wrong* scale (additive vs multiplicative), but that the chimeric epistasis measure mixes **both** measurement scales. Thus, taking a log transform of the fitness measurements will not fix this underlying issue, as we demonstrate in Appendix G and Figure S2.

We also highlight that our mathematical framework, e.g. the multivariate Bernoulli model, has not been previously described in the literature, and we believe that such details would be useful for the field.

Specific comments

p.1 I.1 Epistasis is a property of any trait, not just fitness.

We have updated the Introduction to note that epistasis is a property of any quantitative trait.

Introduction:

While epistasis is a property of any quantitative trait, many studies have measured epistatic interactions using experimental fitness data from *haploid genomes* (reviewed in [6, 7, 8]). Most of these studies measure *pairwise*

I.7, 10 Synergistic vs antagonistic and negative versus positive are essentially different words for the same things; it's confusing to have both of them in the same abstract. Also, neither of them have been defined.

We removed the references to synergistic and antagonistic epistasis in the abstract and instead use “positive” and “negative”. We define synergistic and antagonistic in Section 2.6 when we

study interactions between antibiotics, as these terms are standard in the antibiotic interaction literature.

Results Section 2.6:

The discrepancy between the additive and chimeric measures may lead to different conclusions on the type of interactions between antibiotics, i.e. whether a given combination of antibiotics is *synergistic* (more effective at killing bacteria when taken together versus taken individually, i.e. a *negative* interaction) or *antagonistic* (less effective together versus individually, i.e. a *positive* interaction). For fifth-order interactions, the chimeric measure ϵ^C was

I.15 up Wright never used the term “fitness landscape”, although he did invent the concept of a surface of population mean fitness as a function of genotype frequencies, and peaks of fitness associated with individual genotypes.

We modified the reference to this citation by noting it was first “conceptualized” by Wright 1932:

1 Introduction

A key problem in biology is deriving the map from genotype to fitness, or the average reproductive success of a genotype. This map is often referred to as the *fitness landscape* (first conceptualized in [1]). In its simplest form,

I.11 up and following The reader will be confused by their defining epistasis in terms of fitness, and then citing material that applies it to other traits, e.g. ref. 3 does not mention fitness. They need to correct this; in fact, they could substitute ‘trait’ for fitness in much of what follows.

In the first paragraph of the introduction we now mention that epistasis applies to any quantitative trait, as we note in a previous response.

Moreover, we also removed our reference to Cordell (2002) (i.e. what was previously reference 3) and replaced it with two reviews from Ben Lehner that explicitly discuss studying epistasis using experimental fitness measurements from haploid genomes.

p.2 §.2 It would help to define what is meant by negative vs positive epistasis at this point. The references are not entirely accurate, e.g. ref 23 simply measured the (lack of) departure relation between their (incorrect) measure of relative fitness and number of accumulated mutations, rather than specific interactions between mutations.

We have updated the Introduction to explicitly define what positive and negative epistasis means.

Introduction:

While epistasis is a property of any quantitative trait, many studies have measured epistatic interactions using experimental fitness data from haploid genomes (reviewed in [6, 7, 8]). Most of these studies measure *pairwise epistasis*, or an interaction between a pair of genetic loci that is computed by comparing the observed fitness of the double-mutant to the expected fitness under a null model with no epistasis. The *sign* of the epistatic interaction is determined by whether the observed fitness is greater than or less than the expected fitness, resulting in a positive or negative interaction, respectively, and the choice of the null model for the expected fitness depends on the quantitative trait used as a proxy for epistasis. Nearly all formulae for pairwise epistasis assume either an additive null model, where the expected fitness is the sum $f_{01} + f_{10}$ of the fitness values of the single-mutants, or a multiplicative null model, where the expected fitness is the product $f_{01}f_{10}$ of the fitness values of the single-mutants. For example, an additive null model is often used when fitness is measured using cellular growth rate (e.g. in fitness assays of microbes [9]) or fluorescence (e.g. in proteins [10]), while a multiplicative null model is typically used when fitness is measured by the total size of a clonal population [11]. Under an additive null model, epistasis ϵ is computed as the difference $\epsilon = f_{11} - (f_{10} + f_{01})$ between observed and expected double-mutant fitness.

§3 The statement about recombination is incorrect; these papers deal with the effect of epistasis on selection for recombination modifiers, not the “quantification of recombination”.

We agree with the reviewer that our statement on recombination was incorrect. Thus, we have removed our discussion of recombination from the manuscript since it is not the focus of our manuscript.

p.3 l.13-14 up This is a purely haploid model; diploidy is much more complex (see General Comments).

We now state that our model assumes the genome is haploid.

Results Section 2.1:

2.1 Pairwise epistasis: additive, multiplicative, and chimeric

Pairwise epistasis describes interactions between two genetic loci. We consider haploid genomes and assume that each locus is *biallelic*, i.e. each locus has two alleles labeled 0 and 1. Thus for a pair of loci there are four possible

l.8 up They need to be more precise about what is meant by fitness (see General Comments); ‘mean reproductive success’ is not very meaningful for single-celled organisms reproducing by binary cell division, and rate of population growth (Malthusian parameter) is an appropriate measure. This paper is concerned entirely with data on such organisms (although this is never made explicit), so the authors might as well stick to this measure.

We have updated the Results Section 2.1 to give concrete examples about what is meant by fitness.

Results Section 2.1:

In practice, the fitness f of a genotype cannot be directly measured. Instead, experiments typically measure traits that are expected to be highly correlated with fitness. For example, in populations of microbes or proteins, it is standard to estimate the fitness of a genotype with either its population growth rate or its relative frequency, which are typically assumed to follow an additive or multiplicative scale, respectively (Methods). Accordingly, the two standard null models of fitness for measuring epistasis are the *additive* model and the *multiplicative* model.

Furthermore, as we previously noted, we added a section to the Methods discussing when an additive versus multiplicative fitness model is appropriate.

p.4-5 The material on this page could be presented much more concisely.

We have moved over half a page of material from Sections 2.1-2.2 to the Methods.

p.4 l.1 Ref. 64 is not the latest (1996) edition of this book.

We have updated the reference to the latest edition.

§5 l.5 Ref 23 does not use the chimeric method of estimating pairwise epistasis; I haven't checked the other references.

We respectfully disagree with the reviewer. Both Ref 23 (Elena and Lenski, Nature, 1997, now renumbered as 14 in the revised manuscript) and subsequent papers note that Ref 23 uses the chimeric formula.

In ref. 23 itself, in Figure 3 the authors visualize “Distribution of observed minus expected fitness values”, where they note that expected fitness values are generated assuming “assuming no interactions (multiplicative effects)”; see below for screenshots from the paper.

In our first experiment, genotypes with multiple mutations were produced by successive rounds of mutagenesis. By contrast, in this second experiment, we constructed genotypes with multiple mutations by recombining mutations from genotypes that had single mutations. Thus, we could measure the effect of each particular mutation alone and in combination with other mutations. We chose three mutations, which had a broad range of fitness effects, from each of the three sets of genotypes with single mutations. We used P1 transduction (see Methods) to construct all 27 combinations of two mutations from different sets. The fitness values for each double mutant and its corresponding single mutants were measured simultaneously, with 10-fold replication each. The paired fitness values for two single mutants were used to generate the expected fitness values for the double mutants, assuming no interactions (multiplicative effects).

Moreover, Segre et al. (Nature Genetics 2005, now reference 14) note that Elena and Lenski measure epistasis as $w_{XY} - w_X w_Y$; i.e. the chimeric formula; see below screenshot (ref. 14 is Elena and Lenski).

We first analyzed the distribution of deviations from this multiplicative behavior using previously proposed scales (Supplementary Fig. 1 online) and concentrated in particular on a conventional nonscaled measure of epistatic interactions: $\varepsilon = W_{XY} - W_X W_Y$ (refs. 5,14). In

If the reviewer has additional clarification on the formula used by Elena and Lenski, we are happy to amend our citation.

We emphasize that this use of the pairwise chimeric measure does not fundamentally change the conclusions of the Elena and Lenski paper, since the sign of the chimeric and multiplicative measures is the same in the pairwise case (Theorem 1).

p.5 Table 1 The Walsh coefficients apply to any quantitative trait, not one confined to (0, 1).

We updated Table 1 to note that the fitness values only need to be *proportional to the probabilities p*.

p.6 §2 I.7 This is incorrect; the frequency is proportional to the exponent of the product of time and growth rate; this, of course, assumes asexual reproduction, which is not specified.

We have removed this claim (about a population evolving over one unit of time) from the manuscript.

p.33 They are correct that the MVB type model can be use to represent multi-locus linkage disequilibrium, but the use of log-linear models for this has a long history (e.g. Smouse 1974 *Genetics* 76: 557), so there is nothing especially new here.

As the reviewer correctly notes, log-linear models have been long used to describe linkage disequilibrium. However, the purpose of Section 4.4.4 is to note the ways in which our multivariate Bernoulli model differs from these past works. In particular, we explain why some papers explicitly describe 3-way linkage disequilibrium with a formula that is identical to the 3-way chimeric formula.

Reviewer #3 (Remarks to the Author):

My comments on the noteworthiness and potential significance of the paper are largely the same as those of the reviewers in the previous round. The questions raised by the previous reviewers have been addressed convincingly by the authors. The work is sound and has been made broadly accessible through a number of specific applications and illustrations.

We thank the reviewer for their positive comments on our manuscript.

I have a few minor comments.

1. While reading section 2.4.1, I was initially confused by the fact there is discordance between the multiplicative epistasis measure and the β_S (as seen in Figure 2, for example), since Theorem 1 (Methods) shows that the two are identical. I had to dig into Methods to understand that the "noise" term is actually there *in addition* to the interaction terms, and is not considered as a source of "true" interaction. (This may be especially confusing to those who are familiar with the widely-used Rough Mount Fuji model, where the noise term is the only source of epistasis.) This also explained why there is a discordance for 2-way epistasis in the first subfigure of Figure 2A. It would be good if the authors clarified this in section 2.4.1.

We thank the reviewer for raising this point of confusion and for bringing the Rough Mount Fuji model to our attention. We have added a sentence to the end of Section 2.4.1 clarifying that the discordance is because of the multiplicative Gaussian noise, and we added a sentence to Methods Section 4.5 describing how our simulations differ from the Rough Mount Fuji model.

Results Section 2.4.1

multiplicative measure has zero error, i.e. $\log \epsilon^M = \beta$, matching our theoretical results (Theorem 1, Methods). (Note that Theorem 1 does not apply when there is multiplicative Gaussian noise, i.e. $\sigma > 0$, as this noise will cause the fitness values to not follow a log-linear model.)

2. I would suggest that the authors mention in the Abstract their finding that the chimeric formula falsely detects higher order epistasis when there is none (Section 2.4.2). This is a clear and striking demonstration and has greatly improved the manuscript.

We thank the reviewer for their positive feedback and for the helpful suggestion. We added a sentence to the abstract (highlighted below) stating that the chimeric formula may falsely detect higher-order epistasis.

Abstract

model interactions between the Bernoulli random variables. In simulations, we show that the additive/multiplicative epistasis formulae are more accurate than the chimeric formula, and that the chimeric formula may falsely detect higher-order epistasis that is not present in the data. Furthermore, using multi-gene knockout data in yeast, multi-way

3. The expansion in Eq. (26) is identical to the so-called "Taylor" series expansion of fitness landscapes, see for example (Weinberger, "Fourier and Taylor series on fitness landscapes.")

Biological cybernetics, 1991). The connection of the multiplicative measure to the Taylor expansion is therefore more direct than to the Fourier-Walsh decomposition.

We thank the reviewer for pointing out the connection between equation (26) and the “Taylor series expansion” of a fitness landscape. We have added the Weinberger reference below equation (26).

4. On page 33, authors say that "Theorem 2 also provides a connection between the multivariate Bernoulli distribution and the circuit formulae... as the circuit formulae for a full genotype space are linear combinations of the Walsh coefficients;". This connection is rather circuitous (please forgive the pun) and adds nothing substantive.

We removed this sentence from the manuscript.

5. Typo on page 3: "there is up to substantial (up to 60%)".

We fixed the typo.

6. On pages 6-7, the authors say that "prior literature on higher-order interactions does not rigorously define the chimeric epistasis measure...", and specifically point to Refs [2,42,43]. The complaint that there is lack of "rigor" is a bit uncharitable since the definition of chimeric epistasis has been clearly provided in the references, and the use of cumulants in [43] has been done with sufficient background to make the idea clear. Perhaps the most that can be said that the connections were made incompletely.

We agree with the reviewer's comments. We have updated the text to instead note that the prior works “do not provide a rigorous statistical interpretation of the chimeric epistasis measure”.

7. Typo near the bottom of page 29: I believe " $E[X_{12}]$ " should be " $E[X_1 X_2]$ ".

We fixed the typo.

We thank the reviewers for reviewing and responding to our response document. We have updated the manuscript with the changes requested by Reviewer 2.